# Plug-and-Play Fidelity Optimization for Diffusion Transformer Acceleration via Cumulative Error Minimization

**Tong Shao, Yusen Fu, Guoying Sun, Jingde Kong, Zhuotao Tian**[*]**, Jingyong Su**[*]
School of Computer Science and Technology
Harbin Institute of Technology, Shenzhen, China
{shaotong, 220110516, sunguoying, 2023113058}@stu.hit.edu.cn,
{tianzhuotao, sujingyong}@hit.edu.cn

## Abstract

Although Diffusion Transformer (DiT) has emerged as a predominant architecture for image and video generation, its iterative denoising process results in slow inference, which hinders broader applicability and development. Caching-based methods achieve training-free acceleration, while suffering from considerable computational error. Existing methods typically incorporate error correction strategies such as pruning or prediction to mitigate it. However, their fixed caching strategy fails to adapt to the complex error variations during denoising, which limits the full potential of error correction. To tackle this challenge, we propose a novel fidelity-optimization plugin for existing error correction methods via cumulative error minimization, named CEM. CEM predefines the error to characterize the sensitivity of model to acceleration jointly influenced by timesteps and cache intervals. Guided by this prior, we formulate a dynamic programming algorithm with cumulative error approximation for strategy optimization, which achieves the caching error minimization, resulting in a substantial improvement in generation fidelity. CEM is model-agnostic and exhibits strong generalization, which is adaptable to arbitrary acceleration budgets. It can be seamlessly integrated into existing error correction frameworks and quantized models without introducing any additional computational overhead. Extensive experiments conducted on nine generation models and quantized methods across three tasks demonstrate that CEM significantly improves generation fidelity of existing acceleration models, and outperforms the original generation performance on FLUX.1-dev, PixArt-$\alpha$, StableDiffusion1.5 and Hunyuan. Our code is released publicly at https://github.com/leaves162/CEM.

## 1 Introduction

Diffusion models Ho et al. (2020); Rombach et al. (2022) have significantly advanced visual generation tasks, including image Chen et al. (2023) and video Zheng et al. (2024) generation, and extended to other tasks Nie et al. (2025); Hu et al. (2025); Sun et al. (2025). Diffusion Transformers (DiT) Peebles & Xie (2023) have emerged as the dominant architecture for diffusion models, replacing U-Net Ronneberger et al. (2015) due to their inherent scalability and superior generative capabilities. Despite the impressive performance of these powerful models, their slow inference speed remains a critical barrier to widespread adoption. Currently, image generation typically requires tens of seconds, while video generation can take up to several minutes or even longer. This limitation primarily stems from the sequential nature of the reverse denoising process, which precludes parallel decoding. Moreover, the expansion of parameters, coupled with computationally intensive operations like attention Vaswani et al. (2017), further worsens efficiency.

To accelerate the DiT-based diffusion generation, researchers adopt techniques such as distillation Yin et al. (2024); Salimans & Ho (2022); Gu et al. (2025), learning Huang et al. (2024a);

---

[*]Corresponding authors.

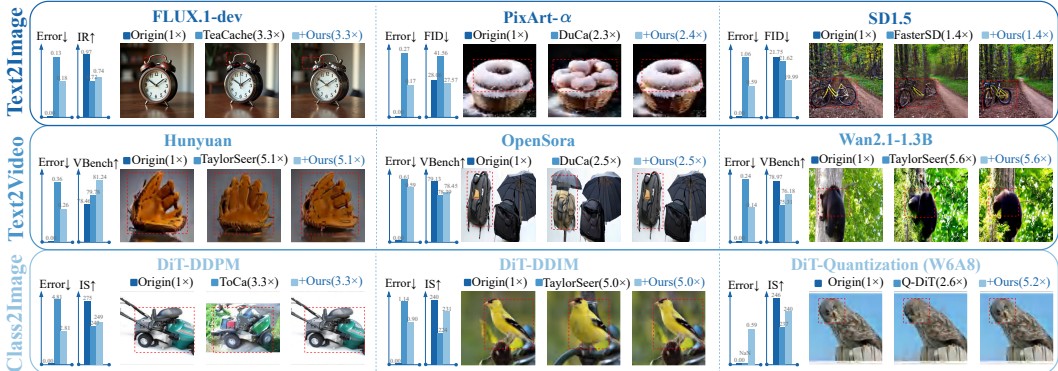

Figure 1: **Our CEM significantly reduces caching error while maintaining acceleration, thereby improving the generation fidelity of existing acceleration methods.** Comprehensive experiments demonstrate the effectiveness and generalization of CEM.

Ma et al. (2024a) and quantization Shang et al. (2023); He et al. (2023); Li et al. (2025), aiming to reduce inference latency by decreasing the model size. However, they necessitate training phases, which entails substantial computational cost, and lack generalization across models. An alternative approach is the caching mechanism Liu et al. (2025b); Saghatchian et al. (2025); Qiu et al. (2025), which leverages the similarity Ma et al. (2024b) between adjacent timesteps or layers to reuse previous cached hidden states, achieving training-free acceleration. Naive caching Selvaraju et al. (2024); Ma et al. (2024b) inevitably accumulates noise during the denoising, with error growing exponentially as the cache interval increases, resulting in a substantial deterioration in generation fidelity.

Therefore, existing caching optimization methods are combined with additional mechanisms to perform error correction, aiming to mitigate the cumulative caching error. For example, methods such as ToCa Zou et al. (2024a), DuCa Zou et al. (2024b), and FastCache Liu et al. (2025a) incorporate the pruning operation when reusing cache to retain the computation of a subset of important tokens, thereby reducing error. In addition, methods such as ICC Chen et al. (2025b) and TaylorSeer Liu et al. (2025c); Guan et al. (2025) leverage historical trends to predict the output variations across different timesteps and guide the update of cached representations, rather than directly reusing cache.

Although the idea of error correction partially reduces the cumulative error for balancing the generation quality and acceleration efficiency, its effectiveness is limited by the error of relatively simple or fixed caching strategies. For example, ToCa Zou et al. (2024a) and DuCa Zou et al. (2024b) adopt the cache schedule that changes linearly with the timestep, while the cache interval of methods like ICC Chen et al. (2025b) and TaylorSeer Liu et al. (2025c); Guan et al. (2025) is a constant. They are incapable of handling the complex dynamics of sensitivity to caching during denoising. *This limitation results in an insufficient characterization of the model's intrinsic sensitivity to caching, thereby preventing the mitigation of error accumulation during acceleration. Consequently, the potential of error correction is limited, leading to degradation in generation fidelity.*

To address this issue, we propose a training-free fidelity optimization plugin via **C**umulative **E**rror **M**inimization, **CEM**, that seamlessly integrates into existing acceleration methods and quantized models. CEM customizes an error to model the intrinsic sensitivity of generation models to different cache intervals during denoising, which is then leveraged as an offline prior. Guided by this prior, CEM employs dynamic programming to derive the optimal cache strategy under a given acceleration budget by minimizing the cumulative error. When integrated as a plugin into existing error correction methods and quantized models, CEM effectively mitigates the cumulative errors, significantly improves generation fidelity, and maintains or enhances their acceleration efficiency.

Although several prior works Qiu et al. (2025) like TeaCache Liu et al. (2025b) have explored caching optimization, they rely on real-time error estimation, introducing additional computational overhead that undermines efficiency gains. Furthermore, they are tied to specific acceleration ratios or model architectures, limiting compatibility with error correction. In contrast, our CEM conducts offline error modeling prior to inference, enabling the use of prior knowledge without incurring runtime cost. CEM is model-agnostic, supports arbitrary acceleration budgets, and integrates seamlessly with both error correction methods and quantized models.

Extensive experiments demonstrate that our CEM, when used as a plug-in, significantly improves the generation fidelity of existing acceleration models across eight generation models while preserving their acceleration efficiency. Specifically, CEM helps TaylorSeer, ToCa, DuCa and FasterSD achieve fidelity surpassing the original models of Hunyuan, FLUX.1-dev, PixArt-$\alpha$ and StableDiffusion1.5, respectively, without sacrificing their acceleration efficiency. In addition, CEM seamlessly integrates with quantized models, achieving a further 2× speed-up on Q-DiT beyond its original acceleration, along with enhanced generation fidelity. In summary, the contributions of this paper are fourfold:

- We propose a novel training-free and plug-in caching-strategy optimization method, CEM, that can be seamlessly integrated into existing error correction methods and quantized models, substantially improving generation fidelity while maintaining acceleration efficiency.
- We propose offline error modeling, which constructs the model's intrinsic sensitivity under different cache intervals through random sample generation. This offline prior subsequently guides the optimization of caching strategy, requiring no extra online computational cost.
- We introduce dynamic programming based on prior errors to derive the optimal caching strategy that minimizes cumulative error, thereby reducing the error of existing caching-based methods.
- Extensive experiments conducted on eight generation models and one quantized model demonstrate that CEM can be effectively employed as a plug-in to improve the generation fidelity of state-of-the-art acceleration methods and quantized models.

## 2  RELATED WORK

**Diffusion Transformer.** Diffusion models Ho et al. (2020); Rombach et al. (2022) have demonstrated remarkable capabilities in image Chen et al. (2023) and video Zheng et al. (2024); Kong et al. (2024) generation. Upon this, DiT Peebles & Xie (2023) replace the convolutional modules in U-Net Ronneberger et al. (2015) with attention Vaswani et al. (2017), which enhances scalability and leads to breakthrough improvements in generation quality. Despite these advancements, the iterative nature of the diffusion combined with the computational complexity of attention results in slow inference. As a result, acceleration techniques have emerged as a key focus of research.

**DiT Acceleration.** General acceleration techniques such as distillation Meng et al. (2023); Yao et al. (2024) and quantization Li et al. (2023b); Liu et al. (2025d) require retraining models, which incur substantial time costs and lacks generalization across models. Consequently, training-free methods have emerged as a promising direction. Caching-based methods Selvaraju et al. (2024); Zou et al. (2025); Lv et al. (2024) accelerate inference by reusing previous hidden states with high similarity, while the reuse of cache inevitably introduces error, which accumulate over time and result in a significant degradation of generation quality. To correct this error, two strategies have been proposed: (1) Token pruning, which retains the computation of a subset of tokens during cache reuse to reduce the error caused by fully relying on cache. For example, DuCa Zou et al. (2024b) applies pruning at specific timesteps to balance generation quality and acceleration efficiency. (2) Predictive reuse, which estimates future representations based on historical evolution of cache rather than directly reusing them. For instance, TaylorSeer Liu et al. (2025c); Guan et al. (2025) performs a taylor expansion on cache and utilizes gradient dynamics to predict features for reuse.

These methods alleviate caching error to some extent, they overlook the optimization of the caching strategy itself, corrections are applied on top of relatively high cumulative error, limiting their effectiveness. Although a few studies have explored strategy optimization, they are difficult to integrate with error correction for quality enhancement. For instance, AdaCache Kahatapitiya et al. (2024) relies on motion trajectories and is restricted to video; AdaptiveDiffusion Ye et al. (2024) introduces overhead from third-order calculations and is specific to the U-Net; TeaCache Liu et al. (2025b) performs real-time estimation and polynomial fitting, while the extra cost offsets the benefits of cache-based acceleration. To address these issues, we propose a plug-and-play acceleration framework that optimizes the caching strategy by offline prior, seamlessly integrates with error correction methods without computational overhead, substantially improving their generation fidelity.

In addition, optimizing the sampling strategy also leads to acceleration. For example, DDIM Song et al. (2020) introduces deterministic sampling to reduce the number of denoising steps. DPM-Solver Lu et al. (2022) and DPM-Solver++ Lu et al. (2025) leverage adaptive high-order solvers to accelerate denoising. Rectified flow Liu et al. (2022); Liu (2022) improves the transport of the distribution in the ordinary differential equation. Our method is orthogonal to these approaches and can be integrated across sampling strategies, even supporting compatibility with quantized models.

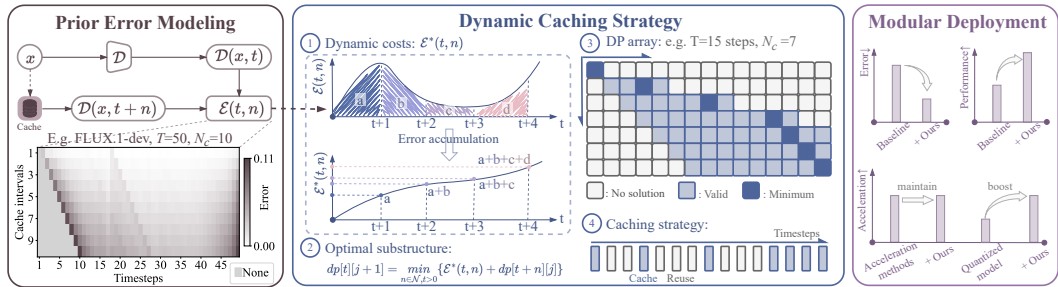

Figure 2: **Overview of our CEM framework.** It first performs **Offline Error Modeling** to characterize the model's intrinsic sensitivity to caching under different timesteps and cache intervals, forming an offline prior. Guided by this prior, it employs **Dynamic Caching Strategy** with dynamic programming to determine the optimal caching strategy that minimizes cumulative error and enhances generation fidelity. Finally, CEM supports **Plug-and-Play Deployment** and can be seamlessly integrated into existing error correction methods and quantized models.

## 3 METHODOLOGY

To mitigate the caching error and address the limitations of simplistic caching strategies in error correction methods, we propose a training-free, plug-and-play acceleration approach, CEM, that significantly improves generation fidelity while preserving the acceleration efficiency. As shown in Fig. 2, CEM first performs **(1) Offline Error Modeling**, where it models the joint influence of denoising timesteps and cache intervals on caching error distribution. Then, we introduce **(2) Dynamic Caching Strategy**, which selects a set of cache intervals that minimizes the cumulative error through dynamic programming. Finally, **(3) Plug-and-Play Deployment** integrates the derived caching strategy seamlessly into the error correction methods, thereby improving generation fidelity.

### 3.1 OFFLINE ERROR MODELING

To maintain acceleration efficiency without incurring the computation overhead of real-time caching strategies, we propose offline error modeling that models caching error by analyzing it under different caching strategies on randomly generated content before inference.

**Error definition.** We model the error distribution by considering the joint variation of denoising steps and cache intervals, rather than relying solely on temporal changes as in real-time methods. Specifically, given the cache interval $n$ (perform caching every $n$ timesteps) and the current timestep $t$, we denote the output of the diffusion model with input $x$ at a specific timestep $t$ as $\mathcal{D}(x, t)$. Furthermore, we flatten $\mathcal{D}$ into $\mathcal{D}'$ and compute the Cosine loss between the ground-truth output of $t$ and the cached output of previous timestep $t + n$ via a normalized inner product:

$$\mathcal{E}(t, n) = \frac{1}{N_s} \sum_{i=0}^{N_s-1} [1 - \frac{\mathcal{D}'(x, t) \cdot \mathcal{D}'(x, t+n)}{||\mathcal{D}'(x, t)||_2 \cdot ||\mathcal{D}'(x, t+n)||_2}], \tag{1}$$

where $\mathcal{E}$ is our error, $N_s$ denotes the number of generation, see Appendix. B.1 for error visualization.

**Offline modeling.** The modeling captures complex intrinsic error variations and provides the basis for adaptive caching by sensitivity errors. However, if the modeling is performed online, it inevitably needs additional computational overhead, hindering acceleration efficiency. Moreover, the sensitivity is model-intrinsic, making repeated modeling during inference redundant and wasteful.

*Therefore, CEM performs the modeling offline by generating multiple ($N_s$ times in Eq. 1) random contents and averaging them, storing it as intrinsic prior. It makes the modeling content-agnostic, so each generation model only needs to be modeled once for permanent use.*

**Error distribution consistency.** A key implicit assumption of error modeling is: *The statistics from randomly generated samples are representative of those encountered during actual inference.* To validate it, we analyze the variation of error distributions across different samples in Fig. 3:

- In Fig. 3(a), the error exhibits small variances across different cache intervals, showing that the variation of both content and cache interval exert only minor effects on the error distribution.

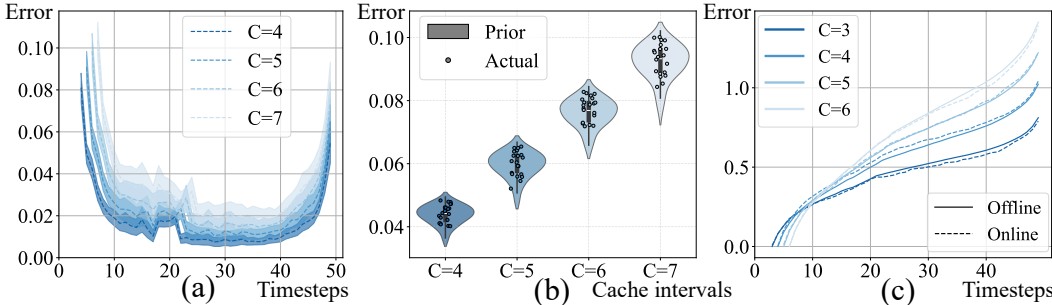

Figure 3: **Error Analysis. (a). Mean-variance of offline error modeling under different cache intervals.** The error variance remains relatively small across various contents and cache intervals. **(b). Consistency between offline modeling and actual inference.** The error points obtained during inference fall within the prior-modeled distribution, indicating strong consistency between prior modeling and real inference. **(c). Offline cumulative error vs. online error.** The cumulative error approximation accurately captures the trend of error variation during actual inference.

- In Fig. 3(b), the errors from actual inference lie within the range of the error distribution from offline modeling, demonstrating that this error distribution remains highly consistent between prior modeling and actual inference scenarios.

These results both demonstrate the robustness of the error distribution to generated content. Furthermore, we conduct extended experiments in Appendix. B.2 using different prompt sources for offline modeling, and observe consistent behavior during inference. This provides further evidence that the error distribution captures the model's intrinsic sensitivity to acceleration, which is content-agnostic. Refer to Sec. 4.3 for more results on robustness, offline cost and the effect of random sample size.

**Joint modeling analysis.** We model the error as a joint function of denoising timesteps and cache intervals. Previous methods Zou et al. (2024a;b); Liu et al. (2025c) only consider differences across timesteps and determine cache intervals heuristically, limiting control over the acceleration budget. In contrast, we explicitly model cache intervals together with timesteps to capture the error distribution and analyze the sensitivity of different denoising stages. This formulation is simple yet accurate, offering flexible acceleration control, while its offline design introduces no additional computational overhead and provides the foundation for our algorithm of cumulative error minimization in Sec. 3.2.

## 3.2 DYNAMIC CACHING STRATEGY

After introducing offline error modeling to quantify the error, we can further leverage these errors to evaluate entire caching strategies, i.e., the combinations of cache intervals throughout the denoising. It inherently possesses an optimal substructure: the minimum cumulative error at each caching operation builds upon the optimal result of the preceding operation. So CEM employs dynamic programming, as shown in Fig. 2, to optimize combination of cache intervals.

**Problem setup.** Given a acceleration budget ($N_c$), let $n$ denote the current cache interval, $t + n$ and $t$ represent the timesteps with full computation with same setting as Sec. 3.1.

**Cumulative error approximation.** Naturally, the modeled error serves as the dynamic cost for dynamic programming. Given a fixed number of $N_c$, we compute the optimal combination of cache intervals that minimizes the error. However, the error in cache acceleration is a continuously accumulating process, which is not considered in the offline error modeling. This is because direct modeling of cumulative error is exponentially inefficient: Simple error modeling completes all timesteps in one forward pass, while cumulative modeling needs $T$ passes for $T$ timesteps.

To incorporate cumulative error effects without sacrificing efficiency, we introduce cumulative error approximation. It estimates the cumulative error based on $\mathcal{E}(t, n)$, achieving a balance between error fidelity and modeling efficiency. Specifically, we observe that a simple cumulative integral over the $\mathcal{E}(t, n)$ yields an effective approximation. The cumulative error $\mathcal{E}^*(t, n)$ at an arbitrary timestep is:

$$\mathcal{E}^*(t, n) = \text{CUMSUM}(\mathcal{E}(t, n), dim = 0), \tag{2}$$

where CUMSUM($\cdot$) denotes the cumulative integral in Numpy and we apply weighting factors to amplify the differences across cache intervals. As shown in Fig. 3(c), this simple approximation

closely matches the actual error. It can be attributed to the high structural similarity between the input and output of each module in DiT, which causes input perturbations to propagate to the output and accumulate across timesteps. Refer to Appendix. C.2 for more analysis.

**Optimal substructure.** Given the specific constraint on acceleration efficiency (the number of caching $N_c$), we construct a dynamic programming array $dp[t][j]$ based on the $\mathcal{E}^*(t,n)$ described above. The array $dp[t][j]$ represents the minimum total caching error when denoising from beginning up to timestep $t$ with $j$ caching operations. We formulate the optimal substructure based on $dp[t][j]$:

$$dp[t][j+1] = \min_{n \in \mathcal{N}, t>0}\{\mathcal{E}^*(t,n) + dp[t+n][j]\}, j \in [1, N_c], t \in [T, 1], \tag{3}$$

where $\mathcal{N}$ denotes the set of candidate cache intervals. Our final objective is to solve for the minimum value of $dp[1][N_c]$. Then, a backtracking procedure is employed to recover the positions of the selected timesteps and associated cache intervals. See Appendix C.1 for implementation details.

**Analysis.** Dynamic programming enables CEM to obtain caching strategy with optimal error. As shown in Fig. 1, it effectively reduces the sensitivity error and improves generation fidelity under the same acceleration efficiency. The algorithm operates on offline errors, so it can derive the optimal cache-interval combination that can be shared across multiple generations without additional overhead given an acceleration budget, analysis can be found in Sec. 4.3 and Appendix.

### 3.3 PLUG-AND-PLAY DEPLOYMENT

Our optimized caching strategy can replace the existing caching components in current acceleration methods, significantly improving generation fidelity while maintaining the acceleration efficiency. It can also be directly applied to generation or quantized models to achieve high-fidelity acceleration. As illustrated in Sec. 4, CEM can be integrated with pruning-based methods such as DuCa and ToCa, and can also enhance prediction-based approaches like TaylorSeer in error correction. Moreover, CEM can also be directly incorporated into the generation process of quantized models such as Q-DiT, effectively doubling its acceleration. In summary, our method is compatible with various generation models, acceleration techniques, and quantized models.

## 4 EXPERIMENT

### 4.1 EXPERIMENT SETTINGS

**Models and baselines.** We conduct extensive experiments across three tasks: text-to-image generation (using StableDiffusion1.5 (SD1.5) Rombach et al. (2022), PixArt-$\alpha$ Chen et al. (2023) and FLUX.1-dev Labs (2024)), text-to-video generation (Hunyuan Kong et al. (2024), Wan2.1-1.3B Wan et al. (2025) and OpenSora Zheng et al. (2024)), and class-to-image generation (DiT-XL/2 with DDPM Ho et al. (2020) and DDIM Song et al. (2020) samplers). Our method is integrated into five SOTA acceleration methods: FasterSD Li et al. (2023a), TeaCache Liu et al. (2025b), ToCa Zou et al. (2024a), DuCa Zou et al. (2024b) and TaylorSeer Liu et al. (2025c). In addition, we ensure compatibility with quantized models by applying our approach to the Q-DiT Chen et al. (2025a).

**Evaluation and metrics.** For text-to-image generation, we use captions of MS-COCO2017 Lin et al. (2014) to generate images and evaluate performance using FID Heusel et al. (2017), CLIP-Score (CLIP) Hessel et al. (2021), ImageReward (IR) Xu et al. (2023), Peak Signal-to-Noise Ratio (PSNR), Structural Similarity Index Measure (SSIM) and Learned Perceptual Image Patch Similarity (LPIPS). Note that variations in sample counts (10K or 50K) and CLIP versions stem from differences in the respective acceleration baselines. For text-to-video generation, we report the average performance across 16 tasks from the VBench Huang et al. (2024b). For the class-to-image generation, we randomly sample 50,000 class labels from ImageNet and compute FID, sFID, InceptionScore (IS) Salimans et al. (2016), Precision (P) and Recall (R). Finally, we evaluate generation speed on RTX4090 or A800 using both FLOPs and latency. Refer to Appendix. D.1 for more details.

### 4.2 MAIN RESULTS

**Text-to-image generation.** In the results shown in Tab. 1, our CEM is integrated into four acceleration methods across three different generation models. Our method improves FasterSD of

Table 1: **Quantitative comparison on text-to-image generation.** ↓/↑ denotes lower/higher values indicate superior performance. "-" denotes the absence of reference results. "+Ours" indicates the baseline with our CEM. **Bold** font highlights our better results.

| | FLOPs (T)↓ | Latency (s)↓ | FID 10K↓ | CLIP (L)↑ | PSNR ↑ | SSIM ↑ | LPIPS ↓ |
|---|---|---|---|---|---|---|---|
| StableDiffusion1.5, DDIM 50 steps, 512×512 | | | | | | | |
| Origin | 37.05 | 1.44 | 21.75 | 30.92 | INF | 1.00 | 0.00 |
| 50% steps | 18.53 | 0.73 | 25.21 | 32.15 | 20.19 | 0.62 | 0.26 |
| FasterSD | 27.35 | 0.33 | 21.62 | 32.54 | 16.42 | 0.56 | 0.36 |
| **+Ours** | **27.35** | **0.33** | **19.99** | **32.85** | **15.77** | **0.60** | **0.35** |

| | FLOPs (T)↓ | Latency (s)↓ | FID 50K↓ | CLIP (L)↑ | PSNR ↑ | SSIM ↑ | LPIPS ↓ |
|---|---|---|---|---|---|---|---|
| PixArt-α, DPM-Solver 20 steps, 256×256 | | | | | | | |
| Origin | 11.18 | 0.86 | 28.06 | 16.29 | INF | 1.00 | 0.00 |
| 50% steps | 5.59 | 0.43 | 37.41 | 15.82 | 18.67 | 0.70 | 0.20 |
| FORA | 5.66 | 0.52 | 29.67 | 16.40 | - | - | - |
| ToCa | 4.26 | 0.44 | 29.73 | 16.45 | - | - | - |
| DuCa($N_5$) | 4.79 | 0.40 | 41.56 | 16.46 | 14.96 | 0.46 | 0.42 |
| **+Ours** | **4.75** | **0.39** | **27.57** | **16.37** | **18.25** | **0.68** | **0.21** |

| | FLOPs (T)↓ | Latency (s)↓ | IR ↑ | CLIP (G)↑ | PSNR ↑ | SSIM ↑ | LPIPS ↓ |
|---|---|---|---|---|---|---|---|
| FLUX.1-dev, Rectified Flow 50 steps, 1024×1024 | | | | | | | |
| Origin | 3719.50 | 35.63 | 0.9649 | 32.57 | INF | 1.00 | 0.00 |
| 25% steps | 967.07 | 8.91 | 0.9310 | 32.72 | 14.71 | 0.58 | 0.46 |
| FORA | 1320.07 | 14.66 | 0.9227 | - | - | - | - |
| ToCa($N_4$) | 1263.22 | 14.60 | 0.9822 | 32.36 | 18.27 | 0.67 | 0.30 |
| **+Ours** | **1263.22** | **14.13** | **1.0151** | **32.67** | **17.72** | **0.67** | **0.31** |
| TeaCache($l_{0.6}$) | 1115.85 | 16.57 | 0.7228 | 30.66 | 17.41 | 0.70 | 0.35 |
| **+Ours** | **1115.85** | **16.05** | **0.7362** | **31.13** | **17.89** | **0.71** | **0.33** |
| TaylorSeer($N_6$) | 744.81 | 10.09 | 0.9410 | 32.57 | 15.59 | 0.60 | 0.41 |
| **+Ours** | **744.81** | **10.09** | **0.9811** | **32.89** | **16.11** | **0.61** | **0.39** |

Table 2: **Quantitative comparison on text-to-video generation.** N (or l): baseline hyperparameter controlling acceleration efficiency.

| | FLOPs (T)↓ | Latency (s)↓ | VBench (%)↑ |
|---|---|---|---|
| OpenSora, rflow 30 steps, 480P, 51 frames | | | |
| Origin | 3283.20 | 102.29 | 79.13 |
| 50% steps | 1641.60 | 51.16 | 76.55 |
| Δ-DiT | 3166.47 | 98.31 | 78.21 |
| ToCa | 1394.03 | 54.70 | 78.34 |
| DuCa($N_3$) | 1315.62 | 50.64 | 78.39 |
| **+Ours** | 1315.62 | **50.24** | **78.45** |

| | FLOPs (T)↓ | Latency (s)↓ | VBench (%)↑ |
|---|---|---|---|
| Hunyuan, flow-solver 50 steps, 480P, 65 frames | | | |
| Origin | 29773.00 | 441.76 | 78.46 |
| 25% steps | 7741.11 | 111.13 | 70.89 |
| FORA | 5960.40 | 93.92 | 78.83 |
| ToCa | 7006.20 | 109.89 | 78.86 |
| TeaCache($l_{0.4}$) | 6550.06 | 108.54 | 77.56 |
| **+Ours** | **6550.06** | **105.94** | **78.15** |
| TaylorSeer($N_6$) | 5939.10 | 95.22 | 79.78 |
| **+Ours** | **5939.10** | **95.01** | **81.24** |

| | FLOPs Spe.↑ | Latency (s)↓ | VBench (%)↑ |
|---|---|---|---|
| Wan2.1-1.3B, unipc 50 steps, 480P, 65 frames | | | |
| Origin | 1.00× | 187.46 | 78.97 |
| 25% steps | 3.85× | 48.97 | 61.64 |
| Δ-DiT | 1.24× | 165.89 | 75.97 |
| TaylorSeer($N_6$) | 5.56× | 39.55 | 75.31 |
| **+Ours** | **5.56×** | **39.37** | **76.18** |

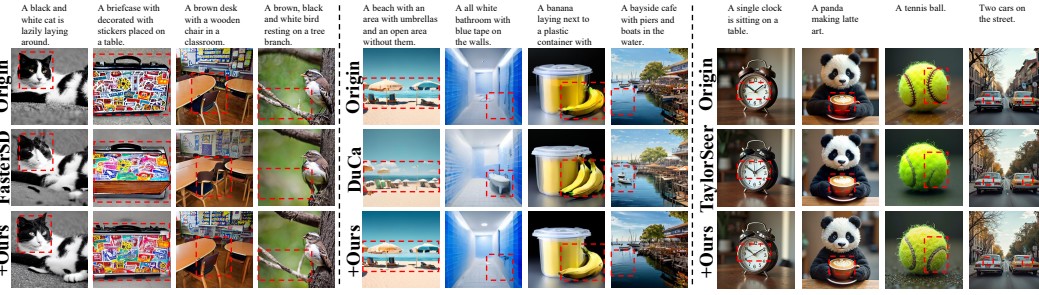

**(a) StableDiffusion1.5**   **(b) PixArt-α**   **(c) FLUX.1-dev**

Figure 4: **Qualitative visualization comparison on text-to-image generation.** We highlight the areas with red dashed boxes to emphasize the comparison. Our CEM achieves higher generation fidelity under the same or higher acceleration efficiency compared with baselines. More experiment results and visualizations are provided in the Appendix. D.2, E.1 and E.2.

SD1.5 by 1.63 on FID and 0.31 on CLIP, surpassing the performance of the original model. Moreover, these improvements come at no additional computational cost, as both FLOPs and latency remain unchanged. On PixArt-α, our enhancement of the DuCa leads to a 13.99 improvement in FID. Furthermore, thanks to our offline cache strategy optimization, we further reduce latency while maintaining the same FLOPs, ensuring improved efficiency without additional computational cost. Similarly, we improve both ToCa and TaylorSeer on FLUX.1-dev. In addition, we integrate our CEM with the online optimization technique TeaCache, achieving a 0.0134 improvement in IR while eliminating its extra computation during inference, thus reducing inference latency.

These text-to-image models span diverse backbones, sampling strategies and resolutions, demonstrating the generalization of CEM, even beyond DiT (SD1.5 is UNet). As shown in the qualitative results in Fig. 4, our CEM produces higher-fidelity images that closely resemble those of the original model, like the cat's face in (a), the umbrella in (b), and the texture of the tennis ball in (c).

**Text-to-video generation.** We validate the effectiveness of our plug-in on Hunyuan, Wan21 and OpenSora. Taking Hunyuan as an example, we improve TaylorSeer in Tab. 2, achieving VBench gains of 1.46 at N=6 (Our method replaces the design of N, so when combined with our CEM, the baseline no longer requires the N to be specified). These improvements significantly improve the generation fidelity of the acceleration models, even surpassing the original model performance

Table 3: **Quantitative comparison on class-to-image generation with DiT-XL/2 and quantized model.** W/A denotes the quantization bit-width of weights and activations. More experiment results and visualizations are provided in the Appendix. D.2 and E.4.

| | Bit-width (W/A)↓ | Size (MB)↓ | Latency (s)↓ | FID↓ | sFID↓ | IS↑ | P↑ | R↑ | PSNR↑ | SSIM↑ | LPIPS↓ |
|---|---|---|---|---|---|---|---|---|---|---|---|
| Origin | 16/16 | 1349 | 0.62 | 5.31 | 17.61 | 245.85 | 0.81 | 0.68 | INF | 1.00 | 0.00 |
| Q-DiT | 6/8 | 518 | 0.45 | 5.44 | 17.61 | 237.34 | 0.80 | 0.68 | 31.10 | 0.95 | 0.04 |
| **+Ours** | 6/8 | **518** | **0.22** | 5.51 | **17.49** | **240.36** | **0.80** | **0.68** | 31.06 | 0.93 | 0.05 |
| Q-DiT | 4/8 | 347 | 0.39 | 6.31 | 17.81 | 209.30 | 0.76 | 0.69 | 24.88 | 0.82 | 0.14 |
| **+Ours** | 4/8 | **347** | **0.20** | 6.20 | 17.62 | 213.50 | 0.76 | 0.69 | 24.99 | 0.82 | 0.14 |
| | Sampling /Steps | FLOPs (T)↓ | Latency (s)↓ | FID↓ | sFID↓ | IS↑ | P↑ | R↑ | PSNR↑ | SSIM↑ | LPIPS↓ |
| Origin | DDPM/250 | 118.68 | 2.51 | 2.23 | 4.57 | 275.64 | 0.83 | 0.58 | INF | 1.00 | 0.00 |
| 50% steps | DDPM/250 | 59.34 | 1.26 | 2.42 | 5.04 | 270.40 | 0.82 | 0.57 | 8.55 | 0.14 | 0.78 |
| FORA | DDPM/250 | 39.95 | 1.01 | 2.80 | 6.21 | - | 0.80 | 0.59 | - | - | - |
| ToCa($N_6$) | DDPM/250 | 36.30 | 0.84 | 3.08 | 6.58 | 246.59 | 0.79 | 0.59 | 20.92 | 0.71 | 0.21 |
| **+Ours** | DDPM/250 | 36.48 | **0.82** | 3.09 | **6.00** | **248.58** | **0.80** | **0.59** | **22.63** | **0.76** | **0.16** |
| Origin | DDIM/50 | 23.74 | 0.53 | 2.25 | 4.33 | 239.93 | 0.80 | 0.59 | INF | 1.00 | 0.00 |
| 33% steps | DDIM/50 | 8.07 | 0.18 | 4.24 | 5.52 | 214.35 | 0.77 | 0.56 | 9.22 | 0.17 | 0.81 |
| AdaCache | DDIM/50 | - | 0.46 | 4.64 | - | - | - | - | - | - | - |
| LazyDiT | DDIM/50 | 11.93 | 0.28 | 2.70 | 4.47 | 237.03 | 0.80 | 0.59 | - | - | - |
| ToCa($N_5$) | DDIM/50 | 7.44 | 0.20 | 6.37 | 7.09 | 199.48 | 0.74 | 0.53 | 16.56 | 0.53 | 0.40 |
| **+Ours** | DDIM/50 | 7.14 | **0.18** | 4.68 | 6.41 | 212.13 | 0.77 | 0.55 | 21.59 | 0.72 | 0.20 |
| DuCa($N_5$) | DDIM/50 | 6.32 | 0.17 | 6.07 | 6.64 | 199.64 | 0.74 | 0.52 | 16.63 | 0.53 | 0.39 |
| **+Ours** | DDIM/50 | 6.73 | **0.17** | **3.96** | **5.87** | **218.66** | **0.78** | **0.55** | **23.00** | **0.76** | **0.16** |
| TaylorSeer($N_6$) | DDIM/50 | 4.76 | 0.14 | 3.56 | 7.52 | 223.83 | 0.79 | 0.56 | 24.69 | 0.80 | 0.13 |
| **+Ours** | DDIM/50 | **4.76** | **0.13** | **3.08** | 6.43 | **231.10** | **0.80** | **0.57** | **25.64** | **0.83** | **0.10** |

Figure 5: **Qualitative visualization comparison on Hunyuan.** Our CEM improves the TaylorSeer for better consistency with the original model. See Appendix. E.3 for more visualizations.

of 80.66. Our method significantly reduces the cumulative error in caching strategies through offline error modeling and dynamic programming, without introducing any additional computational overhead, which is supported by the latency and performance metrics shown in the Tab. 2.

Furthermore, the visualization examples in Fig. 5(a) also demonstrate that CEM is capable of better preserving fine-grained generation details, such as the baseball glove and the village by the beach. Detailed VBench results in Fig. 5(a) also show that our method significantly enhances video generation quality on TaylorSeer, particularly in multi-objects, semantics and quality evaluations.

**Class-to-image generation.** In Tab. 3, we report the results of DiT-XL/2 under DDPM and DDIM sampling strategies. Under DDPM, our CEM improves ToCa by 0.58 on sFID and 1.99 on IS, respectively. Under the more commonly used DDIM, we explore optimizations for ToCa, DuCa and TaylorSeer. Taking DuCa as an example, our CEM achieves more than a 3× acceleration, reducing FID by 2.11, increasing IS by 19.02, and improving PSNR by 6.37. The performance gain mainly arises from the fact that the baseline method inevitably introduces more caching error as the acceleration efficiency increases, leading to a sharp degradation in fidelity. In contrast, our CEM effectively improves generation fidelity by combining offline error modeling with dynamic programming to identify the caching strategy that minimizes cumulative error.

In addition, to demonstrate the effectiveness and generalization of our method, we further integrate it into the quantized model Q-DiT. As shown in Tab. 3, under the manually specified 2× acceleration budget ($N_c$=25 in total 50 steps), our CEM not only further speeds up inference on top of the quantized model, but also improves the generation fidelity. Taking IS as an example, we improve it by 3.02 and 4.20 on W6A8 and W4A8, respectively.

In summary, on DiT-XL/2, our method not only improves the fidelity of existing acceleration models without compromising their efficiency, but also supports direct integration with quantized models, achieving substantial reductions in both computational time and memory usage.

Table 4: **Ablation results on three generation models.** "Vanilla" is naive caching with fixed cache intervals. "DCS" is Dynamic Caching Strategy, and "CEA" is Cumulative Error Approximation.

| PixArt-$\alpha$ | FID↓ | | Hunyuan | VBench(%)↑ | | DiT-XL/2 | FID↓ | IS↑ |
|---|---|---|---|---|---|---|---|---|
| Vanilla | 30.04 | | Vanilla | 77.64 | | Vanilla | 3.83 | 213.12 |
| +DCS | 28.69 | | +DCS | 79.21 | | +DCS | 2.73 | 234.78 |
| +DCS w/ CEA | **27.94** | | +DCS w/ CEA | **80.44** | | +DCS w/ CEA | **2.65** | **235.11** |

## 4.3 ABLATION STUDIES

**Module effectiveness.** To demonstrate the effectiveness of CEM, we conduct ablation studies on three generation tasks (one model per task). As the offline error modeling cannot be ablated in isolation, we perform ablation primarily on the Dynamic Caching Strategy (DCS) built upon it, with particular attention to the effects of the Cumulative Error Approximation (CEA).

As shown in Tab. 4, the dynamic programming in our DCS module significantly improves model performance. It improves the FID by 1.35 on PixArt-$\alpha$, increases the VBench by 1.57 on Hunyuan, and achieves improvements of 1.1 on FID and 21.66 on IS on DiT-XL/2. On the one hand, it demonstrates that the dynamic programming algorithm effectively optimizes the caching error, thereby improving the generation fidelity of generation models while maintaining acceleration. On the other hand, these results provide indirect evidence that our offline error modeling can effectively learn intrinsic caching error distributions of generation models, and apply them to actual inference acceleration without incurring any additional computational cost.

Furthermore, by incorporating the CEA in DCS module, we observe further performance improvements across all three models. The performance increases by 0.75, 1.23, 0.08 and 0.33 across the four metrics on the three models after leveraging cumulative integral to better fit the caching error. We believe this is mainly attributed to the approximate estimation of caching error, which also plays a smoothing role to some extent, reducing the impact of data variations on the error distribution.

**Comparison of error definitions.** In the offline error modeling, CEM defines the feature difference between the current and previous timesteps as the caching error, based on which it builds the offline modeling and subsequent dynamic programming process. For error formulation, we compare three commonly used distance measures: L1, L2 and Cosine distance. As shown in Fig. 6(a), the Cosine distance adopted in our method clearly outperforms the other two. We attribute this advantage to the fact that Cosine distance ignores the scale variance of features during computation, making it more robust to the sparse feature representations in DiT.

Table 5: **Cost analysis of our OEM and DCS module.** See the Appendix. B.4, C.3 for the cost of all models.

| Models | FLUX.1-dev | Hunyuan | DiT-XL/2 |
|---|---|---|---|
| Time of only generation | 1.92h | 4.72h | 19.63m |
| Time of OEM | 2.08h | 5.21h | 25.52m |
| Memory of only generation | 43.42GB | 57.36GB | 4.09GB |
| Memory of OEM | 53.06GB | 72.62GB | 4.65GB |
| Time of DCS | 1.10ms | 0.71ms | 1.13ms |
| Memory of error | 0.88KB | 0.88KB | 0.88KB |

**Offline modeling cost.** Our offline error modeling is obtained through random content generation before the actual inference, and it is performed only once for each generation model for permanent use. We report the cost of the offline process in Tab. 5. As this modeling relies on random content generation, which inherently requires time and memory resources, we first list the cost of generation only and then the cost of our offline error modeling (OEM). For the three major generation models reported in Tab. 5, the additional time overhead of OEM mainly comes from computing caching errors across different timesteps and cache intervals. The extra memory overhead primarily arises from storing the feature representations of previous timesteps for subsequent error computation.

It should be noted that this cost pertains solely to the offline modeling stage. During inference, CEM only needs to load a precomputed error matrix (of size N×T, where N is the number of cache intervals and T the number of timesteps) for dynamic programming. As shown in the lower part of Tab. 5, CEM introduces negligible overhead in both time and memory. Moreover, the dynamic programming solution is computed once for a given acceleration setting and can be reused across multiple generations, further reducing batch-generation cost.

**Effect of the number of random samples.** Our offline error modeling is derived from the random generation process and, as shown in Fig. 3, the estimated prior error remains consistent with that observed during actual inference. We further analyze the influence of the number of offline samples on model performance. As illustrated in Fig. 6(b) (more analysis in Appendix. B.3), the modeling shows minimal sensitivity to sample count, the fidelity on FLUX.1-dev quickly converges once more than 10 samples. This indicates that the error reflects a model-intrinsic sensitivity to caching, rather

Table 6: **Robustness of our CEM across different seeds, CFGs, resolutions, and frames.** We conduct all tests of TaylorSeer(N=6) on the FLUX.1-dev for consistency.

| Seed | Model | IR↑ | CLIP(G)↑ | CFG | Model | IR↑ | CLIP(G)↑ | Size | Model | IR↑ | CLIP(G)↑ | Frames | Model | VBench(%)↑ |
|---|---|---|---|---|---|---|---|---|---|---|---|---|---|---|
| 0 | TaylorSeer | 0.8760 | 32.17 | 3.5 | TaylorSeer | 0.8760 | 32.17 | 256 | TaylorSeer | 0.5756 | 30.97 | 33 | TeaCache | 77.88 |
| | **+Ours** | **0.9205** | **32.66** | | **+Ours** | **0.9205** | **32.66** | | **+Ours** | **0.6792** | **31.40** | | **+Ours** | **77.96** |
| 42 | TaylorSeer | 0.8625 | 32.38 | 5.5 | TaylorSeer | 0.6571 | 31.34 | 512 | TaylorSeer | 0.7495 | 31.80 | 49 | TeaCache | 77.81 |
| | **+Ours** | **0.9405** | **32.86** | | **+Ours** | **0.8867** | **32.76** | | **+Ours** | **0.7700** | **32.31** | | **+Ours** | **78.01** |
| 2025 | TaylorSeer | 0.8169 | 32.50 | 7.5 | TaylorSeer | 0.6924 | 31.75 | 1024 | TaylorSeer | 0.8760 | 32.17 | 65 | TeaCache | 77.56 |
| | **+Ours** | **0.8875** | **32.69** | | **+Ours** | **0.7336** | **32.29** | | **+Ours** | **0.9205** | **32.66** | | **+Ours** | **78.15** |
| 3407 | TaylorSeer | 0.8217 | 31.95 | 9.5 | TaylorSeer | 0.6794 | 31.20 | 2048 | TaylorSeer | 0.1552 | 31.26 | 129 | TeaCache | 76.21 |
| | **+Ours** | **0.9118** | **32.99** | | **+Ours** | **0.7935** | **32.17** | | **+Ours** | **0.2564** | **31.31** | | **+Ours** | **77.31** |

Figure 6: **(a). Comparison of generation fidelity with different error construction methods on DiT-XL/2.** The Cosine distance adopted in our CEM achieves the best generation fidelity. **(b). Effect of offline sample size on generation fidelity.** The offline error modeling captures the intrinsic sensitivity of the model to acceleration, which is content-agnostic and insensitive to the number of offline samples. **(c). Relationship between error and generation fidelity under different cache strategies.** Our quantitative analysis reveals a negative correlation between generation fidelity and caching error, with the curvature of this relationship differing under various acceleration efficiencies.

than content dependence. Additional experiments in the Appendix. B.2, where offline samples are drawn from different datasets but evaluated on the same benchmark, further validate this observation.

**The robustness of our CEM.** To further demonstrate the robustness of our CEM and the offline error modeling, we extend the precomputed errors to various generation settings, as illustrated in Tab. 6, including different seeds, CFG, resolutions and frames. It can be clearly observed that the modeled errors are insensitive to such variations in generation settings, consistently exhibiting strong robustness of our CEM across diverse configurations.

**Relationship between error and generation fidelity.** Finally, by constructing the errors of caching strategies, we can quantitatively and intuitively analyze the relationship between caching strategies and their generation fidelity under the same acceleration efficiency. As shown in Fig. 6(c), multiple candidate caching strategies exist under same acceleration efficiency. We plot their corresponding error and fidelity, and fit a relationship curve between them. We observe that as the error of caching strategy increases, the generation fidelity tends to decrease, and the curvature of this relationship varies with acceleration efficiency. This observation aligns with the empirical understanding, while our CEM provides a quantitative perspective to validate and extend this intuition.

## 5 CONCLUSION

**Summary.** To optimize the caching strategy and fully leverage the potential of caching error correction methods, we propose a novel plug-in acceleration method, CEM. It leverages offline error modeling and a dynamic programming algorithm to derive the optimal caching strategy that minimizes the caching error. By integrating the caching error correction method, we can significantly improve generation fidelity while maintaining acceleration, without introducing any additional computation. Extensive experiments on eight generation models and one quantized model across three tasks demonstrate the plug-in effectiveness of our method on existing accelerated models.

**Limitations.** We provide training-free plug-and-play acceleration for DiT, with compatibility across generation models, acceleration methods, and quantized models. However, our performance still lags behind training-based methods and cannot be applied to one-step generation models.

ACKNOWLEDGMENTS

This work was supported by National Natural Science Foundation of China (grant No. 62350710797), by Guangdong Basic and Applied Basic Research Foundation (grant No. 2023B1515120065, 2025A1515011546), and by the Shenzhen Science and Technology Program (JCYJ20240813105901003, KJZD20240903102901003, ZDCY20250901113000001).

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

## APPENDIX

This is the appendix for our submission titled *Plug-and-Play Fidelity Optimization for Diffusion Transformer Acceleration via Cumulative Error Minimization*. This appendix supplements the main paper with the following content:

## A  STATEMENTS

### A.1  ETHICS STATEMENT

We affirm that the research and writing of this paper strictly adhere to the ICLR Code of Ethics, and do not involve any issues related to human subjects, ethical concerns, or potential conflicts of interest.

### A.2  REPRODUCIBILITY STATEMENT

We affirm that all research presented in this paper is fully reproducible, including the error modeling analysis in Sec. 3.1, the core implementation described in Sec. 3.2 and Appendix **??**, and all results and visualizations shown in figures and tables. The source code will be released publicly later.

### A.3  THE USE OF LARGE LANGUAGE MODELS

We only used GPT4 for minor writing polishing and grammar correction during the writing process. We affirm that no large language model was involved in any other aspect of this work, including idea, coding, experiments and main writing.

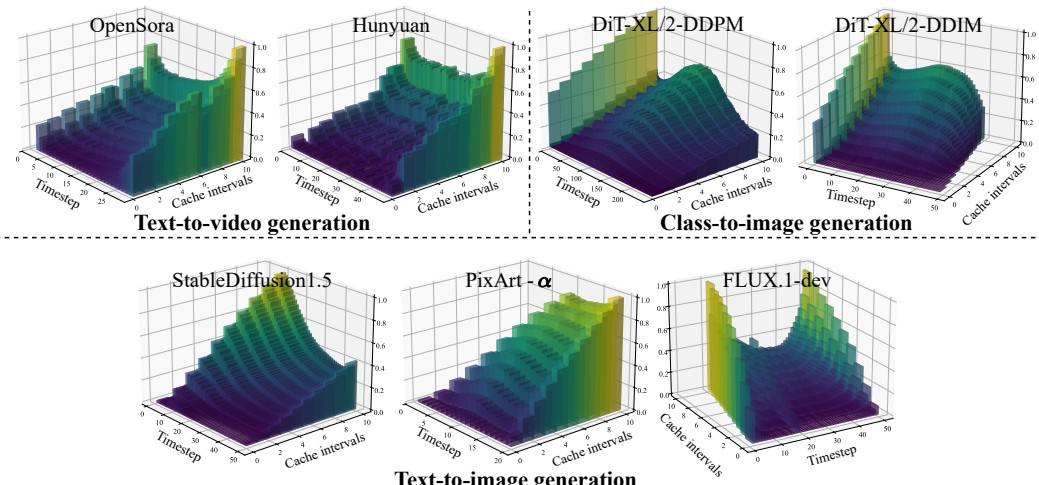

Figure 7: **Offline error modeling.** We model caching error based on the joint variation of cache intervals and denoising timesteps, covering seven generation models across three tasks. In this figure, the caching error is normalized.

Table 7: **Results on DrawBench with Different Data Sources.** We perform offline error modeling on the FLUX.1-dev using prompts from DrawBench, COCO2017 Captions and GPT-Random, and evaluate generation fidelity on the DrawBench benchmark.

| Dataset of OEM/Actual inference | Cosine distance of CLIP | IR↑ | CLIP(G)↑ |
|---|---|---|---|
| Random from GPT / Drawbench | 0.841 | 0.9205 | 32.66 |
| COCO2017 captions / Drawbench | 0.895 | 0.9205 | 32.66 |
| Drawbench / Drawbench | 0.000 | 0.9205 | 32.66 |

## B  MORE DETAILS OF OFFLINE ERROR MODELING

### B.1  VISUALIZATION FOR ERROR DISTRIBUTION

As described in the main paper, we compute the differences between the hidden states stored from previous timesteps in the cache and those computed at the current timestep to model the error distribution across various cache intervals and denoising steps. We present the corresponding error distribution trends for all generation models. Fig. 7 illustrates the complete error distributions of the models evaluated on three tasks. The results clearly show that the error distributions vary notably across models, highlighting that previous fixed or linearly varying caching strategies lack sufficient generalization capability.

Beyond their variation across denoising timesteps, caching errors exhibit non-uniform scaling behaviors under different cache intervals. For instance, on OpenSora, the caching error at an interval of 7 is unexpectedly lower than at smaller intervals. On Hunyuan and SD1.5, the error distributions become progressively more pronounced as the cache interval increases, shifting from an approximately linear pattern

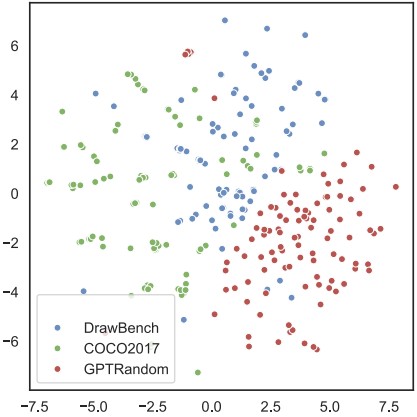

Figure 8: **T-SNE visualization of prompt features from different sources.** The prompt features from DrawBench, COCO2017 captions, and GPT-Random exhibit clear separability, indicating that their distinct data sources result in substantially different feature distributions.

Table 8: **Impact of OEM random sample size on generation fidelity (DiT-XL/2).** We perform offline error modeling on DiT-XL/2 using different numbers of samples and evaluate the generation fidelity after acceleration on the DuCa baseline.

| Sample Num | 10 | 50 | 100 | 200 | 300 | 500 | 1000 |
|---|---|---|---|---|---|---|---|
| Time of OEM | 2.67m | 12.71m | 25.52m | 51.04m | 1.28h | 2.13h | 4.25h |
| FID↓ | 2.84 | 2.83 | 2.80 | 2.80 | 2.80 | 2.80 | 2.80 |
| IS↑ | 235.44 | 235.42 | 235.20 | 235.20 | 235.20 | 235.20 | 235.20 |

Table 9: **Impact of OEM random sample size on generation fidelity (FLUX.1-dev).** We perform offline error modeling on FLUX.1-dev using different numbers of samples and evaluate the generation fidelity after acceleration on the TaylorSeer baseline.

| Sample Num | 10 | 50 | 100 | 200 | 300 | 500 | 1000 |
|---|---|---|---|---|---|---|---|
| Time of OEM | 31.71m | 1.05h | 2.08h | 4.15h | 6.27h | 10.38h | 20.77h |
| IR↑ | 0.9202 | 0.9205 | 0.9205 | 0.9205 | 0.9205 | 0.9205 | 0.9205 |
| CLIP(G)↑ | 32.62 | 32.66 | 32.66 | 32.66 | 32.66 | 32.66 | 32.66 |

to a U-shaped curve. On FLUX.1-dev, an anomalously high caching error occurs during the middle denoising steps when the interval is set to 10. These variations cannot be captured by treating the cache error solely as a function of denoising timesteps, underscoring the importance of incorporating the cache interval as a key variable in error modeling.

This joint modeling approach enables us to simultaneously capture the influences of denoising stages and caching strategies on the resulting cache error. It naturally aligns with the dynamic programming proposed later, allowing our CEM to account for variations across denoising stages and scale differences across cache intervals, thereby deriving an optimal caching strategy that minimizes the overall caching error.

### B.2 IMPACT OF THE SOURCE OF OFFLINE SAMPLES

The offline error modeling captures the model's intrinsic sensitivity to different acceleration operations, as revealed through a limited number of sample generation experiments. This sensitivity is model-intrinsic and content-agnostic, indicating that the prompts used during modeling have negligible impact on the results.

To validate this, we construct modeling prompt sets from three distinct data sources (random prompts from GPT, captions of COCO2017, and prompts of DrawBench) and evaluate the accelerated generation quality on the DrawBench. As shown in Tab. 7, we quantify the distributional differences among the three sources using Cosine distance with CLIP, and include a t-SNE visualization in Fig. 8, which clearly highlights distributional variation. Nevertheless, the offline errors derived from these three sources yield identical quality results on DrawBench, confirming that the error modeling process is indeed content-agnostic.

### B.3 IMPACT OF THE NUMBER OF OFFLINE SAMPLES

Our previous experiments have demonstrated that offline error modeling (OEM) reflects an intrinsic model property independent of sample content. To further examine the modeling difficulty, we evaluate the generation fidelity constructed from different offline sample sets on DiT-XL/2 and FLUX.1-dev.

As shown in Tab. 8, for DiT-XL/2, generation fidelity remains largely unaffected by the number of modeling samples when the sample size exceeds 50, indicating convergence of the modeled error and resulting in consistent cache-interval configurations in the derived strategy. A similar trend is observed on FLUX.1-dev: when more than 10 samples, the generation fidelity remains fully consistent regardless of the number of offline samples. Collectively, the findings from both models indicate that the caching error can be easily captured from a small set of random samples, thereby reflecting the intrinsic sensitivity inherent to the model.

Table 10: **Offline modeling cost of all models.** This modeling is built upon random content generation, which introduces additional overhead beyond generation itself due to the need to store intermediate results and compute feature errors.

| Tasks | Text2Image | | | Text2Video | | | Class2Image |
|---|---|---|---|---|---|---|---|
| Models | FLUX.1-dev | PixArt-$\alpha$ | SD1.5 | Hunyuan | Wan21 | OpenSora | DiT-XL/2 |
| Sample Num | 100 | 100 | 100 | 10 | 10 | 10 | 100 |
| Time w/o. OEM | 1.92h | 13.52m | 38.88m | 4.72h | 2.73h | 3.63h | 19.63m |
| Time w/. OEM | 2.08h(+8.3%) | 14.71m(+8.8%) | 43.83m(+12.7%) | 5.21h(+10.4%) | 4.15h(+52.0%) | 4.65h(+28.1%) | 25.52m(+30.0%) |
| Memory w/o. OEM | 43.42GB | 22.31GB | 3.65GB | 57.36GB | 40.83GB | 52.40GB | 4.09GB |
| Memory w/. OEM | 53.06GB(+22.2%) | 22.65GB(+1.5%) | 3.67GB(+0.5%) | 72.62GB(+26.6%) | 63.46GB(+55.4%) | 52.40GB(+0.0%) | 4.65GB(+13.7%) |

### B.4 OFFLINE MODELING COST

In Tab. 5 of the main paper, we summarize the costs and analyses of the primary models across three tasks. Here, we provide the costs for all models.

We report the time and memory overhead of offline error modeling for each generation model discussed in the paper, with the following clarifications:

- Offline error modeling is performed once per model, and the resulting error can be permanently reused with strong generalization across configurations.
- Random content generation inherently incurs overhead (see "w/o. OEM" in Tab. 10). Since our modeling measures the sensitivity of these generations to acceleration, the additional cost beyond content generation represents the true overhead of the modeling process.
- The offline modeling overhead does not reflect the inference cost. After modeling, invoking CEM and performing cache-strategy optimization incur only negligible additional overhead (see Tab. 12, DCS overhead).

The additional memory overhead primarily results from storing intermediate features during inference to compute differences across cache intervals. On average (including Wan21 we added), offline error modeling introduces only a 15.8% increase in memory usage and a 16.8% increase in modeling time relative to random content generation, both well below 20%. Considering the performance gains of CEM and its zero inference-time overhead, this cost is entirely acceptable.

### B.5 ROBUSTNESS OF MODELED ERROR

We further evaluate the robustness of modeled error under varying conditions, based on Tab. 6 in Sec. 4.3. For each generation model, the caching error is modeled once and reused across all experimental settings to assess the stability of the offline modeling.

Specifically, we evaluate CEM under variations in random seeds, CFG values, resolutions and frames. As shown in Tab. 6, on FLUX.1-dev, CEM consistently improves the generation fidelity of the baseline (TaylorSeer) at equal acceleration efficiency, regardless of changes in seed, CFG, or resolution. Similarly, on Hunyuan, CEM also enhances generation fidelity across different frame numbers.

Overall, the modeled error demonstrates strong robustness, remaining effective without re-modeling when configurations vary. This confirms the practicality and ease of deployment of CEM as a training-free solution for real-world generative applications.

### B.6 SCALABILITY OF ERROR MODELING

The overhead of offline error modeling is influenced by the model scale. The time overhead mainly depends on the model's inherent inference speed, which varies with its stochastic generation process, while our additional cost remains relatively small (averaging 16.8% from Tab. 10). Similarly, the memory overhead is dominated by the model parameters themselves, with our caching and computation contributing only about 15.8% (from Tab. 10) on average.

Overall, larger models generally incur higher absolute overhead. However, two points should be noted:

---

**Algorithm 1** Dynamic Programming Strategy

---

**Input:** Total steps $T$, number of caching $N_c$, cache interval candidate set $\mathcal{N}$, dynamic approximate costs $\mathcal{E}^*(t, n)$

**Output:** Cache interval set $\mathcal{C}$

1: Initialize $dp[t][j] \leftarrow \infty, dp[T][1] \leftarrow 0, prev[t][j] \leftarrow$ None, $\forall t \in [T, 1], \forall j \in [1, N_c]$
2: **for** $j = 1$ to $N_c$ **do**
3:     **for** $t = T$ down to 1 **do**
4:         **if** $dp[t][j] < \infty$ **then**
5:             **for all** $n \in \mathcal{N}$ **do**
6:                 **if** $t > 0$ **then**
7:                     **if** $dp[t][j+1] > dp[t+n][j] + \mathcal{E}^*(t, n)$ **then**
8:                         $dp[t][j+1] \leftarrow dp[t+n][j] + \mathcal{E}^*(t, n)$
9:                         $prev[t][j+1] \leftarrow (t, n)$
10: Backtracking: $\mathcal{C} \leftarrow \{\}, t \leftarrow 1, j \leftarrow N_c$
11: **while** $j > 0$ **do**
12:     $(t_{\text{next}}, n) \leftarrow prev[t][j]$
13:     $\mathcal{C} \leftarrow \mathcal{C} \cup \{t\}$
14:     $t \leftarrow t_{\text{next}}, j \leftarrow j - 1$
15: **return** $\mathcal{C}$

---

- Most of the overhead originates from the model itself rather than from our modeling process.
- The relationship is not strictly monotonic, for example, SD15 is smaller than DiT-XL/2, yet its modeling time is longer.

## C  MORE DETAILS OF DYNAMIC CACHING STRATEGY

### C.1  DYNAMIC PROGRAMMING PSEUDOCODE

We provide the pseudo-code of the dynamic programming algorithm in the appendix to clarify its implementation details. Due to the approximate error introduced by cumulative integration, the cost in our dynamic programming formulation is non-static—it varies across denoising timesteps according to the selected cache interval. The algorithm is built upon the optimal substructure property described in the main paper and ultimately identifies the minimal cumulative error for the entire caching strategy. The optimal cache interval at each stage is obtained via backtracking from the computed total error. Specifically, as shown in the last five lines of Alg. 1, we iteratively trace back the timesteps associated with the previously minimized error and record their corresponding cache intervals, thereby constructing the complete caching strategy.

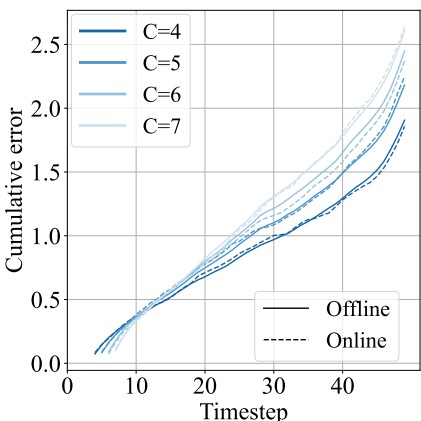

Figure 9: **Cumulative error vs. real error.** The cumulative error approximation captures the trend of real-error variation during denoising under different cache intervals on Hunyuan.

### C.2  CUMULATIVE ERROR VS. ACTUAL ERROR

The cumulative error approximation models the continuous accumulation of caching error during accelerated denoising. To demonstrate its simplicity and effectiveness, we provide supporting evidence from multiple perspectives.

**Additional analysis on cumulative error approximation.** The cumulative error approximation simulates the progressive buildup of caching errors during denoising, grounded in the offline error modeling. Based on this foundation, our design motivations for this module are as follows:

Table 11: **Cumulative Error vs. Real Error.** Using a cache interval of 5 as an example, we report the error comparison on FLUX.1-dev and Hunyuan.

| Timestep | | 0 | 5 | 10 | 15 | 20 | 25 | 30 | 35 | 40 | 45 | 49 |
|---|---|---|---|---|---|---|---|---|---|---|---|---|
| FLUX.1-dev | GT error | 0.000 | 0.004 | 0.279 | 0.434 | 0.603 | 0.682 | 0.714 | 0.840 | 0.893 | 1.025 | 1.233 |
| | Cumulative error | 0.000 | 0.001 | 0.289 | 0.432 | 0.560 | 0.667 | 0.745 | 0.823 | 0.902 | 1.023 | 1.221 |
| Hunyuan | GT error | 0.000 | 0.154 | 0.391 | 0.526 | 0.794 | 0.971 | 1.079 | 1.285 | 1.532 | 1.755 | 2.141 |
| | Cumulative error | 0.000 | 0.082 | 0.363 | 0.552 | 0.748 | 0.934 | 1.101 | 1.272 | 1.490 | 1.747 | 2.182 |

- The offline error modeling omits the influence of previously accumulated errors on the current timestep, as directly modeling cumulative errors would incur exponentially increasing computational costs, rendering the process highly inefficient.
- We aim to realize this approximation through a simple operation, avoiding complex computations that could undermine the intended acceleration efficiency.
- We conduct quantitative experiments comparing the cumulative-integration caching error with the actual error and find that it effectively meets our design objective.

**Quantitative difference between cumulative error and actual error.** Building on the above analysis, we conduct a quantitative evaluation to measure the difference between the cumulative error and the actual error.

As shown in Fig. 3(c) of the main paper, the evolution of these differences on FLUX.1-dev under various acceleration efficiencies has already been illustrated. Here, we further report the key-timestep error differences on FLUX.1-dev and Hunyuan, along with additional line-chart comparisons for Hunyuan.

As presented in Tab. 11, the final difference between cumulative and ground-truth (GT) errors is only 0.80% on FLUX.1-dev and 2.89% on Hunyuan, indicating that the cumulative error closely approximates the actual error. The Fig. 3(c) and Fig. 9 further show highly consistent trajectories between cumulative and actual errors across cache intervals, confirming the accuracy and effectiveness of our approximation.

**Additional theoretical analysis.** In addition to the experimental evaluation of the approximation effect, we further provide a theoretical analysis of how the cumulative error approximates the real error. Together, these theoretical and empirical results comprehensively validate the rationality and effectiveness of our cumulative error approximation module.

**The bound between the cumulative error and the actual error.**

We obtained content-agnostic prior errors from the offline modeling and derived the estimated cumulative error $\mathcal{E}^*$ through integral accumulation. To better illustrate the difference between them, we define the actual error under the same operation during formal inference as $\mathcal{E}$ (Unlike the $\mathcal{E}$ defined in Eq. 1, the symbol here is introduced merely to distinguish notation within the current proof), while the true propagated error of the cached latent during the denoising process is defined as $\hat{\mathcal{E}}$.

We assume the DiT is Lipschitz continuous (common in Diffusion models for bounded error propagation), i.e., $\|\mathcal{D}(x,t) - \mathcal{D}(y,t)\| \le L\|x - y\|$ for some Lipschitz constant $L > 0$.

First, we need to establish the approximate relationship between the offline cumulative error $\mathcal{E}^*$ and the cumulative error $\mathcal{E}$ that occurs during actual inference. The offline $\mathcal{E}^*$ is an empirical estimate of the error distribution using $N_s$ random samples, while $\mathcal{E}$ is computed for a specific inference sample.

**Theorem 1.** *Under the assumption of unified error distribution, the difference $|\mathcal{E}^*(t,n) - \mathcal{E}(t,n)|$ is bounded by:*

$$|\mathcal{E}^*(t,n) - \mathcal{E}(t,n)| \le \sqrt{\frac{\log(2/\delta)}{2N_s}} + \epsilon_{var}, \tag{4}$$

*where $\delta \in (0,1)$ is a confidence parameter, and $\epsilon_{var}$ is a small variance term (empirically small, as per Fig. 3(a)).*

**Proof.** Since diffusion-generated content follows the same underlying distribution, we assume that the caching error in DiT also follows the unified distribution.

Let $\mu(t, n)$ be the true population mean of the cosine error over all possible contents. Then, $\mathcal{E}(t, n)$ for offline is the empirical mean $\hat{\mu}(t, n) = \frac{1}{N_s} \sum_i \mathcal{E}_i(t, n)$, and for online it's a single-sample estimate (or small batch).

By Hoeffding's inequality (since cosine errors are bounded in [0,1]), with probability at least $1 - \delta$:

$$|\hat{\mu}(t, n) - \mu(t, n)| \leq \sqrt{\frac{\log(2/\delta)}{2N_s}}. \tag{5}$$

The online $\mathcal{E}(t, n)$ deviates from $\mu(t, n)$ by at most the empirical variance $\epsilon_{\text{var}}$ (small, as per the analysis: low variance across contents and intervals).

For cumulation: Since CUMSUM is a linear operator, the bound propagates:

$$|\mathcal{E}^*(t, n) - \mathcal{E}(t, n)| \leq \sum_{\tau=1}^{t} |\mathcal{E}^*(\tau, n) - \mathcal{E}(\tau, n)| \leq t \left( \sqrt{\frac{\log(2/\delta)}{2N_s}} + \epsilon_{\text{var}} \right). \tag{6}$$

(For total timesteps $T$, the worst-case bound is $O(T/\sqrt{N_s})$, but empirically tighter due to low $\epsilon_{\text{var}}$.) This holds because extended experiments (Appendix. B.2) show consistency across prompt sources, confirming the i.i.d. assumption. Thus, $\mathcal{E}^* \approx \mathcal{E}$ with high probability (Appendix. C.2 supports this).

**Theorem 2.** *Assuming DiT modules have high structural similarity (perturbations propagate near-linearly, as observed in Fig. 3(c)), the difference $|\mathcal{E}(t, n) - \hat{\mathcal{E}}(t, n)|$ is bounded by:*

$$|\mathcal{E}(t, n) - \hat{\mathcal{E}}(t, n)| \leq L \cdot \sum_{\tau=1}^{t} \mathcal{E}(\tau, n) + \epsilon_{prop}, \tag{7}$$

*where $L$ is the Lipschitz constant of $\mathcal{D}$, and $\epsilon_{prop}$ is a small propagation residual (empirically near-zero, as the approximation matches GT).*

**Proof.** In DiT, each timestep's output is $\mathcal{D}(x_t, t) = f(\mathcal{D}(x_{t+1}, t+1) + \delta)$, where $f$ is the transformer layer, and $\delta$ is noise/caching perturbation. Caching introduces error $\mathcal{E}(t, n)$ at each step, which propagates to future steps. The true propagated error $\hat{\mathcal{E}}(t, n)$ satisfies a recurrence:

$$\hat{\mathcal{E}}(t, n) = \mathcal{E}(t, n) + g(\hat{\mathcal{E}}(t + 1, n)), \tag{8}$$

where $g$ models propagation (approximately linear due to DiT's residual connections and attention linearity). The CUMSUM approximation assumes $g \approx \text{Id}$ (identity, i.e., direct summation), which holds because of "high structural similarity between input and output" (as noted in Liu et al. (2025b)).

By the Lipschitz assumption, propagation error is bounded: $|g(e) - e| \leq L \cdot e + \epsilon_{\text{prop}}$, where $\epsilon_{\text{prop}}$ captures non-linear residuals (small in DiT). Unrolling the recurrence over $t$ steps yields the bound via triangle inequality. For the caching strategy (DP-selected intervals), the bound extends to the full sequence, as DP preserves substructure:

$$|\mathcal{E}(t, n) - \hat{\mathcal{E}}(t, n)| \leq L \cdot \sum_{\tau=1}^{t} \mathcal{E}(\tau, n) + \epsilon_{\text{prop}}. \tag{9}$$

Thus, $\mathcal{E} \approx \hat{\mathcal{E}}$, with the bound tightening for smaller single-step errors.

Finally, by chaining Theorems 1 and 2 (triangle inequality):

$$|\mathcal{E}^*(t, n) - \hat{\mathcal{E}}(t, n)| \leq |\mathcal{E}^*(t, n) - \mathcal{E}(t, n)| + |\mathcal{E}(t, n) - \hat{\mathcal{E}}(t, n)| \tag{10}$$

$$\leq t \left( \sqrt{\frac{\log(2/\delta)}{2N_s}} + \epsilon_{\text{var}} \right) + L \cdot \sum_{\tau=1}^{t} \mathcal{E}(\tau, n) + \epsilon_{\text{prop}}. \tag{11}$$

Table 12: **Online cost of the DCS module across all models.** During actual inference, DCS primarily loads the prior error matrix into memory and solves a dynamic programming problem to obtain the optimal caching strategy, which can be shared across batch generation for improved efficiency.

| Tasks | Text2Image | | | Text2Video | | | Class2Image | |
|---|---|---|---|---|---|---|---|---|
| Models | FLUX.1-dev | PixArt-$\alpha$ | SD1.5 | Hunyuan | Wan21 | OpenSora | DiT-DDPM | DiT-DDIM |
| Time of DPS | 1.10ms | 0.12ms | 0.80ms | 0.71ms | 0.85ms | 0.27ms | 6.96ms | 1.13ms |
| Shape of error | 50*9 | 20*9 | 50*9 | 50*9 | 50*9 | 30*9 | 250*9 | 50*9 |
| Memory of error | 0.88KB | 0.35KB | 0.88KB | 0.88KB | 0.88KB | 0.53KB | 4.39KB | 0.88KB |

This upper bound is $O(T/\sqrt{N_s} + L \cdot \mathcal{E}_{\text{total}})$, where $\mathcal{E}_{\text{total}}$ is total single-step error. Empirically, it's small, confirming the approximation.

**Theoretical analysis of dynamic programming.**

**Time Complexity:** The DP table has $O(T \cdot N_c)$ entries ($t \in [1, T]$, $j \in [1, N_c]$). For each entry $dp[t][j+1]$ (for $j \geq 0$), we evaluate the min over $|\mathcal{N}|$ possible intervals $n$, each requiring $O(1)$ time to compute $\mathcal{E}^*(t, n) + dp[t+n][j]$ (assuming $\mathcal{E}^{(}t, n)$ is precomputed offline in $O(T \cdot |\mathcal{N}|)$ time, as it's content-agnostic). Thus, filling the table takes $O(T \cdot N_c \cdot |\mathcal{N}|)$ time. Backtracking: $O(T)$ time (traverse the path of $N_c$ choices, each step $O(1)$).
Overall Time Complexity: $O(T \cdot N_c \cdot |\mathcal{N}|)$, which is efficient since $T$ is small (e.g., 50), $N_c < T$ (budget-constrained), and $|\mathcal{N}|$ is small (e.g., 10 practical intervals). The analysis notes no additional overhead during inference, as DP runs offline once per model. The actual time consumption of the dynamic programming (DP) process can be found in Appendix. C.3 and Tab. 12.

**Space Complexity:** DP table: $O(T \cdot N_c)$ space (store floats for errors). Precomputed $\mathcal{E}^*(t, n)$: $O(T \cdot |\mathcal{N}|)$ space. Backtracking can use the table itself (no extra space) or a separate predecessor array ($O(T \cdot N_c)$). This complexity is polynomial and scalable, enabling "optimal cache-interval combination that can be shared across multiple generations without additional overhead" (Sec. 3.2 analysis).

**Proof of Optimality:** Suppose there exists an optimal strategy $S^*$ for $dp[t][j+1]$ that chooses interval $n$, but the sub-strategy $S'$ for $dp[t+n][j]$ is not optimal (i.e., there exists a better sub-strategy $S''$ with lower error for $dp[t+n][j]$). Then, replacing $S'$ with $S''$ in $S^*$ would yield a new strategy with total error $\mathcal{E}^*(t, n) + \text{error}(S'') < \mathcal{E}^*(t, n) + \text{error}(S')$, contradicting the optimality of $S^*$. Thus, subproblems must be optimal.
This holds because: **1).** The denoising process is sequential and acyclic (timesteps decrease from $T$ to 1). **2).** Errors are additive $\mathcal{E}^*(t, n)$ is independent of future choices, depending only on the current interval and offline modeling). **3).** The budget $N_c$ is fixed, and choices do not overlap (each caching operation covers distinct timestep segments).

## C.3 ONLINE COST

During inference, the computational overhead of CEM mainly stems from loading the pre-modeled error distribution and solving the dynamic programming optimization.

The memory cost is negligible, as each model only stores an array of size N×T (where N is the number of cache intervals and T the number of timesteps). The time overhead is similarly minimal, the dynamic programming process involves at most T iterations, resulting in computation times on the order of milliseconds. Moreover, the derived optimal caching strategy can be shared across multiple generations with the same acceleration efficiency, further amortizing this minor cost.

# D   MORE EXPERIMENTS

## D.1   MORE IMPLEMENTATION DETAILS

We conduct comprehensive experiments across three major generative tasks covering representative text-to-image, text-to-video, and class-to-image diffusion models, as well as multiple SOTA acceleration techniques.

**Generation models.** Text-to-Image Generation: We evaluate three diffusion-based text-to-image generation models: (1) Stable Diffusion v1.5 (SD1.5) Rombach et al. (2022), a latent diffusion model (LDM) trained on LAION-5B, generating images at a resolution of 512×512. (2) PixArt-$\alpha$ Chen et al. (2023), a transformer-based diffusion model that performs efficient pixel-space modeling at 256×256 resolution. (3) FLUX.1-dev Labs (2024), a state-of-the-art diffusion transformer trained at 1024×1024 resolution, capable of producing high-fidelity, photorealistic images.

Text-to-Video Generation: We include three representative video diffusion models: (1) Hunyuan-Video Kong et al. (2024), generating 65 frames at 480p resolution, emphasizing realistic motion and temporal coherence. (2) Wan2.1-1.3B Wan et al. (2025), a lightweight and efficient video diffusion transformer generating 65 frames at 480p. (3) OpenSora Zheng et al. (2024), an open research model producing 2-second clips at 480p, enabling temporally consistent text-conditioned generation.

Class-to-Image Generation: We adopt DiT-XL/2 Peebles & Xie (2023), a large diffusion transformer trained on ImageNet, evaluated with both DDPM Ho et al. (2020) and DDIM Song et al. (2020) samplers at a 256×256 resolution.

**Acceleration Baselines.** Our method (CEM) is integrated into six representative acceleration or efficiency improvement approaches: FasterSD Li et al. (2023a) accelerates diffusion by reusing features extracted from shallow network layers to reduce redundant computation in deeper ones; ToCa Zou et al. (2024a) combines caching and pruning mechanisms to jointly optimize computational reuse and step reduction; DuCa Zou et al. (2024b) integrates conservative and aggressive caching strategies to maintain a balance between fidelity and acceleration; TaylorSeer Liu et al. (2025c) predicts reusable features from historical cache states through a Taylor-series expansion rather than directly reusing cached results; TeaCache Liu et al. (2025b) adaptively determines the caching policy based on the relationship between input and output activations; and Q-DiT Chen et al. (2025a) is a training-free quantized Diffusion Transformer that serves as our platform to validate the compatibility of CEM with quantized models.

**Evaluation and Metrics.** For text-to-image generation, all models produce images conditioned on captions from the MS-COCO 2017 dataset Lin et al. (2014). For text-to-video generation, evaluations are conducted on VBench Huang et al. (2024b), a comprehensive benchmark of 16 sub-tasks assessing spatial fidelity, motion consistency, and text–video alignment. For class-to-image generation, DiT-XL/2 is evaluated on ImageNet Deng et al. (2009) by generating images for its 1,000 labeled categories.

We employ standard metrics to assess fidelity, perceptual quality, diversity, and semantic alignment. Fréchet Inception Distance (FID) Heusel et al. (2017) measures distributional similarity between generated and real images; lower values indicate better fidelity. CLIPScore (CLIP) Hessel et al. (2021) evaluates semantic alignment between text and images, and higher scores denote stronger consistency. ImageReward (IR) Xu et al. (2023) reflects human aesthetic preference; higher is better. Peak Signal-to-Noise Ratio (PSNR) and Structural Similarity Index Measure (SSIM) evaluate pixel-level and structural fidelity, both maximized for better reconstruction quality. Learned Perceptual Image Patch Similarity (LPIPS) quantifies perceptual difference in deep feature space; lower indicates higher similarity.

For video generation, VBench Huang et al. (2024b) jointly evaluates realism, temporal smoothness, and motion–text consistency, with higher scores representing better video quality. In class-to-image generation, we report Sliced FID (sFID) for image fidelity, Inception Score (IS) Salimans et al. (2016) for diversity, and Precision (P) and Recall (R) to measure the trade-off between fidelity and diversity.

Finally, generation efficiency is reported using both theoretical FLOPs and empirical latency. Image models are tested on RTX 4090, while FLUX.1-dev and all video models are evaluated on A800 GPUs due to higher computational demands.

## D.2 Results Under Different Acceleration Efficiencies

**Text-to-image generation.** Tab. 13 presents a comprehensive quantitative comparison across diverse text-to-image generation settings, demonstrating the effectiveness and generality of our proposed CEM when integrated into various acceleration frameworks.

Table 13: **Quantitative comparison on text-to-image generation under different acceleration efficiencies.** ↓/↑ denotes lower/higher values indicate superior performance. "-" denotes the absence of reference results. "+Ours" indicates the baseline with our CEM. **Bold** font highlights our better results.

| | FLOPS(T)↓ | Spe↑ | Lat(s)↓ | Spe↑ | FID↓ | CLIP(L)(%)↑ | PSNR↑ | SSIM↑ | LPIPS↓ |
|---|---|---|---|---|---|---|---|---|---|
| | SD1.5, MSCOCO2017 10K, DDIM 50 steps, 512×512, RTX4090 | | | | | | | | |
| Origin | 37.05 | 1.00× | 1.44 | 1.00× | 21.75 | 30.92 | INF | 1.00 | 0.00 |
| 50% steps | 18.53 | 2.00× | 0.73 | 1.97× | 25.21 | 32.15 | 20.19 | 0.62 | 0.26 |
| DeepCache | - | - | 0.63 | 2.27× | 21.53 | 30.80 | - | - | - |
| FasterSD | 27.35 | 1.35× | 0.33 | 4.35× | 21.62 | 32.54 | 16.42 | 0.56 | 0.36 |
| +Ours | **27.35** | **1.35×** | **0.33** | **4.35×** | **19.99** | **32.85** | 15.77 | **0.60** | **0.35** |
| | PixArt-$\alpha$, MSCOCO2017 30K, DPM-Solver 20 steps, 256×256, RTX4090 | | | | | | | | |
| | FLOPS(T)↓ | Spe↑ | Lat(s)↓ | Spe↑ | FID↓ | CLIP(L)(%)↑ | PSNR↑ | SSIM↑ | LPIPS↓ |
| Origin | 11.18 | 1.00× | 0.86 | 1.00× | 28.06 | 16.29 | INF | 1.00 | 0.00 |
| 50% steps | 5.59 | 2.00× | 0.43 | 2.00× | 37.41 | 15.82 | 18.67 | 0.70 | 0.20 |
| FORA | 5.66 | 1.98× | 0.52 | 1.64× | 29.67 | 16.40 | - | - | - |
| DeepCache | - | - | 0.62 | 1.39× | 31.57 | 16.24 | - | - | - |
| ToCa | 4.26 | 2.62× | 0.44 | 1.97× | 29.73 | 16.45 | - | - | - |
| DuCa(N=3) | 6.19 | 1.81× | 0.50 | 1.72× | 28.39 | 16.44 | 17.79 | 0.63 | 0.24 |
| +Ours | 6.54 | 1.71× | 0.53 | 1.62× | **27.06** | **16.44** | **21.45** | **0.79** | **0.13** |
| DuCa(N=4) | 5.90 | 1.89× | 0.49 | 1.76× | 35.36 | 16.45 | 15.99 | 0.52 | 0.35 |
| +Ours | 5.94 | 1.88× | **0.49** | **1.76×** | **27.20** | 16.42 | **20.94** | **0.78** | **0.14** |
| DuCa(N=5) | 4.79 | 2.33× | 0.40 | 2.15× | 41.56 | 16.46 | 14.96 | 0.46 | 0.42 |
| +Ours | **4.75** | **2.35×** | **0.39** | **2.20×** | **27.57** | 16.37 | **18.25** | **0.68** | **0.21** |
| | FLUX.1-dev, DrawBench, Rectified Flow 50 steps, 1024×1024, A800 | | | | | | | | |
| | FLOPS(T)↓ | Spe↑ | Lat(s)↓ | Spe↑ | IR↑ | CLIP(G)(%)↑ | PSNR↑ | SSIM↑ | LPIPS↓ |
| Origin | 3719.50 | 1.00× | 35.63 | 1.00× | 0.9649 | 32.57 | INF | 1.00 | 0.00 |
| 50% steps | 1859.75 | 2.00× | 17.82 | 2.00× | 0.9874 | 32.77 | 17.23 | 0.67 | 0.32 |
| 25% steps | 967.07 | 3.85× | 8.91 | 4.00× | 0.9310 | 32.72 | 14.71 | 0.58 | 0.46 |
| Δ-DiT | 1686.76 | 2.21× | 18.27 | 1.95× | 0.8561 | - | - | - | - |
| FORA | 1320.07 | 2.82× | 14.66 | 2.43× | 0.9227 | - | - | - | - |
| ToCa(N=4) | 1263.22 | 2.94× | 14.60 | 2.44× | 0.9822 | 32.36 | 18.27 | 0.67 | 0.30 |
| +Ours | **1263.22** | **2.94×** | **14.13** | **2.52×** | **1.0151** | **32.67** | 17.72 | **0.67** | 0.31 |
| TeaCache(l=0.4) | 1413.41 | 2.63× | 25.45 | 1.40× | 0.7040 | 30.72 | 18.70 | 0.74 | 0.29 |
| +Ours | **1413.41** | **2.63×** | **23.60** | **1.51×** | **0.7545** | **31.34** | 17.24 | 0.69 | 0.34 |
| TeaCache(l=0.6) | 1115.85 | 3.33× | 16.57 | 2.15× | 0.7228 | 30.66 | 17.41 | 0.70 | 0.35 |
| +Ours | **1115.85** | **3.33×** | **16.05** | **2.22×** | **0.7362** | **31.13** | **17.89** | **0.71** | **0.33** |
| TeaCache(l=0.8) | 892.68 | 4.17× | 12.33 | 2.89× | 0.7136 | 30.74 | 16.50 | 0.66 | 0.40 |
| +Ours | **892.68** | **4.17×** | **12.08** | **2.95×** | **0.7139** | **30.77** | **16.96** | **0.67** | **0.39** |
| TaylorSeer(N6O1) | 744.81 | 4.99× | 10.09 | 3.53× | 0.9410 | 32.57 | 15.59 | 0.60 | 0.41 |
| +Ours | **744.81** | **4.99×** | **10.09** | **3.53×** | **0.9811** | **32.89** | **16.11** | **0.61** | **0.39** |
| TaylorSeer(N7O1) | 668.97 | 5.56× | 8.61 | 4.14× | 0.9233 | 32.55 | 14.94 | 0.57 | 0.46 |
| +Ours | **668.97** | **5.56×** | **8.59** | **4.15×** | **0.9449** | **32.59** | **15.71** | **0.58** | **0.42** |
| TaylorSeer(N8O1) | 595.12 | 6.25× | 7.41 | 4.81× | 0.8760 | 32.17 | 14.24 | 0.54 | 0.49 |
| +Ours | **595.12** | **6.25×** | **7.41** | **4.81×** | **0.9205** | **32.66** | **15.41** | **0.56** | **0.46** |

On SD1.5, CEM consistently improves generation quality over all baselines. Under identical acceleration configurations (e.g., same FLOPs and latency as FasterSD), it reduces FID from 21.62 to 19.99 and increases perceptual scores such as CLIP and SSIM, indicating enhanced fidelity without extra computational cost.

On PixArt-$\alpha$, CEM achieves superior speed–quality trade-offs. Across different step-reduction levels in DuCa (N=3–5), our method markedly lowers FID (e.g., from 41.56 to 27.57 at N=5) while maintaining similar acceleration ratios. Perceptual metrics (SSIM/LPIPS) also improve consistently, confirming CEM's robustness across architectures and noise schedules.

On FLUX.1-dev, evaluated with DrawBench, CEM further enhances both image realism (IR) and perceptual alignment (CLIP(G)) across all error correction baselines (ToCa, TeaCache, and TaylorSeer). These gains come with no additional FLOPs or latency, showing that CEM mitigates quality loss even under aggressive step reduction or caching.

Overall, across all diffusion backbones—from SD1.5 and PixArt-$\alpha$ to FLUX.1-dev—CEM consistently boosts generation fidelity under equal or faster inference conditions. These results demon-

Table 14: **Quantitative comparison on class-to-image generation with DiT-XL/2 and quantized model under different acceleration efficiencies.** W/A denotes the quantization bit-width of weights and activations.

| | FLOPS(T)↓ | Spe↑ | Lat(s)↓ | Spe↑ | FID↓ | sFID↓ | IS↑ | P↑ | R↑ | PSNR↑ | SSIM↑ | LPIPS↓ |
|---|---|---|---|---|---|---|---|---|---|---|---|---|
| DiT-XL/2, ImageNet 50K, DDPM 250 steps, 256×256, RTX4090 | | | | | | | | | | | | |
| Origin | 118.68 | 1.00× | 2.51 | 1.00× | 2.23 | 4.57 | 275.64 | 0.83 | 0.58 | INF | 1.00 | 0.00 |
| 50% steps | 59.34 | 2.00× | 1.26 | 1.99× | 2.42 | 5.04 | 270.40 | 0.82 | 0.57 | 8.55 | 0.14 | 0.78 |
| FORA | 39.95 | 2.97× | 1.01 | 2.49× | 2.80 | 6.21 | - | 0.80 | 0.59 | - | - | - |
| ToCa(N=4) | 43.42 | 2.73× | 1.02 | 2.46× | 2.59 | 5.73 | 255.93 | 0.80 | 0.59 | 22.61 | 0.77 | 0.17 |
| +Ours | 43.58 | 2.72× | **0.99** | **2.54×** | 2.66 | **5.33** | **257.85** | **0.81** | **0.59** | **23.78** | **0.80** | **0.13** |
| ToCa(N=5) | 39.25 | 3.02× | 0.92 | 2.73× | 2.82 | 6.03 | 251.84 | 0.80 | 0.59 | 21.81 | 0.74 | 0.19 |
| +Ours | 39.39 | 3.01× | **0.89** | **2.82×** | 2.82 | **5.82** | **252.47** | **0.80** | **0.59** | **23.04** | **0.78** | **0.15** |
| ToCa(N=6) | 36.30 | 3.27× | 0.84 | 2.99× | 3.08 | 6.58 | 246.59 | 0.79 | 0.59 | 20.92 | 0.71 | 0.21 |
| +Ours | 36.48 | 3.25× | **0.82** | **3.06×** | 3.09 | **6.00** | **248.58** | **0.80** | **0.59** | **22.63** | **0.76** | **0.16** |
| DiT-XL/2, ImageNet 50K, DDIM 50 steps, 256×256, RTX4090 | | | | | | | | | | | | |
| Origin | 23.74 | 1.00× | 0.53 | 1.00× | 2.25 | 4.33 | 239.93 | 0.80 | 0.59 | INF | 1.00 | 0.00 |
| 50% steps | 11.87 | 2.00× | 0.27 | 1.96× | 2.87 | 4.58 | 231.05 | 0.79 | 0.58 | 9.05 | 0.16 | 0.80 |
| 33% steps | 8.07 | 2.94× | 0.18 | 2.94× | 4.24 | 5.52 | 214.35 | 0.77 | 0.56 | 9.22 | 0.17 | 0.81 |
| AdaCache | - | - | 0.46 | 1.15× | 4.64 | - | - | - | - | - | - | - |
| TeaCache | - | - | 0.32 | 1.66× | 5.09 | - | - | - | - | - | - | - |
| Δ-DiT | 16.14 | 1.47× | 0.21 | 2.52× | 3.75 | 5.70 | 207.57 | - | - | - | - | - |
| FORA | 8.58 | 2.77× | 0.24 | 2.21× | 3.55 | 6.36 | 229.02 | - | - | - | - | - |
| LazyDiT | 11.93 | 1.99× | 0.28 | 1.89× | 2.70 | 4.47 | 237.03 | 0.80 | 0.59 | - | - | - |
| ToCa(N=4) | 8.73 | 2.72× | 0.25 | 2.12× | 3.64 | 5.14 | 228.44 | 0.79 | 0.55 | 22.20 | 0.75 | 0.18 |
| +Ours | **8.70** | **2.73×** | **0.24** | **2.21×** | **3.19** | **5.12** | **229.59** | **0.79** | **0.57** | **23.57** | **0.79** | **0.14** |
| ToCa(N=5) | 7.44 | 3.19× | 0.20 | 2.65× | 6.37 | 7.09 | 199.48 | 0.74 | 0.53 | 16.56 | 0.53 | 0.40 |
| +Ours | **7.14** | **3.32×** | **0.18** | **2.94×** | **4.68** | **6.41** | **212.13** | **0.77** | **0.55** | **21.59** | **0.72** | **0.20** |
| ToCa(N=6) | 7.02 | 3.38× | 0.18 | 2.94× | 6.79 | 7.41 | 187.32 | 0.72 | 0.55 | 17.62 | 0.56 | 0.36 |
| +Ours | **6.72** | **3.53×** | **0.17** | **3.12×** | **5.38** | **6.84** | **205.52** | **0.76** | **0.55** | **20.83** | **0.69** | **0.23** |
| DuCa(N=3) | 9.58 | 2.48× | 0.25 | 2.12× | 3.05 | 4.66 | 233.11 | 0.80 | 0.57 | 24.62 | 0.82 | 0.12 |
| +Ours | **9.49** | **2.50×** | 0.25 | 2.12× | **2.80** | **4.64** | **235.20** | **0.80** | **0.58** | **25.47** | **0.84** | **0.10** |
| DuCa(N=4) | 7.66 | 3.10× | 0.20 | 2.65× | 3.39 | 4.91 | 226.33 | 0.79 | 0.56 | 22.40 | 0.75 | 0.18 |
| +Ours | **7.40** | **3.21×** | **0.19** | **2.79×** | **3.36** | 5.15 | **226.59** | **0.79** | **0.57** | **23.69** | **0.79** | **0.14** |
| DuCa(N=5) | 6.32 | 3.76× | 0.17 | 3.12× | 6.07 | 6.64 | 199.64 | 0.74 | 0.52 | 16.63 | 0.53 | 0.39 |
| +Ours | 6.73 | 3.53× | **0.17** | **3.12×** | **3.96** | **5.87** | **218.66** | **0.78** | **0.55** | **23.00** | **0.76** | **0.16** |
| DuCa(N=6) | 5.86 | 4.05× | 0.15 | 3.53× | 6.38 | 6.65 | 189.97 | 0.73 | 0.54 | 17.47 | 0.54 | 0.37 |
| +Ours | **5.69** | **4.17×** | **0.14** | **3.79×** | **5.06** | 6.75 | **206.03** | **0.77** | **0.54** | **21.62** | **0.70** | **0.21** |
| TaylorSeer(N3O3) | 8.55 | 2.78× | 0.31 | 1.71× | 2.34 | 4.69 | 238.42 | 0.80 | 0.59 | 35.13 | 0.96 | 0.02 |
| +Ours | **8.55** | **2.78×** | **0.30** | **1.77×** | **2.31** | **4.55** | **242.08** | **0.81** | **0.59** | **36.16** | **0.97** | **0.01** |
| TaylorSeer(N4O4) | 6.66 | 3.56× | 0.27 | 1.96× | 2.49 | 5.19 | 235.83 | 0.80 | 0.59 | 30.74 | 0.93 | 0.04 |
| +Ours | **6.66** | **3.56×** | **0.27** | **1.96×** | **2.46** | **4.80** | **238.28** | **0.80** | **0.59** | **31.76** | **0.94** | **0.03** |
| TaylorSeer(N5O3) | 5.34 | 4.45× | 0.22 | 2.41× | 2.65 | 5.36 | 231.59 | 0.80 | 0.59 | 28.48 | 0.90 | 0.07 |
| +Ours | **5.34** | **4.45×** | **0.22** | **2.41×** | **2.64** | 5.48 | **233.75** | **0.80** | **0.59** | **28.74** | **0.90** | **0.07** |
| TaylorSeer(N6O1) | 4.76 | 4.99× | 0.14 | 3.79× | 3.56 | 7.52 | 223.83 | 0.79 | 0.56 | 24.69 | 0.80 | 0.13 |
| +Ours | **4.76** | **4.99×** | **0.13** | **4.08×** | **3.08** | **6.43** | **231.10** | **0.80** | **0.57** | **25.64** | **0.83** | **0.10** |

| | Size(MB)↓ | Com↑ | Lat(s)↓ | Spe↑ | FID↓ | sFID↓ | IS↑ | P↑ | R↑ | PSNR↑ | SSIM↑ | LPIPS↓ |
|---|---|---|---|---|---|---|---|---|---|---|---|---|
| DiT-XL/2, ImageNet 10K, DDIM 50 steps, 256×256, quantized, RTX4090 | | | | | | | | | | | | |
| Origin | 1349 | 1.00× | 0.62 | 1.00× | 5.31 | 17.61 | 245.85 | 0.81 | 0.68 | INF | 1.00 | 0.00 |
| Q-DiT(W6A8) | 518 | 2.60× | 0.45 | 1.38× | 5.44 | 17.61 | 237.34 | 0.80 | 0.68 | 31.10 | 0.95 | 0.04 |
| +Ours | **518** | **2.60×** | **0.22** | **2.82×** | 5.51 | **17.49** | **240.36** | **0.80** | **0.68** | 31.06 | 0.93 | 0.05 |
| Q-DiT(W4A8) | 347 | 3.89× | 0.39 | 1.59× | 6.31 | 17.81 | 209.30 | 0.76 | 0.69 | 24.88 | 0.82 | 0.14 |
| +Ours | **347** | **3.89×** | **0.20** | **3.10×** | **6.20** | **17.62** | **213.50** | **0.76** | **0.69** | **24.99** | **0.82** | **0.14** |

strate that our offline error modeling and dynamic programming framework offer a simple yet robust enhancement that generalizes effectively across diverse text-to-image diffusion models.

**Class-to-image generation.** Tab. 14 summarizes the quantitative results for class-to-image generation using DiT-XL/2 under various acceleration and quantization settings. The results show that integrating our proposed CEM consistently enhances both visual quality and efficiency across different diffusion configurations, sampling schedules, and precision levels.

Under the DDPM sampler, when combined with step-reduction and caching methods such as ToCa, CEM improves quality while maintaining comparable computational cost. For example, at ToCa (N=4), sFID decreases from 5.73 to 5.33 and PSNR increases from 22.61 to 23.78, with no notice-

Table 16: **Comparison with learning-based acceleration methods on DiT-XL/2.**

| Method | Train | FLOPS(T)↓ | Spe↑ | Lat(s)↓ | Spe↑ | FID↓ | sFID↓ | IS↑ | P↑ | R↑ |
|---|---|---|---|---|---|---|---|---|---|---|
| Origin | – | 23.74 | 1.00× | 0.53 | 1.00× | 2.25 | 4.33 | 239.93 | 0.80 | 0.59 |
| L2C Ma et al. (2024a) | yes | – | – | – | 1.25× | 2.62 | 4.50 | 233.26 | 0.79 | 0.59 |
| HarmoniCa Huang et al. (2024a) | yes | – | – | – | 1.30× | 2.36 | 4.24 | 238.74 | 0.81 | 0.60 |
| **TaylorSeer+Ours** | **no** | **11.87** | **2.00×** | **0.37** | **1.43×** | **2.27** | 4.42 | **242.45** | **0.81** | 0.59 |

able increase in FLOPs or latency. The stable Precision/Recall values further indicate that CEM preserves generation diversity while improving fidelity.

Under the DDIM sampler, across multiple acceleration baselines (ToCa, DuCa, TaylorSeer), CEM delivers consistent performance gains. Even in high-speed scenarios (over 3× acceleration), it effectively mitigates quality degradation, reducing FID from 3.64 to 3.19 (ToCa, N=4) and from 6.79 to 5.38 (ToCa, N=6). Perceptual quality is also enhanced, as evidenced by higher SSIM and lower LPIPS scores, confirming that CEM accurately compensates for offline-modeled errors under aggressive acceleration.

In low-bit quantization settings, CEM maintains comparable fidelity while further improving efficiency. For Q-DiT (W6A8), latency is reduced by more than 2× with nearly unchanged FID (5.44→5.51), while for Q-DiT (W4A8), FID remains stable (6.31→6.20) as runtime improves from 1.59× to 3.10×. These results demonstrate that CEM and quantization are compatible, our method effectively leverages hardware-level compression without introducing additional degradation.

Overall, CEM provides a lightweight and general enhancement for diffusion transformers, robustly stabilizing the generation process across different acceleration ratios, sampling schedules, and quantization precisions, thereby achieving a balanced improvement in both fidelity and efficiency.

## D.3 HIGHER RESOLUTIONS AND LONGER FRAMES

Experiments at 480p are conducted to ensure fair comparisons with the baselines. We have additionally included comparative experiments at 720p resolution and with longer frame sequences.

According to the Tab. 15, our CEM enhances the VBench of the TeaCache baseline across both 480p and 720p resolutions, as well as for longer 129 frames. These results further validate the effectiveness and robustness of our approach under more challenging generation settings.

Table 15: **Improvement of our CEM on Hunyuan under higher resolutions or longer frame settings.**

| Resolution/Frames | Method | VBench(%)↑ |
|---|---|---|
| 480P-65f | TeaCache | 77.56 |
| | **+Ours** | **78.15** |
| 480P-129f | TeaCache | 76.21 |
| | **+Ours** | **77.31** |
| 720P-65f | TeaCache | 78.13 |
| | **+Ours** | **78.42** |
| 720P-129f | TeaCache | 77.22 |
| | **+Ours** | **78.43** |

## D.4 COMPARISON WITH LEARNING-BASED METHODS

The two methods mentioned, L2C and HarmoniCa, are learning-based approaches with relatively low acceleration ratios (below 2×). A direct comparison with our CEM would therefore be somewhat inequitable, as our method is completely training-free. Nevertheless, to better illustrate the superiority of our CEM, we adjust the acceleration efficiency to align with that of L2C and HarmoniCa and conduct a comparative evaluation in terms of generation quality.

It can be observed that in Tab. 16 our CEM achieves higher acceleration efficiency and better fidelity even under a training-free setting.

## D.5 COMPARISON WITH ONE-STEP DIFFUSION

Both few-step diffusion and caching acceleration aim to speed up existing DiT by reducing the number of denoising iterations. However, their motivations and focuses are fundamentally different in Tab. 17:

Table 17: **One-step diffusion vs. our CEM.** Since direct comparison across models is not feasible, we report the results in percentage form.

| Model | Method | Fidelity↑(Method/Origin)(%) | Speed↑ | Training Costs↓ |
|-------|--------|------------------------------|--------|-----------------|
| ImageNet | Origin | 100.0 | 1.00× | – |
| | ShortCut(ICLR25) Frans et al. (2024) | 42.9 | ∼100 | TPUv3 (1–2 days) |
| | **Ours** | **99.3** | **3.56** | **Free** |
| PixArt-α | Origin | 100.0 | 1.00× | – |
| | SIM(NIPS24) Luo et al. (2024) | 81.2 | ∼30 | 4 A100s (2 days) |
| | **Ours** | **100.1** | **2.35** | **Free** |
| SD1.5 | Origin | 100.0 | 1.00× | – |
| | EDM(CVPR24) Yin et al. (2024) | 76.4 | 28.7 | 72 A100s (36 hours) |
| | **Ours** | **108.8** | **4.35** | **Free** |

Speed-quality trade-off. Few-step diffusion models achieve few-step generation through retraining, leading to substantial acceleration but often at the expense of visual quality. In contrast, our CEM provides training-free acceleration while preserving the original model's generation fidelity.

Cost of Acceleration. The computational costs of the two approaches differ significantly. Few-step diffusion entails substantial retraining overhead (in above table). In contrast, we offer training-free acceleration. Our CEM is plug-and-play, achieving a superior balance between acceleration and generation quality without incurring additional computational cost.

Generalization. Few-step diffusion relies on complex, experience-driven modules and costly training, limiting its ability to generalize across model. In contrast, our CEM is plug-and-play and can be directly applied to various visual generative DiT models, demonstrating strong generalization.

Hence, caching-based acceleration and few-step diffusion have remained independent research directions. Although this is beyond the scope of our current work, we propose a potential direction: our offline error modeling can also be applied to capture pruning-induced errors, enabling adaptive pruning for one-step diffusion to further improve efficiency.

# E    MORE VISUALIZATION

## E.1    STABLEDIFFUSION1.5

We provide additional visual results and analyses for the text-to-image task, focusing on SD1.5 (Fig.10) and PixArt-α (Fig.11). In SD1.5, integrating CEM into FasterSD effectively mitigates common generation artifacts such as feature distortion (e.g., the bike in column 3, the bear's head in column 10, and the commuter train in column 11) and incomplete synthesis (e.g., the bird's head in column 6). The optimized FasterSD also shows improved structural consistency and fidelity with the original model (e.g., the cat's posture in column 1, the trailer in column 4, and the police motorcycle in column 9). Notably, in some cases it even surpasses the original model (e.g., the restored teddy bear eyes in column 5 and the refined airplane in column 8), confirming the effectiveness of our proposed method.

## E.2    PIXART-α

In PixArt-α, our method significantly enhances the visual fidelity of the DuCa model, particularly in recovering fine-grained details. With CEM applied, DuCa produces outputs more consistent with those of the original model—for example, the seagull in column 1, the donuts in column 8, and the soda can in column 11. Furthermore, our method effectively reduces generation failures, including visual artifacts (e.g., the distorted handle in column 4 and the inaccurate mirror in column 9) and blurry regions (e.g., the tree in column 2).

These visualizations further support the quantitative findings in Tab. 1, providing clear evidence that our method, when used as a plug-in, enhances caching strategies and improves the fidelity of existing acceleration approaches.

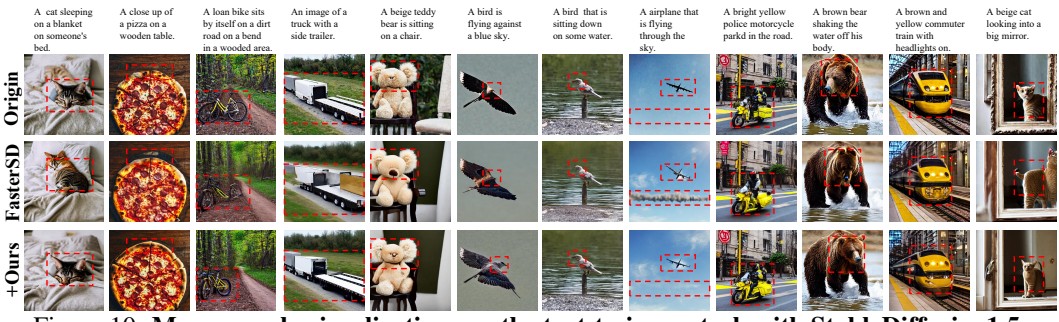

Figure 10: **More sample visualizations on the text-to-image task with StableDiffusion1.5.**

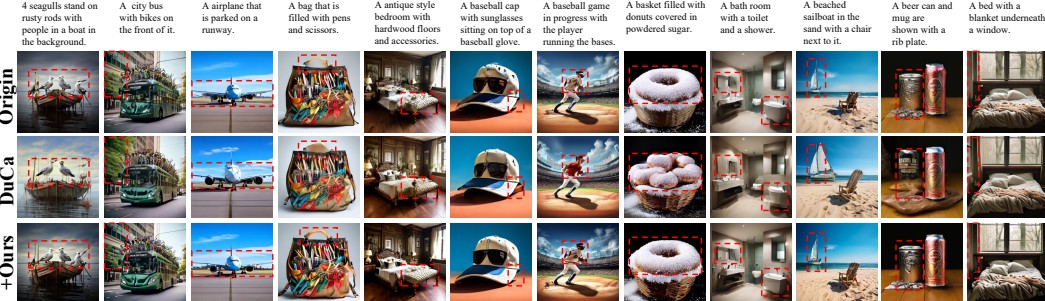

Figure 11: **More sample visualizations on the text-to-image task with PixArt-$\alpha$.**

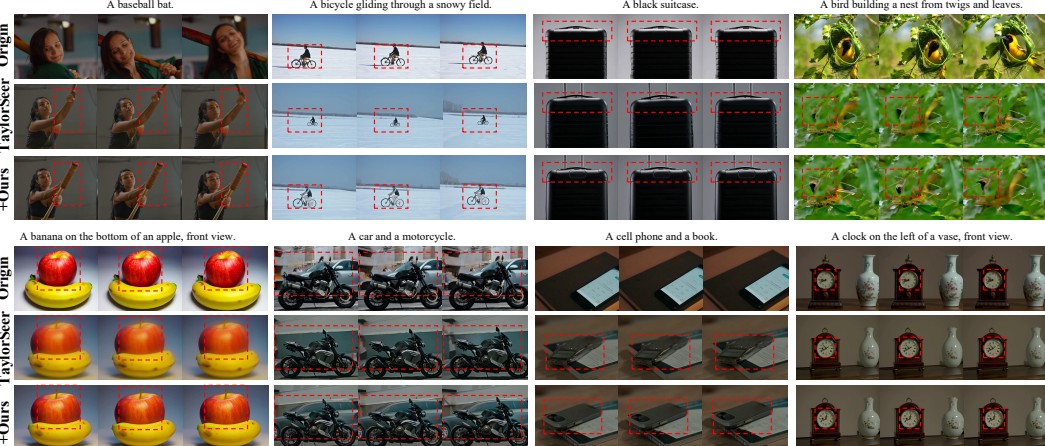

Figure 12: **More sample visualizations on the text-to-video task with Hunyuan.**

### E.3 HUNYUAN

For the text-to-video task, we primarily present visual comparisons on the advanced Hunyuan model. Although the original TaylorSeer achieves up to 5× acceleration, the visualizations in Fig. 12 reveal noticeable quality degradation, including blurriness (e.g., the bird in row 1, column 4, and the apple in row 2, column 1), undesired viewpoint shifts (e.g., the person in row 1, column 2), and temporal inconsistencies (e.g., the car in row 2, column 2).

After integrating our method, these artifacts are effectively alleviated. For instance, in row 1, column 2, the viewpoint discrepancy is significantly reduced, yielding a result much closer to the original. In row 2, column 1, the apple appears markedly clearer, and in row 1, column 3, our method better preserves suitcase details, achieving higher consistency with the original output.

### E.4 DiT-XL/2

Finally, we provide additional visualizations in Fig. 14 based on both the DDPM and DDIM sampling strategies using the DiT-XL/2 model. It is worth noting that we also conduct extensive experiments on the quantized model Q-DiT. In this case, our method directly apply the caching strategy to

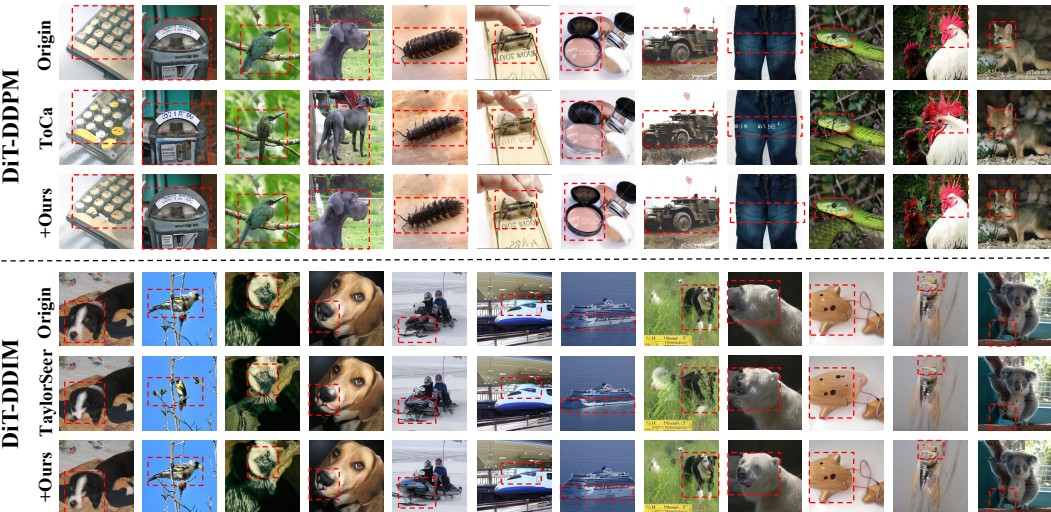

Figure 13: **More sample visualizations on the class-to-image task with DiT-XL/2.**

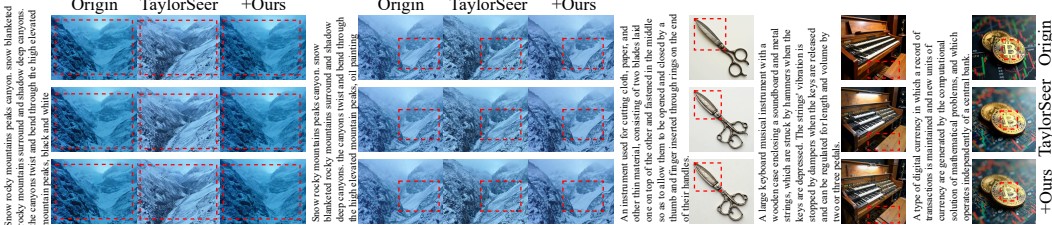

Figure 14: **More sample visualizations with complex prompts on FLUX.1-dev and Hunyuan.**

quantized model, resulting in an additional 2× acceleration while maintaining comparable or even better generation quality. The primary contribution of our method on Q-DiT lies in further improving acceleration efficiency and demonstrating compatibility with quantization techniques. However, since the fidelity improvements are not particularly significant in this setting, we do not include additional visualizations of Q-DiT here.

DiT-XL/2 is trained using the 1,000 class IDs from ImageNet, which provides more structured and less ambiguous prompts than those in text-to-image models. This simple prompt constraint reduces the complexity of the generation task and enables a more direct and interpretable analysis of the role of self-attention during the synthesis process.

Under the DDPM sampling, the generation results of the ToCa model combined with our method become more aligned with those of the original model when combined with our method. For instance, the keyboard in column 1, the bird in column 3, and the dog in column 4 are all more faithfully reproduced. Additionally, we observe that our method unexpectedly suppresses the generation of undesired artifacts to some extent, for example, the watermark on the pants in column 9 is noticeably reduced. Under the DDIM sampling strategy, our method also significantly improves generation fidelity. It demonstrates stronger capability in preserving fine-grained details, rather than suffering from distortions caused by the omission of timesteps or tokens during acceleration. For example, the dog's face in column 1, the dog's mouth in column 4, and the polar bear in column 9 all illustrate the effectiveness of our method in restoring detailed features.

In summary, the visual results across various tasks and models, together with the quantitative results presented in the main paper, collectively demonstrate the effectiveness of our method.

### E.5 VISUALIZATION WITH EXTREMELY COMPLEX PROMPTS

We further evaluate the CEM using complex prompts to examine its performance in challenging scenarios. As illustrated, CEM enhances the overall visual style and color consistency after acceleration (e.g., Example 1), producing results closer to the original video. It also better preserves fine details, such as the textures on the snow mountain (e.g., Example 2). These results demonstrate that CEM effectively improves the generation fidelity of accelerated models, even in complex scenes.

