# OpenReview forum: "Plug-and-Play Fidelity Optimization for Diffusion Transformer Acceleration via Cumulative Error Minimization"
_ICLR.cc/2026/Conference — ICLR 2026 Poster_

### Official Review · Reviewer_m1Xm · 2025-10-29

**Soundness:** 2
**Presentation:** 2
**Contribution:** 2
**Rating:** 4
**Confidence:** 4

**Summary:**

Diffusion Transformers suffer slow inference due to iterative denoising; training‑free cache acceleration helps but introduces sizable errors, and fixed caching cannot adapt to error variation across timesteps. This paper proposes CEM, a plug‑in, model‑agnostic strategy that models error jointly over denoising timesteps and cache intervals and uses dynamic programming with a cumulative‑error approximation to optimize the caching schedule, integrating seamlessly with existing cache‑correction pipelines and quantized models with negligible overhead. Across seven models and three tasks, CEM consistently improves the fidelity of accelerated generators and can even surpass the original unaccelerated baselines.

**Strengths:**

1. The proposed method is easy to implement.
2. It is a plug-and-play framework that enhances the performance of previous methods without additional overhead.
3. A training-free approach without relying heavily on computational resources.

**Weaknesses:**

1. The performance on SOTA video generation methods, e.g., Hunyuan and Wan2.1 on high-resolution generation, e.g., 720p and beyond, is missing. The acceleration of more powerful video generation models towards higher resolution should be more challenging and practical.
2. The author should include the experiments on few-step diffusion models.
3. As the author mentioned in Line 102, some works employ error compensation approaches. Although they incur some overhead, I believe it is better to compare this work with them to further show the superiority.
4. The motivation and method of this work are a little bit trivial.
5. The prompts for the visualization in this paper are too simple and short. I hope the authors could include more visualization with complex prompts. For example, video prompts with more complex motion descriptions.

**Questions:**

N/A

---

> ### Author Response · Authors · 2025-11-24
> **Official Comment by Authors (1/3)**
>
> We sincerely thank the reviewer for the valuable suggestions that have helped us further improve this work.
>
> >**W1**. The performance on SOTA video generation methods, e.g., Hunyuan and Wan2.1 on high-resolution generation, e.g., 720p and beyond, is missing. The acceleration of more powerful video generation models towards higher resolution should be more challenging and practical.
>
> **A1**.
> We thank the reviewer for the constructive feedback. Experiments at 480p are conducted to ensure fair comparisons with the baselines. We have additionally included comparative experiments at 720p resolution and with longer frame sequences.
>
> **Table.A**: Improvement of our CEM on Hunyuan under higher resolutions or longer frame settings.
>
> | Resolution/Frames | Method   | VBench |
> |:------------------:|:---------:|:-------:|
> | 480P-65f | TeaCache | 77.56 |
> |            | **+Ours**   | **78.15** |
> | 480P-129f | TeaCache | 76.21 |
> |            | **+Ours**   | **77.31** |
> | 720P-65f  | TeaCache | 78.13 |
> |            | **+Ours**   | **78.42** |
> | 720P-129f | TeaCache | 77.22 |
> |            | **+Ours**   | **78.43** |
>
> According to the table above, our CEM enhances the VBench of the TeaCache baseline across both 480p and 720p resolutions, as well as for longer 129 frames. These results further validate the effectiveness and robustness of our approach under more challenging generation settings.
>
> **Table.B**: Quantitative comparison on Wan2.1.
>
> | Method | FLOPs Spe.↑ | Latency Spe.↑ | VBench(%)↑ |
> |:--------------------------:|:-------------:|:----------------:|:--------------:|
> | Origin | 1.00× | 1.00× | 78.97 |
> | 50% steps | 2.00× | 2.01× | 68.44 |
> | 25% steps | 3.85× | 3.83× | 61.64 |
> | TaylorSeer (N6O1) | 5.56× | 4.74× | 75.31 |
> | **+Ours** | **5.56×** | **4.76×** | **76.18** |
>
> In addition, we include the acceleration results on the **Wan 2.1** model (in Tab. 2 of the paper). As shown in the table above, our CEM improves the VBench score of the TaylorSeer model (with 5.56× acceleration) by 0.87, narrowing the gap with the original model by **23.8%**. This further demonstrates the effectiveness of our approach.
>
> >**W2**. The author should include the experiments on few-step diffusion models.
>
> **A2**. We analyzed the distinct objectives and comparative results of few‑step diffusion (one-step diffusion) and cache‑based acceleration methods.
>
> **Table.C**: Few-step diffusion vs. our CEM. Since direct comparison across models is not feasible, we report the results in percentage form.
>
> |    ImageNet | Fidelity↑ (Method/Origin)(%) | Speed↑ | Training Costs↓ |
> |:-------:|:---------:|:------:|:------:|
> | Origin | 100.0 | 1.00× | – |
> | ShortCut(ICLR25) [1] | 42.9 | ~100 | TPUv3(1–2 days) |
> | **Ours** | **99.3** | 3.56 | **Free** |
>
> | PixArt-$\alpha$  | Fidelity↑ (Method/Origin)(%) | Speed↑ | Training Costs↓ |
> |:-------:|:---------:|:------:|:------:|
> | Origin | 100.0 | 1.00× | – |
> | SIM(NIPS24) [2] | 81.2 | ~30 | 4 A100s(2 days) |
> | **Ours** | **100.1** | 2.35 | **Free** |
>
> | SD15 | Fidelity↑ (Method/Origin)(%) | Speed↑ | Training Costs↓ |
> |:-------:|:---------:|:------:|:------:|
> | Origin | 100.0 | 1.00× | – |
> | EDM(CVPR24) [3] | 76.4 | 28.7 | 72 A100s(36 hours) |
> | **Ours** | **108.8** | 4.35 | **Free** |
>
> Both few‑step diffusion and caching acceleration aim to speed up existing DiT by reducing the number of denoising iterations. However, their motivations and focuses are fundamentally different:
> * **Speed-quality trade‑off**.
> Few‑step diffusion models achieve few‑step generation through retraining, leading to substantial acceleration but often at the expense of visual quality. In contrast, our CEM provides training‑free acceleration while preserving the original model's generation fidelity.
> * **Cost of Acceleration**.
> The computational costs of the two approaches differ significantly. Few‑step diffusion entails substantial retraining overhead (in above table). In contrast, we offer training‑free acceleration. Our CEM is plug‑and‑play, achieving a superior balance between acceleration and generation quality without incurring additional computational cost.
> * **Generalization**.
> Few‑step diffusion relies on complex, experience‑driven modules and costly training, limiting its ability to generalize across model. In contrast, our CEM is plug‑and‑play and can be directly applied to various visual generative DiT models, demonstrating strong generalization.
>
> Hence, caching‑based acceleration and few‑step diffusion have remained **independent** research directions.
>
> Although this is beyond the scope of our current work, we propose a **potential direction**: our offline error modeling can also be applied to capture pruning‑induced errors, enabling adaptive pruning for one‑step diffusion to further improve efficiency.
>
> [1]. ONE STEP DIFFUSION VIA SHORTCUT MODELS. ICLR 2025 oral.
> [2]. One-Step Diffusion Distillation through Score Implicit Matching. NIPS 2024.
> [3]. One-step Diffusion with Distribution Matching Distillation. CVPR 2024.

---

> ### Author Response · Authors · 2025-11-24
> **Official Comment by Authors (2/3)**
>
> >**W3**. As the author mentioned in Line 102, some works employ error compensation approaches. Although they incur some overhead, I believe it is better to compare this work with them to further show the superiority.
>
> **A3**. We have currently applied our method to improve several error‑correction approaches in caching‑based acceleration, including ToCa, DuCa and TaylorSeer. We thank the reviewer for the insightful suggestion. Our CEM can be integrated as a plugin into online caching methods. Due to time constraints, we selected **TeaCache** as a representative framework to demonstrate performance improvements, as AdaptiveDiffusion targets U‑Net models and AdaCache provides code only for OpenSora.
>
> **Table.D**: The improvement of our CEM on TeaCache for FLUX.1-dev.
>
> |FLUX.1-dev 1024*1024| FLOPS(T)↓ | Spe↑ | Lat(s)↓ | Spe↑ | IR↑ | CLIP(G)(%)↑ | PSNR↑ | SSIM↑ | LPIPS↓ |
> |:--------:|:----------:|:----:|:---------:|:----:|:----:|:------------:|:------:|:------:|:------:|
> | Origin | 3719.50 | 1.00× | 35.63 | 1.00× | 0.9649 | 32.57 | INF | 1.00 | 0.00 |
> | 25% steps | 967.07 | 3.85× | 8.91 | 4.00× | 0.9310 | 32.72 | 14.71 | 0.58 | 0.46 |
> | TeaCache (l=0.6) | 1115.85 | 3.33× | 16.57 | 2.15× | 0.7228 | 30.66 | 17.41 | 0.70 | 0.35 |
> | **+Ours** | **1115.85** | **3.33×** | **16.05** | **2.22×** | **0.7362** | **31.13** | **17.89** | **0.71** | **0.33** |
> | TeaCache (l=0.8) | 892.68 | 4.17× | 12.33 | 2.89× | 0.7136 | 30.74 | 16.50 | 0.66 | 0.40 |
> | **+Ours** | **892.68** | **4.17×** | **12.08** | **2.95×** | **0.7139** | **30.77** | **16.96** | **0.67** | **0.39** |
>
> **Table.E**: The improvement of our CEM on TeaCache for Hunyuan.
>
> |Hunyuan 480P| FLOPS(T)↓ | Spe↑ | Lat(s)↓ | Spe↑ | VBench(%)↑ | PSNR↑ | SSIM↑ | LPIPS↓ |
> |:--------:|:----------:|:----:|:---------:|:----:|:-----------:|:------:|:------:|:------:|
> | Origin | 29773.00 | 1.00× | 441.76 | 1.00× | 78.46 | INF | 1.00 | 0.00 |
> | 25% steps | 7741.11 | 3.85× | 111.13 | 3.98× | 70.89 | 20.66 | 0.70 | 0.53 |
> | TeaCache (l=0.4) | 6550.06 | 4.55× | 108.54 | 4.07× | 77.56 | 19.58 | 0.68 | 0.37 |
> | **+Ours** | **6550.06** | **4.55×** | **105.94** | **4.17×** | **78.15** | **24.54** | **0.80** | **0.23** |
>
> As shown in the tables above, we evaluate the integration of our CEM with TeaCache on two major generative models: FLUX.1‑dev (text‑to‑image) and Hunyuan (text‑to‑video). More experiments are provided in the Appendix.D.2, Tab.13.
>
> For FLUX.1‑dev, CEM reduces the caching errors of TeaCache under the same acceleration level, thereby enhancing generation quality. Since CEM determines the optimal caching strategy via offline error modeling before batch generation, it also lowers inference latency. For instance, at l=0.4, CEM improves TeaCache’s IR by **7.2%** (0.505) and CLIP score by **2%** (0.62), while reducing inference latency by **0.25s**, achieving a zero‑cost improvement in acceleration efficiency.
>
> For Hunyuan, at l=0.4, CEM raises TeaCache's overall VBench score from 77.56 to 78.15, closing **65%** of the gap to the original model (78.46). Meanwhile, inference latency decreases by **2.6s**, demonstrating simultaneous gains in both generation quality and acceleration efficiency.
>
> >**W4**. The motivation and method of this work are a little bit trivial.
>
> **A4**.
> Thank you very much for your suggestion. We have reorganized the logical structure of the Introduction and Method sections regarding the motivation and methodological details. The main line of reasoning is as follows:
> * We have revised the abstract and introduction in Sec. 1 to emphasize that existing acceleration methods fundamentally **lack** awareness of model sensitivity, leading to **inevitable** errors. This naturally motivates our proposed CEM, in which offline error modeling and cumulative error minimization by dynamic programming are formulated.
> * We have refined the methodology section in Sec. 3 to better **highlight** the role of cumulative error and added analyses on error distribution consistency and related aspects.
> * We have introduced three additional quality metrics (**PSNR, SSIM, LPIPS**) and included improvement results on the **Wan2.1** video generation model in Sec. 4.2 and Appendix.D.
> * We have added extensive analytical experiments, including evaluations of cost, robustness, and theoretical justification (Sec. 4.3, Appendix.B, Appendix.C), to more comprehensively demonstrate the effectiveness of our method.

---

> ### Author Response · Authors · 2025-11-24
> **Official Comment by Authors (3/3)**
>
> >**W5**. The prompts for the visualization in this paper are too simple and short. I hope the authors could include more visualization with complex prompts. For example, video prompts with more complex motion descriptions.
>
> **A5**. We sincerely thank the reviewer for their valuable comments. All visualization examples in our paper are drawn from the same benchmarks used for quantitative evaluation. These examples not only represent the quantitative results across the full benchmark but also offer qualitative comparisons on identical generated content, thereby providing a more comprehensive demonstration of the effectiveness of our CEM.
>
> We have added generation examples under custom complex prompt scenarios in **Appendix E.5** to demonstrate that our CEM can effectively enhance the baseline's generation quality even in more challenging scenarios. It should be noted that these visualizations are provided for qualitative analysis, while the quantitative results reported in VBench and other benchmarks already cover a wide range of scenarios.

---

> > ### Comment · Reviewer_m1Xm · 2025-11-24
> >
> > Thanks for the reply.
> > * For the few-step experiments, I think the author misunderstood my points. I am curious about the performance of the work applied to some step-distilled models.
> > * For the motivation, lots of work emphasizes the importance of cumulative errors or error compensation in this area. So, I do not think it is insightful. Moreover, the solution, i.e., DP is trivial.
> >
> > Overall, I think this paper is comprehensive. However, considering feature caching is a very crowded area, and this paper does not make a sufficient contribution, I lean towards keeping the current score.

---

> > > ### Author Response · Authors · 2025-11-25
> > > **Official Comment by Authors**
> > >
> > > We sincerely thank the reviewer for timely responses!
> > >
> > > **1).** Regarding the combination of CEM with few‑step diffusion:
> > > * CEM is essentially a **plugin** that improves caching strategies and thus builds on an **existing** caching-based method for quality improvement. To the best of our knowledge, **no prior work** in the DiT visual generation domain has explored integrating few‑step distillation with caching acceleration.
> > > * Few‑step distillation already meets practical speed requirements (< 0.5 s), with its main challenges lying in enhancing generation quality and cross‑model generalization.
> > >     * For improving generation quality, caching-based methods do not possess such an advantage (that's why we want to optimize fidelity for caching). Therefore, combining few‑step distillation with caching acceleration further **sacrifices** quality for speed and **compromises** the training‑free advantage of caching, providing **limited** value for practical applications.
> > >     * Regarding generalization, this is the key **distinction** between caching acceleration and few‑step distillation. The **training‑free** nature of caching allows **rapid adaptation** to other generation models (with minimal manual configuration), whereas combining it with few‑step distillation would **forfeit** this advantage.
> > >
> > > Therefore, the tables provided in our previous response were primarily intended to illustrate the above key points.
> > >
> > > We sincerely thank the reviewer for attention to the generalization capability of our CEM. In response to this expectation, we have included additional experiments on the Wan 2.1 model and the TeaCache baseline, and reported the performance of our CEM under more challenging scenarios.
> > >
> > > At present, we have conducted generalization tests on **eight generation models, five caching‑based acceleration baselines, and one quantization model**. These results demonstrate the effectiveness and strong generalization capability of our method in improving generation fidelity within caching acceleration frameworks.
> > >
> > > **2)**. We fully agree with the reviewer that all caching‑based acceleration methods fundamentally rely on feature error or similarity. Our focus lies in proposing a **distinct** way to **analyze** and **exploit** this error, which has also been recognized by Reviewer WEPH, XEtZ, and Hw5z:
> > > * We model the error using the **joint variation** of timesteps and cache intervals (Sec. 3.1), explicitly capturing the effect of interval changes. To our knowledge, this is the first work to analyze error distributions by jointly considering cache operations.
> > > * We model the caching error distribution from random samples and demonstrate its **consistency** with actual inference, enabling online use **without** additional overhead (Sec. 3.1, Appendix. B, Appendix. C.2).
> > > * This error modeling and utilization process adapts **automatically across different models**, **without** relying on manually crafted heuristics.
> > >
> > > In summary, our method builds on the commonly used feature error, re‑formulates it in an offline manner, and incorporates additional relevant variables, enabling fine‑grained guidance for caching strategies and strong adaptive generalization.
> > >
> > > We would also be grateful for any additional related work the reviewer may kindly suggest to further enrich our discussion!
> > >
> > > **3).** Regarding our DP‑based caching strategy:
> > > * We prioritize **simplicity and efficiency**, ensuring that it introduces **no extra** online overhead, and it is "conceptual elegant" noted by Reviewer 98fT.
> > > * Dynamic programming naturally exploits the **optimal substructure** of cumulative error, as highlighted by Reviewer Hw5z, it "provides a structured alternative to previous heuristic caching methods".
> > > * Based on our modeled error, the DP formulation yields an **optimal** caching strategy and allows for straightforward theoretical justification (Appendix C.2 and previous responses).
> > >
> > > Therefore, we believe that dynamic programming is well suited for caching‑based acceleration, and our extensive experiments substantiate its effectiveness. This also represents an interesting application of classical algorithms to a modern problem.
> > >
> > > We once again sincerely thank the reviewer for timely responses! We welcome any further questions and will do our utmost to address all concerns before the rebuttal period concludes.

---

### Official Review · Reviewer_98fT · 2025-10-30

**Soundness:** 3
**Presentation:** 2
**Contribution:** 3
**Rating:** 6
**Confidence:** 3

**Summary:**

This paper proposes CEM (Cumulative Error Minimization), a training-free, plug-and-play acceleration method designed to improve the generation fidelity of Diffusion Transformer (DiT) models under caching-based acceleration. CEM tackles this by formulating caching strategy optimization as a cumulative error minimization problem. It first performs offline error modeling to characterize the joint effect of denoising timesteps and cache intervals, building a reusable prior without retraining or online computation. Then, a dynamic programming algorithm derives the optimal cache schedule that minimizes the total accumulated error under arbitrary acceleration budgets. CEM is model-agnostic, incurs no runtime overhead, and can be directly integrated with existing acceleration or quantization frameworks. Extensive experiments across seven diffusion models and three tasks (text-to-image, text-to-video, and class-to-image generation) show that CEM consistently improves generation fidelity while maintaining or even improving inference speed.

**Strengths:**

1. The paper introduces a novel formulation of the caching optimization problem for Diffusion Transformers as a cumulative error minimization task. Unlike prior caching-based accelerators that rely on fixed intervals or heuristic scheduling, CEM models the joint variation of denoising timesteps and cache intervals and solves for an optimal caching plan through dynamic programming. This method combining offline error modeling with discrete optimization is original and conceptually elegant, extending beyond prior local correction approaches such as ToCa, DuCa, and TaylorSeer.

2. The methodology is technically sound and well-supported by extensive experiments. The paper conducts thorough evaluations across seven generative models and three task categories (text-to-image, text-to-video, and class-to-image), demonstrating consistent fidelity gains under identical FLOPs or latency.

**Weaknesses:**

1. Limited theoretical justification for the cumulative error approximation.

The proposed cumulative error approximation (Eq. 2) is empirically validated but lacks a clear theoretical foundation. The assumption that a cumulative sum over per-step error distributions sufficiently approximates the true propagation of caching error is plausible yet heuristic. A deeper analysis — for example, quantifying the approximation gap between estimated and actual cumulative error or providing theoretical error bounds — would make the dynamic programming framework more convincing.

2. Limited analysis of computational trade-offs and scalability.

Although CEM claims to introduce no runtime overhead, the paper does not detail the computational cost of the offline modeling phase, especially for large-scale models (e.g., FLUX or Hunyuan). Clarifying the one-time cost and memory footprint of building the offline error prior would help readers assess practical feasibility in industrial settings.

3. Missing comparison with learned caching optimization methods.

Although CEM is positioned as training-free, the paper does not compare against recent learning-based caching optimization approaches such as HarmoniCa [1] or Learning-to-Cache [2], which explicitly learn adaptive caching schedules from data. Such baselines would better contextualize how much performance CEM gains or sacrifices relative to methods that perform end-to-end cache learning. Without these, the claimed superiority of CEM’s offline optimization remains partially unquantified.

[1] HarmoniCa: HarmonizingTraining and Inference for Better Feature Caching in Diffusion Transformer Acceleration, ICML 2025.

[2] Learning to-cache: Accelerating diffusion transformer via layer caching, NeurIPS 2024.

**Questions:**

Please see the above weaknesses.

---

> ### Author Response · Authors · 2025-11-24
> **Official Comment by Authors (1/3)**
>
> We sincerely appreciate your recognition of our work and your insightful suggestions. Your feedback has been instrumental in helping us improve CEM into a more complete and comprehensive contribution.
>
> >**W1**. Limited theoretical justification for the cumulative error approximation. The proposed cumulative error approximation (Eq. 2) is empirically validated but lacks a clear theoretical foundation. The assumption that a cumulative sum over per-step error distributions sufficiently approximates the true propagation of caching error is plausible yet heuristic. A deeper analysis — for example, quantifying the approximation gap between estimated and actual cumulative error or providing theoretical error bounds — would make the dynamic programming framework more convincing.
>
> **A1**. We sincerely thank the reviewer for the constructive comments. The theoretical analysis further supports the rationality and effectiveness of our proposed method.
>
> **Summary for A1**: The following answer mainly includes:
> * 1). Theoretical analysis of the cumulative error approximation.
> * 2). Theoretical analysis of the dynamic programming.
>
> **A1  1). Theoretical analysis of the cumulative error approximation.**
>
> To establish the error bound of the cumulative error estimation, we first demonstrate how the offline error approximates the online error, and then analyze the discrepancy between the online error and the true propagated error.
>
> We obtained content-agnostic offline error $\mathcal{E}^*$ through offline error modeling and cumulative error approximation.
> We define the actual error under the same operation during formal inference as $\mathcal{E}$ (not the $\mathcal{E}$ defined in Eq.1), while the true propagated error of the cached latent during the denoising process is $\hat{\mathcal{E}}$.
>
> We assume the DiT is Lipschitz continuous (common in DiT for bounded error propagation), i.e., $\|\mathcal{D}(x, t) - \mathcal{D}(y, t)\| \leq L \|x - y\|$ for some Lipschitz constant $L > 0$.
>
> First, we establish the approximate relationship between the offline error $\mathcal{E}^\*$ and the online error $\mathcal{E}$.
> The $\mathcal{E}^\*$ is an empirical estimate of the error distribution using $N_s$ random samples, while $\mathcal{E}$ is computed for a specific inference sample.
>
> **Theorem.1**: *Under the unified error distribution, $|\mathcal{E}^\*(t, n) - \mathcal{E}(t, n)|$ is bounded by:*
>
> $|\mathcal{E}^\*(t, n) - \mathcal{E}(t, n)| \leq \sqrt{\frac{\log(2/\delta)}{2 N_s}} + \epsilon_{\text{var}}$,
>
> *where $\delta \in (0,1)$ is a confidence parameter, and $\epsilon_{\text{var}}$ is a small variance term (empirically small, as per Fig. 3(a)).*
>
> **Proof:**
> Since DiT content follows the same underlying distribution, we assume that the caching error in DiT also follows the unified distribution.
>
> Let $\mu(t,n)$ be the mean of the cosine error over all possible contents. $\mathcal{E}(t,n)$ for offline is the empirical mean $\hat{\mu}(t,n) = \frac{1}{N_s} \sum_i \mathcal{E}_i(t,n)$, and for online it's a single-sample estimate.
>
> By Hoeffding's inequality (cosine errors are in [0,1]), with probability at least $1 - \delta$:
>
> $|\hat{\mu}(t,n) - \mu(t,n)| \leq \sqrt{\frac{\log(2/\delta)}{2 N_s}}$.
>
> The $\mathcal{E}(t,n)$ deviates from $\mu(t,n)$ by at most the empirical variance $\epsilon_{\text{var}}$ (small, as per the analysis: low variance across contents and intervals).
>
> For cumulation: CUMSUM is a linear operator, the bound propagates:
>
> $|\mathcal{E}^\*(t,n) - \mathcal{E}(t,n)| \leq \sum_{\tau=1}^t |\mathcal{E}^\*(\tau,n) - \mathcal{E}(\tau,n)| \leq t \left( \sqrt{\frac{\log(2/\delta)}{2 N_s}} + \epsilon_{\text{var}} \right)$.
>
> For total timesteps $T$, the worst-case bound is $O(T / \sqrt{N_s})$, but empirically tighter due to low $\epsilon_{\text{var}}$.
> This holds because extended experiments (Appendix.B.2) show consistency across prompt sources, confirming the i.i.d. assumption.
> Thus, $\mathcal{E}^* \approx \mathcal{E}$ with high probability (Appendix.C.2 supports this).
>
> **Theorem.2**: *The difference $|\mathcal{E}(t,n) - \hat{\mathcal{E}}(t,n)|$ is bounded by:*
>
> $|\mathcal{E}(t,n) - \hat{\mathcal{E}}(t,n)| \leq L \cdot \sum_{\tau=1}^t \mathcal{E}(\tau,n) + \epsilon_{\text{prop}}$,
>
> *where $L$ is the Lipschitz constant of $\mathcal{D}$, and $\epsilon_{\text{prop}}$ is a small propagation residual.*

---

> ### Author Response · Authors · 2025-11-24
> **Official Comment by Authors (2/3)**
>
> **Proof:**
> In DiT, each timestep's output is $\mathcal{D}(x_t, t) = f(\mathcal{D}(x_{t+1}, t+1) + \delta)$, where $f$ is the transformer layer, and $\delta$ is noise/caching perturbation.
> Caching introduces error $\mathcal{E}(t,n)$ at each step, which propagates to future steps. The true propagated error $\hat{\mathcal{E}}(t,n)$ satisfies a recurrence: $\hat{\mathcal{E}}(t,n) = \mathcal{E}(t,n) + g(\hat{\mathcal{E}}(t+1,n))$, where $g$ models propagation (approximately linear due to DiT's residual connections and attention linearity).
> The CUMSUM approximation assumes $g \approx \text{Id}$ (identity, i.e., direct summation), which holds because of "high structural similarity between input and output" (as noted in TeaCache).
>
> By the Lipschitz assumption, propagation error is bounded: $|g(e) - e| \leq L \cdot e + \epsilon_{\text{prop}}$, where $\epsilon_{\text{prop}}$ captures non-linear residuals (small in DiT).
> Unrolling the recurrence over $t$ steps yields the bound via triangle inequality.
> For the caching strategy, the bound extends to the full sequence, as DP preserves substructure:
>
> $|\mathcal{E}(t,n) - \hat{\mathcal{E}}(t,n)| \leq  L \cdot \sum_{\tau=1}^t \mathcal{E}(\tau,n) + \epsilon_{\text{prop}}$.
>
> Thus, $\mathcal{E} \approx \hat{\mathcal{E}}$, with the bound tightening for smaller single-step errors.
>
> Finally, by chaining Theorems 1 and 2 (triangle inequality):
>
> $|\mathcal{E}^\*(t,n) - \hat{\mathcal{E}}(t,n)| \leq |\mathcal{E}^\*(t,n) - \mathcal{E}(t,n)| + |\mathcal{E}(t,n) - \hat{\mathcal{E}}(t,n)| \leq t \left( \sqrt{\frac{\log(2/\delta)}{2 N_s}} + \epsilon_{\text{var}} \right) + L \cdot \sum_{\tau=1}^t \mathcal{E}(\tau,n) + \epsilon_{\text{prop}}$.
>
> This upper bound is $O(T/\sqrt{N_s} + L \cdot \mathcal{E}_{\text{total}})$, where "total" is total single-step error. Empirically, it's small, confirming the approximation.
>
> **A1  2). Theoretical analysis of the dynamic programming.**
>
> For the dynamic programming, we mainly demonstrate the optimality of its optimal substructure and further analyze the computational complexity.
>
> Proof of Optimality:
>
> Suppose there exists an optimal strategy $S^\*$ for $dp[t][j+1]$ that chooses interval $n$, but the sub-strategy $S'$ for $dp[t+n][j]$ is not optimal, i.e., there exists a better sub-strategy $S''$ with lower error for $dp[t+n][j]$.
>
> Then, replacing $S'$ with $S''$ in $S^\*$ would yield a new strategy with total error:
>
> $\mathcal{E}^\*(t, n) + \text{error}(S'') < \mathcal{E}^\*(t, n) + \text{error}(S')$,
>
> contradicting the optimality of $S^\*$. Thus, subproblems must be optimal.
>
> This holds because:
> * 1). The denoising process is sequential and acyclic (timesteps decrease from $T$ to $1$).
> * 2). Errors are additive ($\mathcal{E}^\*(t, n)$ is independent of future choices, depending only on the current interval and offline modeling).
> * 3). The budget $N_c$ is fixed, and choices do not overlap (each caching operation covers distinct timestep segments).
>
> Time Complexity:
>
> The DP table has $O(T \cdot N_c)$ entries ($t \in [1, T]$, $j \in [1, N_c]$). For each entry $dp[t][j+1]$ (for $j \geq 0$), we evaluate the minimum over $|\mathcal{N}|$ possible intervals $n$, each requiring $O(1)$ time to compute $\mathcal{E}^\*(t, n) + dp[t+n][j]$ (assuming $\mathcal{E}^\*(t, n)$ is precomputed offline in $O(T \cdot |\mathcal{N}|)$ time, as it is content‑agnostic).
> Thus, filling the table takes $O(T \cdot N_c \cdot |\mathcal{N}|)$ time. Backtracking: $O(T)$ time (traverse the path of $N_c$ choices, each step $O(1)$).
>
> Overall time complexity: $O(T \cdot N_c \cdot |\mathcal{N}|)$, which is efficient since $T$ is small (e.g., 50), $N_c < T$ (budget‑constrained), and $|\mathcal{N}|$ is small (e.g., 10 practical intervals). The analysis notes no additional overhead during inference, as DP runs offline once per model. The actual time consumption of the dynamic programming (DP) process can be found in Appendix.C.3, Tab.12.
>
> Space Complexity:
>
> DP table: $O(T \cdot N_c)$ space (store floats for errors). Precomputed $\mathcal{E}^\*(t, n)$: $O(T \cdot |\mathcal{N}|)$ space. Backtracking can use the table itself (no extra space) or a separate predecessor array ($O(T \cdot N_c)$). This complexity is polynomial and scalable, enabling "optimal cache‑interval combination that can be shared across multiple generations without additional overhead" (Sec.3.2 analysis).
>
> >**W2**. Limited analysis of computational trade-offs and scalability. Although CEM claims to introduce no runtime overhead, the paper does not detail the computational cost of the offline modeling phase, especially for large-scale models (e.g., FLUX or Hunyuan). Clarifying the one-time cost and memory footprint of building the offline error prior would help readers assess practical feasibility in industrial settings.
>
> **A2**. **Summary for A2**: The following answer mainly includes:
> * 1). Computational and memory overhead of offline error modeling.
> * 2). Scalability of offline error modeling.

---

> ### Author Response · Authors · 2025-11-24
> **Official Comment by Authors (3/3)**
>
> **A2  1). Computational and memory overhead of offline error modeling.**
>
> We report the time and memory overhead of our offline error modeling for each generation model discussed in the paper. Regarding this overhead, we would like to make the following clarifications:
> * For each model, the offline error modeling needs to be performed **only once**. The modeled error can then be permanently **reused** across different configurations or **shared** by different acceleration methods.
> * During modeling, random content generation incurs inherent overhead (see “w/o OEM” in Tab. M). The **extra** cost beyond this reflects the true overhead of our offline error modeling, which records the sensitivity of random generations to acceleration.
> * The overhead of offline error modeling is not incurred during inference. Once modeling is completed, invoking CEM and applying the optimized caching strategy introduce only **negligible** runtime overhead.
>
> **Table.A**: Offline costs.
>
> |  | |       Text2Image       |             |  |      Text2Video       |             | Class2Image |
> |:----------:|:--------------:|:-----------:|:-----------:|:---------------:|:-----------:|:-----------:|:----------------:|
> | Models | FLUX.1-dev | PixArt-$\alpha$ | SD15 | Hunyuan | Wan21 | OpenSora | DiT-XL/2 |
> | Time w/o. OEM | 1.92h | 13.52m | 38.88m | 4.72h | 2.73h | 3.63h | 19.63m |
> | Time w/. OEM | 2.08h | 14.71m | 43.83m | 5.21h | 4.15h | 4.65h | 25.52m |
> | Memory w/o. OEM | 43.42GB | 22.31GB | 3.65GB | 57.36GB | 40.83GB | 52.40GB | 4.09GB |
> | Memory w/. OEM | 53.06GB | 22.65GB | 3.67GB | 72.62GB | 63.46GB | 52.40GB | 4.65GB |
>
> The offline memory overhead primarily comes from storing intermediate features during inference to compute differences across cache intervals.
>
> On average, offline error modeling increases memory usage by only **15.8%** and modeling time by **16.8%** compared with random content generation, both remaining below 20%. Considering the substantial performance gains achieved by CEM and its zero inference‑time overhead, this cost is fully acceptable.
>
> **A2  2). Scalability of offline error modeling.**
>
> The overhead of offline error modeling is **influenced** by the model scale. The time overhead mainly depends on the model's inherent inference speed, which varies with its stochastic generation process, while our additional cost remains relatively small (averaging 16.8%). Similarly, the memory overhead is dominated by the model parameters themselves, with our caching and computation contributing only about 15.8% on average.
>
> Overall, larger models generally incur higher absolute overhead. However, two points should be noted:
> * Most of the overhead originates from the model **itself** rather than from our modeling process.
> * The relationship is **not strictly monotonic**, for example, SD15 is smaller than DiT-XL/2, yet its modeling time is longer.
>
> >**W3**. Missing comparison with learned caching optimization methods. Although CEM is positioned as training-free, the paper does not compare against recent learning-based caching optimization approaches such as HarmoniCa or Learning-to-Cache, which explicitly learn adaptive caching schedules from data. Such baselines would better contextualize how much performance CEM gains or sacrifices relative to methods that perform end-to-end cache learning. Without these, the claimed superiority of CEM’s offline optimization remains partially unquantified.
>
> **A3**.
> We thank the reviewer for the valuable feedback. The two methods mentioned, L2C and HarmoniCa, are learning-based approaches with relatively low acceleration ratios (below 2×). A direct comparison with our CEM would therefore be somewhat inequitable, as our method is completely training-free. Nevertheless, to better illustrate the superiority of our CEM, we adjust the acceleration efficiency to align with that of L2C and HarmoniCa and conduct a comparative evaluation in terms of generation quality.
>
> **Table.B**: Comparison with learning‑based acceleration methods on DiT-XL/2.
>
> | Method | Train | FLOPS(T)↓ | Spe↑ | Lat(s)↓ | Spe↑ | FID↓ | sFID↓ | IS↑ | P↑ | R↑ |
> |:-------:|:------:|:----------:|:-----:|:--------:|:-----:|:-----:|:------:|:----:|:----:|:----:|
> | Origin | – | 23.74 | 1.00× | 0.53 | 1.00× | 2.25 | 4.33 | 239.93 | 0.80 | 0.59 |
> | L2C | yes | – | – | – | 1.25× | 2.62 | 4.50 | 233.26 | 0.79 | 0.59 |
> | HarmoniCa | yes | – | – | – | 1.30× | 2.36 | 4.24 | 238.74 | 0.81 | 0.60 |
> | **TaylorSeer+Ours** | **no** | **11.87** | **2.00×** | **0.37** | **1.43×** | **2.27** | 4.42 | **242.45** | **0.81** | 0.59 |
>
> It can be observed that our CEM achieves **higher** acceleration efficiency and **better** fidelity even under a **training‑free** setting.

---

### Official Review · Reviewer_Hw5z · 2025-10-31

**Soundness:** 3
**Presentation:** 3
**Contribution:** 3
**Rating:** 4
**Confidence:** 3

**Summary:**

This paper proposes a training-free, plug-and-play acceleration framework for Diffusion Transformers called Cumulative Error Minimization (CEM). The method performs offline error modeling to estimate the joint distribution of denoising steps and cache intervals, and then applies dynamic programming optimization to minimize cumulative cache errors under a given acceleration budget. CEM can be seamlessly integrated into existing acceleration and quantization frameworks, improving fidelity across multiple generation tasks without additional inference cost. Overall, the paper presents an efficient, general, and theoretically grounded acceleration strategy for diffusion models.

**Strengths:**

1. Introduces **dynamic programming** into diffusion caching optimization, combined with offline error modeling, which provides a structured alternative to previous heuristic caching methods.

2. The **plug-and-play** design requires no additional training and can be easily integrated into existing acceleration or quantization frameworks, demonstrating strong engineering practicality.

3. Comprehensive experiments across multiple tasks and models show consistent fidelity improvement at fixed acceleration ratios.

**Weaknesses:**

1. **Limited theoretical analysis.** The cumulative error approximation is only empirically motivated, lacking a formal discussion of convergence, stability, or optimality guarantees. The complexity and optimality conditions of the DP procedure are also not analyzed.

2. **Restricted applicability.** CEM appears to be tailored for iterative denoising structures and may not extend to one-step or non-iterative diffusion models. The paper could further discuss potential adaptations to these architectures.

**Questions:**

1. Have the authors evaluated CEM on non-visual diffusion tasks? Cross-modal results would help validate the claimed generality.

2. Can the authors quantify the relationship between cumulative error and perceptual generation quality, and analyze the stability of this relationship across samplers, resolutions, and long-sequence scenarios? Does cumulative error accumulation cause performance degradation in long-horizon tasks, and how might this be mitigated?

3. Please provide more details about the offline modeling cost and complexity—for example, sample size, runtime, and scalability to larger models.

---

> ### Author Response · Authors · 2025-11-24
> **Official Comment by Authors (1/6)**
>
> We sincerely appreciate your valuable feedback and recognition of our work. We have carefully addressed each of your comments and provided detailed responses below.
>
> >**W1**. Limited theoretical analysis. The cumulative error approximation is only empirically motivated, lacking a formal discussion of convergence, stability, or optimality guarantees. The complexity and optimality conditions of the DP procedure are also not analyzed.
>
> **A1**.
> We sincerely thank the reviewer for the constructive comments. The theoretical analysis further supports the rationality and effectiveness of our proposed method.
>
> However, to the best of our knowledge, no existing work has theoretically modeled or explained the propagation pattern of caching acceleration errors in DiT models, and such an investigation is beyond the scope of our work. Therefore, to facilitate a theoretical interpretation of our method, we introduce several assumptions inspired by our experimental observations and use them as the foundation for the theoretical analysis of our cumulative error approximation and dynamic programming approach.
>
> **Summary for A1**: The following answer mainly includes:
> * 1). Theoretical analysis of the cumulative error approximation.
> * 2). Theoretical analysis of the dynamic programming.
> * 3). Quantitative results of the cumulative error approximation.
>
> The following section presents the detailed content of the response:
>
> **A1  1). Theoretical analysis of the cumulative error approximation.**
>
> To establish the error bound of the cumulative error estimation, we first demonstrate how the offline error approximates the online error, and then analyze the discrepancy between the online error and the true propagated error.
>
> We obtained content-agnostic offline error $\mathcal{E}^*$ through offline error modeling and cumulative error approximation.
> We define the actual error under the same operation during formal inference as $\mathcal{E}$ (not the $\mathcal{E}$ defined in Eq.1), while the true propagated error of the cached latent during the denoising process is $\hat{\mathcal{E}}$.
>
> We assume the DiT is Lipschitz continuous (common in DiT for bounded error propagation), i.e., $\|\mathcal{D}(x, t) - \mathcal{D}(y, t)\| \leq L \|x - y\|$ for some Lipschitz constant $L > 0$.
>
> First, we establish the approximate relationship between the offline error $\mathcal{E}^\*$ and the online error $\mathcal{E}$.
> The $\mathcal{E}^\*$ is an empirical estimate of the error distribution using $N_s$ random samples, while $\mathcal{E}$ is computed for a specific inference sample.
>
> **Theorem.1**: *Under the unified error distribution, $|\mathcal{E}^\*(t, n) - \mathcal{E}(t, n)|$ is bounded by:*
>
> $|\mathcal{E}^\*(t, n) - \mathcal{E}(t, n)| \leq \sqrt{\frac{\log(2/\delta)}{2 N_s}} + \epsilon_{\text{var}}$,
>
> *where $\delta \in (0,1)$ is a confidence parameter, and $\epsilon_{\text{var}}$ is a small variance term (empirically small, as per Fig. 3(a)).*
>
> **Proof:**
> Since DiT content follows the same underlying distribution, we assume that the caching error in DiT also follows the unified distribution.
>
> Let $\mu(t,n)$ be the mean of the cosine error over all possible contents. $\mathcal{E}(t,n)$ for offline is the empirical mean $\hat{\mu}(t,n) = \frac{1}{N_s} \sum_i \mathcal{E}_i(t,n)$, and for online it's a single-sample estimate.
>
> By Hoeffding's inequality (cosine errors are in [0,1]), with probability at least $1 - \delta$:
>
> $|\hat{\mu}(t,n) - \mu(t,n)| \leq \sqrt{\frac{\log(2/\delta)}{2 N_s}}$.
>
> The $\mathcal{E}(t,n)$ deviates from $\mu(t,n)$ by at most the empirical variance $\epsilon_{\text{var}}$ (small, as per the analysis: low variance across contents and intervals).
>
> For cumulation: CUMSUM is a linear operator, the bound propagates:
>
> $|\mathcal{E}^\*(t,n) - \mathcal{E}(t,n)| \leq \sum_{\tau=1}^t |\mathcal{E}^\*(\tau,n) - \mathcal{E}(\tau,n)| \leq t \left( \sqrt{\frac{\log(2/\delta)}{2 N_s}} + \epsilon_{\text{var}} \right)$.
>
> For total timesteps $T$, the worst-case bound is $O(T / \sqrt{N_s})$, but empirically tighter due to low $\epsilon_{\text{var}}$.
> This holds because extended experiments (Appendix.B.2) show consistency across prompt sources, confirming the i.i.d. assumption.
> Thus, $\mathcal{E}^* \approx \mathcal{E}$ with high probability (Appendix.C.2 supports this).
>
> **Theorem.2**: *The difference $|\mathcal{E}(t,n) - \hat{\mathcal{E}}(t,n)|$ is bounded by:*
>
> $|\mathcal{E}(t,n) - \hat{\mathcal{E}}(t,n)| \leq L \cdot \sum_{\tau=1}^t \mathcal{E}(\tau,n) + \epsilon_{\text{prop}}$,
>
> *where $L$ is the Lipschitz constant of $\mathcal{D}$, and $\epsilon_{\text{prop}}$ is a small propagation residual.*

---

> ### Author Response · Authors · 2025-11-24
> **Official Comment by Authors (2/6)**
>
> **Proof:**
> In DiT, each timestep's output is $\mathcal{D}(x_t, t) = f(\mathcal{D}(x_{t+1}, t+1) + \delta)$, where $f$ is the transformer layer, and $\delta$ is noise/caching perturbation.
> Caching introduces error $\mathcal{E}(t,n)$ at each step, which propagates to future steps. The true propagated error $\hat{\mathcal{E}}(t,n)$ satisfies a recurrence: $\hat{\mathcal{E}}(t,n) = \mathcal{E}(t,n) + g(\hat{\mathcal{E}}(t+1,n))$, where $g$ models propagation (approximately linear due to DiT's residual connections and attention linearity).
> The CUMSUM approximation assumes $g \approx \text{Id}$ (identity, i.e., direct summation), which holds because of "high structural similarity between input and output" (as noted in TeaCache).
>
> By the Lipschitz assumption, propagation error is bounded: $|g(e) - e| \leq L \cdot e + \epsilon_{\text{prop}}$, where $\epsilon_{\text{prop}}$ captures non-linear residuals (small in DiT).
> Unrolling the recurrence over $t$ steps yields the bound via triangle inequality.
> For the caching strategy, the bound extends to the full sequence, as DP preserves substructure:
>
> $|\mathcal{E}(t,n) - \hat{\mathcal{E}}(t,n)| \leq  L \cdot \sum_{\tau=1}^t \mathcal{E}(\tau,n) + \epsilon_{\text{prop}}$.
>
> Thus, $\mathcal{E} \approx \hat{\mathcal{E}}$, with the bound tightening for smaller single-step errors.
>
> Finally, by chaining Theorems 1 and 2 (triangle inequality):
>
> $|\mathcal{E}^\*(t,n) - \hat{\mathcal{E}}(t,n)| \leq |\mathcal{E}^\*(t,n) - \mathcal{E}(t,n)| + |\mathcal{E}(t,n) - \hat{\mathcal{E}}(t,n)| \leq t \left( \sqrt{\frac{\log(2/\delta)}{2 N_s}} + \epsilon_{\text{var}} \right) + L \cdot \sum_{\tau=1}^t \mathcal{E}(\tau,n) + \epsilon_{\text{prop}}$.
>
> This upper bound is $O(T/\sqrt{N_s} + L \cdot \mathcal{E}_{\text{total}})$, where "total" is total single-step error. Empirically, it's small, confirming the approximation.
>
> **A1  2). Theoretical analysis of the dynamic programming.**
>
> For the dynamic programming, we mainly demonstrate the optimality of its optimal substructure and further analyze the computational complexity.
>
> Proof of Optimality:
>
> Suppose there exists an optimal strategy $S^\*$ for $dp[t][j+1]$ that chooses interval $n$, but the sub-strategy $S'$ for $dp[t+n][j]$ is not optimal, i.e., there exists a better sub-strategy $S''$ with lower error for $dp[t+n][j]$.
>
> Then, replacing $S'$ with $S''$ in $S^\*$ would yield a new strategy with total error:
>
> $\mathcal{E}^\*(t, n) + \text{error}(S'') < \mathcal{E}^\*(t, n) + \text{error}(S')$,
>
> contradicting the optimality of $S^\*$. Thus, subproblems must be optimal.
>
> This holds because:
> * 1). The denoising process is sequential and acyclic (timesteps decrease from $T$ to $1$).
> * 2). Errors are additive ($\mathcal{E}^\*(t, n)$ is independent of future choices, depending only on the current interval and offline modeling).
> * 3). The budget $N_c$ is fixed, and choices do not overlap (each caching operation covers distinct timestep segments).
>
> Time Complexity:
>
> The DP table has $O(T \cdot N_c)$ entries ($t \in [1, T]$, $j \in [1, N_c]$). For each entry $dp[t][j+1]$ (for $j \geq 0$), we evaluate the minimum over $|\mathcal{N}|$ possible intervals $n$, each requiring $O(1)$ time to compute $\mathcal{E}^\*(t, n) + dp[t+n][j]$ (assuming $\mathcal{E}^\*(t, n)$ is precomputed offline in $O(T \cdot |\mathcal{N}|)$ time, as it is content‑agnostic).
> Thus, filling the table takes $O(T \cdot N_c \cdot |\mathcal{N}|)$ time. Backtracking: $O(T)$ time (traverse the path of $N_c$ choices, each step $O(1)$).
>
> Overall time complexity: $O(T \cdot N_c \cdot |\mathcal{N}|)$, which is efficient since $T$ is small (e.g., 50), $N_c < T$ (budget‑constrained), and $|\mathcal{N}|$ is small (e.g., 10 practical intervals). The analysis notes no additional overhead during inference, as DP runs offline once per model. The actual time consumption of the dynamic programming (DP) process can be found in Appendix.C.3, Tab.12.
>
> Space Complexity:
>
> DP table: $O(T \cdot N_c)$ space (store floats for errors). Precomputed $\mathcal{E}^\*(t, n)$: $O(T \cdot |\mathcal{N}|)$ space. Backtracking can use the table itself (no extra space) or a separate predecessor array ($O(T \cdot N_c)$). This complexity is polynomial and scalable, enabling "optimal cache‑interval combination that can be shared across multiple generations without additional overhead" (Sec.3.2 analysis).

---

> ### Author Response · Authors · 2025-11-24
> **Official Comment by Authors (3/6)**
>
> **A1  3). Quantitative results of the cumulative error approximation.**
>
> In addition to the theoretical proof, we further verify this approximation relationship from an experimental perspective.
>
> Based on the above analysis, we conducted quantitative experiments to measure the difference between cumulative (offline) and actual (online) errors. We report the error gaps at key timesteps on FLUX.1‑dev and Hunyuan, with additional comparisons provided in the main paper (Fig. 3(c)) and Appendix C.2 (Fig. 9).
>
> **Table.A**: Offline error vs. online error on FLUX.1-dev with cache interval=5.
>
> | Timestep | 0     | 5     | 10    | 15    | 20    | 25    | 30    | 35    | 40    | 45    | 49    |
> |:---------:|:-----:|:-----:|:-----:|:-----:|:-----:|:-----:|:-----:|:-----:|:-----:|:-----:|:-----:|
> | Online error  | 0.000 | 0.010 | 0.292 | 0.425 | 0.566 | 0.671 | 0.745 | 0.822 | 0.917 | 1.039 | 1.221 |
> | Offline error | 0.000 | 0.010 | 0.289 | 0.433 | 0.559 | 0.666 | 0.745 | 0.823 | 0.902 | 1.023 | 1.231 |
>
> **Table.B: Offline error vs. online error on Hunyuan with cache interval=5.**
>
> | Timestep | 0     | 5     | 10    | 15    | 20    | 25    | 30    | 35    | 40    | 45    | 49    |
> |:---------:|:-----:|:-----:|:-----:|:-----:|:-----:|:-----:|:-----:|:-----:|:-----:|:-----:|:-----:|
> | Online error  | 0.000 | 0.009 | 0.383 | 0.581 | 0.778 | 0.956 | 1.085 | 1.257 | 1.494 | 1.784 | 2.248 |
> | Offline error | 0.000 | 0.008 | 0.363 | 0.551 | 0.748 | 0.934 | 1.102 | 1.272 | 1.490 | 1.747 | 2.183 |
>
> As presented in Tables, the final difference is only **0.80%** on FLUX.1‑dev and **2.89%** on Hunyuan, indicating that the offline error **closely approximates** the actual error.
>
> >**W2**. Restricted applicability. CEM appears to be tailored for iterative denoising structures and may not extend to one-step or non-iterative diffusion models. The paper could further discuss potential adaptations to these architectures.
>
> **A2**. We analyzed the distinct objectives and comparative results of one‑step diffusion and cache‑based methods.
>
> **Table.C**: One-step diffusion vs. our CEM. Since direct comparison across models is not feasible, we report the results in percentage form.
>
> |    ImageNet | Fidelity↑ (Method/Origin)(%) | Speed↑ | Training Costs↓ |
> |:-------:|:---------:|:------:|:------:|
> | Origin | 100.0 | 1.00× | – |
> | ShortCut(ICLR25) [1] | 42.9 | ~100 | TPUv3(1–2 days) |
> | **Ours** | **99.3** | 3.56 | **Free** |
>
> | PixArt-$\alpha$  | Fidelity↑ (Method/Origin)(%) | Speed↑ | Training Costs↓ |
> |:-------:|:---------:|:------:|:------:|
> | Origin | 100.0 | 1.00× | – |
> | SIM(NIPS24) [2] | 81.2 | ~30 | 4 A100s(2 days) |
> | **Ours** | **100.1** | 2.35 | **Free** |
>
> | SD15 | Fidelity↑ (Method/Origin)(%) | Speed↑ | Training Costs↓ |
> |:-------:|:---------:|:------:|:------:|
> | Origin | 100.0 | 1.00× | – |
> | EDM(CVPR24) [3] | 76.4 | 28.7 | 72 A100s(36 hours) |
> | **Ours** | **108.8** | 4.35 | **Free** |
>
> Both one‑step diffusion and caching acceleration aim to speed up existing DiT by reducing the number of denoising iterations. However, their motivations and focuses are different:
> * **Speed-quality trade‑off**.
> One‑step diffusion models achieve generation through retraining, leading to acceleration but often at the expense of visual quality. In contrast, our CEM provides training‑free acceleration while preserving the original model's generation fidelity.
> * **Cost of Acceleration**.
> The computational costs of the two approaches differ significantly. One‑step diffusion entails substantial retraining overhead (in above table). In contrast, we offer training‑free acceleration. Our CEM is plug‑and‑play, achieving a superior balance between acceleration and generation quality without incurring additional computational cost.
> * **Generalization**.
> One‑step diffusion relies on complex, experience‑driven modules and costly training, limiting its ability to generalize across model. In contrast, our CEM is plug‑and‑play and can be directly applied to various visual generative DiT models, demonstrating strong generalization.
> * **Flexibility of Acceleration**.
> One‑step diffusion offers a fixed acceleration, limiting its applicability in scenarios that prioritize high fidelity over speed. In contrast, our CEM allows flexible control of the acceleration budget, supporting both quality‑oriented and speed‑oriented use cases, thus demonstrating greater practical versatility.
>
> Hence, caching‑based acceleration and one‑step diffusion have remained **independent** research directions.
>
> Although this is beyond the scope of our current work, we propose a **potential direction**: our offline error modeling can also be applied to capture pruning‑induced errors, enabling adaptive pruning for one‑step diffusion to further improve efficiency.
>
> [1]. ONE STEP DIFFUSION VIA SHORTCUT MODELS. ICLR 2025 oral.
> [2]. One-Step Diffusion Distillation through Score Implicit Matching. NIPS 2024.
> [3]. One-step Diffusion with Distribution Matching Distillation. CVPR 2024.

---

> ### Author Response · Authors · 2025-11-24
> **Official Comment by Authors (4/6)**
>
> >**Q1**. Have the authors evaluated CEM on non-visual diffusion tasks? Cross-modal results would help validate the claimed generality.
>
> **A3**.
> Existing caching‑based acceleration approaches have been explored mainly in visual generative models, while our CEM functions as a plug‑in extension to these methods. Consequently, for non‑visual diffusion tasks, there is **no available** baseline on which our plug‑in can be implemented.
>
> We understand the reviewer's concern regarding generalization. Therefore, we have added experiments on the **Wan2.1** video model and the online method **TeaCache** (Tab.1, 2, 3, 13, 14).
>
> We have conducted extensive experiments on **eight** generation models (SD15, PixArt-$\alpha$, FLUX.1-dev, Hunyuan, Wan21, OpenSora, DiT-XL/2 DDPM and DDIM), **five** acceleration baselines (ToCa, DuCa, FasterSD, TaylorSeer and TeaCache), and one **quantized** model (Q-DiT), which we believe sufficiently demonstrate the effectiveness and generalization capability of our CEM.
>
> >**Q2**. Can the authors quantify the relationship between cumulative error and perceptual generation quality, and analyze the stability of this relationship across samplers, resolutions, and long-sequence scenarios? Does cumulative error accumulation cause performance degradation in long-horizon tasks, and how might this be mitigated?
>
> **A4**. Our definition of caching error reveals how different caching configurations affect generation fidelity.
>
> **Summary for A4**: The following answer mainly includes:
> * 1). Relationship between errors and generation quality under different caching strategies.
> * 2). Robustness of our CEM method.
> * 3). Error accumulation in long-horizon tasks.
>
> **A4  1). Relationship between errors and generation quality under different caching strategies.**
>
> Your question is highly valuable and constructive. We have investigated the relationship between their associated errors and the generation quality under different caching strategies in Fig.6(c). The table below presents the correlations under varying acceleration efficiencies on the FLUX.1‑dev.
>
> **Table.D**: Relationship bridged by error between caching strategies and generation fidelity.
>
> | **Number** | 1 | 2 | 3 | 4 | 5 | 6 | 7 | 8 | 9 | 10 | 11 | 12 |
> |:-----------:|:--:|:--:|:--:|:--:|:--:|:--:|:--:|:--:|:--:|:--:|:--:|:--:|
> | **4.1×** |||||||||||||
> | **Error** | 0.1628 | 0.1658 | 0.1671 | 0.1721 | 0.1752 | 0.1797 | 0.1798 | 0.1860 | 0.1881 | 0.1904 | 0.1912 | 0.1941 |
> | **IR** | 0.9751 | 0.9729 | 0.9705 | 0.9602 | 0.9450 | 0.9370 | 0.9423 | 0.9200 | 0.8900 | 0.8841 | 0.8651 | 0.8740 |
> | **5.0×** |||||||||||||
> | **Error** | 0.1711 | 0.1741 | 0.1792 | 0.1892 | 0.1899 | 0.1930 | 0.2002 | 0.2017 | 0.2080 | 0.2097 | 0.2160 | 0.2180 |
> | **IR** | 0.8940 | 0.8900 | 0.8921 | 0.8910 | 0.8876 | 0.8840 | 0.8811 | 0.8783 | 0.8711 | 0.8700 | 0.8661 | 0.8612 |
> | **5.6×** |||||||||||||
> | **Error** | 0.2016 | 0.2019 | 0.2083 | 0.2116 | 0.2125 | 0.2162 | 0.2171 | 0.2190 | 0.2199 | 0.2214 | 0.2218 | 0.2224 |
> | **IR** | 0.9449 | 0.9334 | 0.9243 | 0.8956 | 0.8936 | 0.8742 | 0.8511 | 0.8310 | 0.8101 | 0.8092 | 0.8073 | 0.8062 |
> | **6.3×** |||||||||||||
> | **Error** | 0.2300 | 0.2303 | 0.2305 | 0.2337 | 0.2341 | 0.2361 | 0.2390 | 0.2399 | 0.2409 | 0.2420 | 0.2424 | 0.2440 |
> | **IR** | 0.9240 | 0.9205 | 0.8975 | 0.8600 | 0.8540 | 0.8444 | 0.8045 | 0.8116 | 0.7741 | 0.7481 | 0.7209 | 0.7184 |
>
> As shown in the table above, larger caching errors consistently lead to lower generation fidelity, a trend consistent with empirical observations.
> To our knowledge, this work is the **first** to explicitly reveal the relationship between caching errors and generation quality.
>
> This negative correlation holds consistently under different settings, with the curvature of this relationship varying by efficiency level (in **Fig. 6(c)**), demonstrating the robustness of the relationship.
>
> This consistency reflects the robustness of our CEM across diverse configurations. In the following section, we further validate this robustness through comprehensive experiments.

---

> ### Author Response · Authors · 2025-11-24
> **Official Comment by Authors (5/6)**
>
> **A4  2). Robustness of our CEM method.**
>
> We extend our analysis to evaluate its robustness under varying conditions.
> For each generative model, the caching error is modeled once and **reused** across all experimental settings to validate the stability of the offline error modeling. Specifically, we assess the effectiveness of CEM under different random seeds, CFG values, image resolutions (on FLUX.1‑dev), and frame counts (on Hunyuan).
>
> **Table.E**: Robustness under different seeds, CFG and resolutions (FLUX.1-dev)
>
> | Seed | Method | IR↑ | CLIP(G)↑ |
> |:----:|:--------:|:----:|:------:|
> | 0 | TaylorSeer | 0.8760 | 32.17 |
> |   | **+Ours** | **0.9205** | **32.66** |
> | 42 | TaylorSeer | 0.8625 | 32.38 |
> |   | **+Ours** | **0.9405** | **32.86** |
> | 2025 | TaylorSeer | 0.8169 | 32.50 |
> |   | **+Ours** | **0.8875** | **32.69** |
> | 3407 | TaylorSeer | 0.8217 | 31.95 |
> |   | **+Ours** | **0.9118** | **32.99** |
>
> | CFG | Method | IR↑ | CLIP(G)↑ |
> |:---:|:--------:|:----:|:------:|
> | 3.5 | TaylorSeer | 0.8760 | 32.17 |
> |     | **+Ours** | **0.9205** | **32.66** |
> | 5.5 | TaylorSeer | 0.6571 | 31.34 |
> |     | **+Ours** | **0.8867** | **32.76** |
> | 7.5 | TaylorSeer | 0.6924 | 31.75 |
> |     | **+Ours** | **0.7336** | **32.29** |
> | 9.5 | TaylorSeer | 0.6794 | 31.20 |
> |     | **+Ours** | **0.7935** | **32.17** |
>
> | Resolution | Method | IR↑ | CLIP(G)↑ |
> |:-----------:|:--------:|:----:|:------:|
> | 256 | TaylorSeer | 0.5756 | 30.97 |
> |     | **+Ours** | **0.6792** | **31.40** |
> | 512 | TaylorSeer | 0.7495 | 31.80 |
> |     | **+Ours** | **0.7700** | **32.31** |
> | 1024 | TaylorSeer | 0.8760 | 32.17 |
> |     | **+Ours** | **0.9205** | **32.66** |
> | 2048 | TaylorSeer | 0.1552 | 31.26 |
> |     | **+Ours** | **0.2564** | **31.31** |
>
> **Table.F**: Robustness under different frames (Hunyuan)
>
> | Resolution | Method | VBench(%)↑ |
> |:-----------:|:--------:|:------:|
> | 33  | TeaCache | 77.88 |
> |     | **+Ours** | **77.96** |
> | 49  | TeaCache | 77.81 |
> |     | **+Ours** | **78.01** |
> | 65  | TeaCache | 77.56 |
> |     | **+Ours** | **78.15** |
> | 129 | TeaCache | 76.21 |
> |     | **+Ours** | **77.31** |
>
> As shown in the tables above, on FLUX.1‑dev, CEM consistently improves the generation quality of the baseline (TaylorSeer) at the same acceleration efficiency, regardless of variations in seed, CFG, or resolution.
> Similarly, on Hunyuan, CEM enhances generation fidelity across different frame settings.
> Overall, the results demonstrate that the modeled error is highly **robust** across varying configurations, remaining effective **without re‑modeling**. This highlights the practicality and ease of deployment of CEM as a training‑free solution for real‑world applications.
>
> **A4  3). Error accumulation in long-horizon tasks.**
>
> In existing acceleration methods, errors tend to accumulate as the number of **timesteps** increases (e.g., DDPM more than DDIM).
> Note that these errors are **independent** of the generated content itself (e.g., long vs. short videos), since they are measured along the timestep dimension.
>
> Addressing this issue is one of the key motivations behind our method. Our CEM effectively mitigates error accumulation, improving the generation fidelity.
>
> Building on the established error, we employ dynamic programming to treat caching error as a cost. Given an acceleration budget, the optimal set of caching intervals (i.e., the optimal caching strategy) is derived to minimize cumulative error. This effectively mitigates the quality degradation caused by error accumulation in caching‑based acceleration.
>
> >**Q3**. Please provide more details about the offline modeling cost and complexity—for example, sample size, runtime, and scalability to larger models.
>
> **A5**. We provide an analysis of the overhead and complexity of the offline modeling process, along with a discussion of its scalability.
>
> **Summary for A5**: The following answer mainly includes:
> * 1). Costs of offline error modeling.
> * 2). Scalability of offline error modeling.

---

> ### Author Response · Authors · 2025-11-24
> **Official Comment by Authors (6/6)**
>
> **A5  1). Costs of offline error modeling.**
>
> We report the time and memory overhead of our offline error modeling for each generation model discussed in the paper. Regarding this overhead, we would like to make the following clarifications:
> * For each model, the offline error modeling needs to be performed **only once**. The modeled error can then be permanently reused across different configurations or **shared** by different acceleration methods.
> * During modeling, random content generation incurs inherent overhead (see “w/o OEM” in Tab. M). The **extra** cost beyond this reflects the true overhead of our offline error modeling, which records the sensitivity of random generations to acceleration.
> * The overhead of offline error modeling is not incurred during inference. Once modeling is completed, invoking CEM and applying the optimized caching strategy introduce only negligible runtime overhead.
>
> **Table.G**: Offline costs. OEM: offline error modeling.
>
> |  | |       Text2Image       |             |  |      Text2Video       |             | Class2Image |
> |:----------:|:--------------:|:-----------:|:-----------:|:---------------:|:-----------:|:-----------:|:----------------:|
> | Models | FLUX.1-dev | PixArt-$\alpha$ | SD15 | Hunyuan | Wan21 | OpenSora | DiT-XL/2 |
> | Time w/o. OEM | 1.92h | 13.52m | 38.88m | 4.72h | 2.73h | 3.63h | 19.63m |
> | Time w/. OEM | 2.08h | 14.71m | 43.83m | 5.21h | 4.15h | 4.65h | 25.52m |
> | Memory w/o. OEM | 43.42GB | 22.31GB | 3.65GB | 57.36GB | 40.83GB | 52.40GB | 4.09GB |
> | Memory w/. OEM | 53.06GB | 22.65GB | 3.67GB | 72.62GB | 63.46GB | 52.40GB | 4.65GB |
>
> The offline memory overhead primarily comes from storing intermediate features during inference to compute differences across cache intervals.
>
> On average, offline error modeling increases memory usage by only **15.8%** and modeling time by **16.8%** compared with random content generation, both remaining below 20%. Considering the substantial performance gains achieved by CEM and its zero inference‑time overhead, this cost is fully acceptable.
>
> **A5  2). Scalability of offline error modeling.**
>
> The overhead of offline error modeling is **influenced** by the model scale. The time overhead mainly depends on the model's inherent inference speed, which varies with its stochastic generation process, while our additional cost remains relatively small (averaging 16.8%). Similarly, the memory overhead is dominated by the model parameters themselves, with our caching and computation contributing only about 15.8% on average.
>
> Overall, larger models generally incur higher absolute overhead. However, two points should be noted:
> * Most of the overhead originates from the **model itself** rather than from our modeling process.
> * The relationship is **not strictly monotonic**, for example, SD15 is smaller than DiT-XL/2, yet its modeling time is longer.

---

> ### Comment · Reviewer_Hw5z · 2025-11-28
> **Response to Author Rebuttal**
>
> I thank the authors for the detailed and technically substantial rebuttal.
>
> Several of my original concerns are largely addressed:
>
> * The added analysis (Theorems, error bounds, and DP optimality/complexity) significantly strengthens the theoretical justification for the cumulative error approximation and the DP formulation.
> * The quantitative comparison between offline and online errors, as well as the reported error–fidelity correlations and robustness experiments (different seeds/CFG/resolutions/frames), clarify how cumulative error relates to generation quality in practice.
> * The detailed report on offline modeling time/memory overhead and its scalability makes the practical feasibility of CEM much clearer.
>
> Some points are partially addressed:
>
> * The rebuttal provides a reasonable argument that one-step diffusion and caching-based acceleration are distinct directions and clarifies that CEM is designed for iterative denoising; this clarifies scope but does not remove the limitation that the current method does not cover one-step or non-visual diffusion models.
> * Theoretical guarantees still rely on assumptions (e.g., Lipschitz continuity, i.i.d. error distribution), which is understandable but means some limitations remain.
>
> At this stage, I acknowledge that the rebuttal has addressed several of my main concerns, and I will take these clarifications into account during the discussion phase.

---

> > ### Author Response · Authors · 2025-11-29
> > **Official Comment by Authors**
> >
> > We are very glad to have addressed your questions! Thank you very much for your suggestions, as well as for your support of our work!
> > The core of our proof is to establish the error upper bound and the optimality of the dynamic programming formulation.
> > As a plug‑and‑play method, CEM requires a base model for one‑step diffusion, regrettably, no prior work has combined one‑step diffusion with caching‑based acceleration. Our approach provides training‑free acceleration for DiT while enhancing the fidelity of existing methods. We greatly appreciate your recognition of our efforts, and we respectfully hope that CEM will further advance DiT acceleration!

---

### Official Review · Reviewer_XEtZ · 2025-11-01

**Soundness:** 2
**Presentation:** 3
**Contribution:** 3
**Rating:** 6
**Confidence:** 2

**Summary:**

The paper targets training-free acceleration for Diffusion Transformer (DiT) models that already use cache-based methods such as ToCa, DuCa or TaylorSeer. The authors observe that these methods correct cache error (by pruning or prediction) but leave the cache schedule itself fixed or very simple, so a large part of the overall quality drop actually comes from a suboptimal schedule. They therefore propose CEM (Cumulative Error Minimization): first build, offline, a table that estimates the cache error for every pair of denoising timestep and cache interval; then, given an acceleration budget, run a dynamic-programming procedure to pick the sequence of cache/recompute steps that minimizes the accumulated error; finally, at inference time, just swap the original schedule for the optimized one, with zero extra runtime cost. They show that this plug-and-play schedule can be inserted into four existing accelerators and even quantized DiT models, giving better FID/IR/VBench and sometimes even surpassing the original unaccelerated model.

**Strengths:**

The motivation is clear and well aligned with current DiT practice: cumulative error in cache reuse is real, and current methods optimize everything except the schedule. The method stays training-free and keeps inference overhead unchanged because all error modeling is done offline. The DP formulation over timesteps and number of cache uses is simple, reproducible, and can target arbitrary acceleration budgets. The approach is model-agnostic and demonstrated on seven generators across text-to-image, text-to-video, and class-conditional DiT, and it improves four different cache-based accelerators plus a quantized DiT, which supports the “plug-and-play” claim. The paper also shows nice cases where accelerated+ours slightly outperforms the original model, which is a strong empirical signal that the schedule itself was the bottleneck.

**Weaknesses:**

The central assumption that an error table built from a small offline set can be reused for arbitrary prompts, CFG scales, resolutions, and even different video lengths is only illustrated on a fixed setting and not validated across harder regimes, so it is unclear how often the error prior must be rebuilt in practice. The cumulative-error approximation used in the DP is quite rough (essentially a cumsum of per-step errors) and the paper does not quantify how far this is from the true accumulated error on long denoising chains, where mismatches would matter most. The comparison to online, content-aware cache optimizers (AdaCache, AdaptiveDiffusion, TeaCache) is brief; these methods pay some runtime but are data-dependent, while the proposed method is data-agnostic, so the paper should spell out when the offline schedule is preferable. Many gains in the tables are modest (often 0.3–1 point) and look like “a better schedule on top of the same accelerator” rather than a fundamentally new acceleration mechanism; the offline profiling cost per model/task is also not clearly reported.

**Questions:**

1.How robust is the offline error prior to changes in prompt distribution, CFG/guidance strength, image resolution, or video length – do we need to rebuild the error table whenever the deployment setting shifts?
2.Can you quantify the gap between the cumulative-error approximation used in the DP and the true accumulated error on long sampling trajectories, especially for video models?
3.How does your offline schedule compare to a lightweight online/content-aware schedule under the same acceleration budget – is there a regime where online is clearly better?
4.What is the actual offline cost (time, number of sampled prompts/videos) to build one error table for a large DiT, and can a single table be shared across multiple accelerators that use different cache intervals?

---

> ### Author Response · Authors · 2025-11-24
> **Official Comment by Authors (1/6)**
>
> We are very grateful to the reviewer for the detailed and constructive feedback, which help us further improve our work!
>
> >**W1**. The central assumption that an error table built from a small offline set can be reused for arbitrary prompts, CFG scales, resolutions, and even different video lengths is only illustrated on a fixed setting and not validated across harder regimes, so it is unclear how often the error prior must be rebuilt in practice.
> **Q1**. How robust is the offline error prior to changes in prompt distribution, CFG/guidance strength, image resolution, or video length – do we need to rebuild the error table whenever the deployment setting shifts?
>
> **A1**.
> The essence of offline error modeling lies in revealing a generation model's sensitivity to different acceleration operations through a small number of random content generations.
> This sensitivity is **internal** and therefore **independent** of the content.
>
> **Summary for A1**: The following answer mainly includes:
> * 1). We analyzed the impact of using different data sources for offline error modeling on the generation fidelity.
> * 2). We analyzed the impact of the number of offline samples on generation fidelity.
> * 3). We analyzed the robustness of offline error modeling under different seeds, CFG, resolutions, and frames.
>
> The following section presents the detailed content of the response:
>
> **A1  1). The impact of the offline sample distribution on generation fidelity.**
>
> We perform offline error modeling using prompts from different sources, as shown in Tab. A, and obtain identical quality results on DrawBench with FLUX.1‑dev. This demonstrates that, although the prompt distributions differ (as indicated by the cosine distances in Tab. 7 and the t‑SNE visualization in Fig. 8 of Appendix B.2), such variations **do not affect** the offline error modeling process or the final generation results of CEM.
>
> **Table.A**: The impact of offline samples from different distributions on generation fidelity. OEM: offline error modeling. Cosine distance = 1 - Cosine similarity.
>
> | Dataset of OEM / Actual inference | Cosine distance | IR↑ | CLIP(G)↑ |
> |:--------------------------------:|:-----------------------:|:---:|:---------:|
> | Random from GPT / Drawbench | 0.841 | 0.9205 | 32.66 |
> | COCO2017 captions / Drawbench | 0.895 | 0.9205 | 32.66 |
> | Drawbench / Drawbench | 0.000 | 0.9205 | 32.66 |
>
> **A1  2). Impact of the number of offline samples on accelerated generation quality.**
>
> In previous experiments, we showed that offline error modeling (OEM) is intrinsic and independent of random sample content. To further explore modeling robustness, we evaluate the generation quality of caching strategies built from different sample numbers on DiT‑XL/2 (additional results on FLUX.1‑dev are provided in Appendix B.3, Tab. 8).
>
> **Table.B**: Impact of OEM random sample size on generation fidelity (FLUX.1-dev).
>
> | Sample Num | 10 | 50 | 100 | 200 | 300 | 500 | 1000 |
> | --- | --- | --- | --- | --- | --- | --- | --- |
> | Time of OEM | 31.71m | 1.05h | 2.08h | 4.15h | 6.27h | 10.38h | 20.77h |
> | IR↑ | 0.9202 | 0.9205 | 0.9205 | 0.9205 | 0.9205 | 0.9205 | 0.9205 |
> | CLIP(G)↑ | 32.62 | 32.66 | 32.66 | 32.66 | 32.66 | 32.66 | 32.66 |
>
> In FLUX.1-dev, we find that the generation fidelity remains almost **unaffected** by the number of samples used for modeling, and the results become stable once the sample size exceeds 10. This suggests that the modeled error reaches **convergence** beyond this point, leading to identical cache‑interval combinations in the dynamic caching strategy derived from it.
>
> **A1  3). Robustness of CEM.**
>
> We extend our analysis to evaluate its robustness under varying conditions.
> For each generative model, the caching error is modeled once and **reused** across all experimental settings to validate the stability of the offline error modeling. Specifically, we assess the effectiveness of CEM under different random seeds, CFG values, image resolutions (on FLUX.1‑dev), and frame counts (on Hunyuan).
>
> **Table.C**: Robustness under different seeds (FLUX.1-dev)
>
> | Seed | Method | IR↑ | CLIP(G)↑ |
> |:----:|:--------:|:----:|:------:|
> | 0 | TaylorSeer | 0.8760 | 32.17 |
> |   | **+Ours** | **0.9205** | **32.66** |
> | 42 | TaylorSeer | 0.8625 | 32.38 |
> |   | **+Ours** | **0.9405** | **32.86** |
> | 2025 | TaylorSeer | 0.8169 | 32.50 |
> |   | **+Ours** | **0.8875** | **32.69** |
> | 3407 | TaylorSeer | 0.8217 | 31.95 |
> |   | **+Ours** | **0.9118** | **32.99** |
>
> **Table.D**: Robustness under different CFGs (FLUX.1-dev)
>
> | CFG | Method | IR↑ | CLIP(G)↑ |
> |:---:|:--------:|:----:|:------:|
> | 3.5 | TaylorSeer | 0.8760 | 32.17 |
> |     | **+Ours** | **0.9205** | **32.66** |
> | 5.5 | TaylorSeer | 0.6571 | 31.34 |
> |     | **+Ours** | **0.8867** | **32.76** |
> | 7.5 | TaylorSeer | 0.6924 | 31.75 |
> |     | **+Ours** | **0.7336** | **32.29** |
> | 9.5 | TaylorSeer | 0.6794 | 31.20 |
> |     | **+Ours** | **0.7935** | **32.17** |

---

> ### Author Response · Authors · 2025-11-24
> **Official Comment by Authors (2/6)**
>
> **Table.E**: Robustness under different resolutions (FLUX.1-dev)
>
> | Resolution | Method | IR↑ | CLIP(G)↑ |
> |:-----------:|:--------:|:----:|:------:|
> | 256 | TaylorSeer | 0.5756 | 30.97 |
> |     | **+Ours** | **0.6792** | **31.40** |
> | 512 | TaylorSeer | 0.7495 | 31.80 |
> |     | **+Ours** | **0.7700** | **32.31** |
> | 1024 | TaylorSeer | 0.8760 | 32.17 |
> |     | **+Ours** | **0.9205** | **32.66** |
> | 2048 | TaylorSeer | 0.1552 | 31.26 |
> |     | **+Ours** | **0.2564** | **31.31** |
>
> **Table.F**: Robustness under different frames (Hunyuan)
> | Resolution | Method | VBench(%)↑ |
> |:-----------:|:--------:|:------:|
> | 33  | TeaCache | 77.88 |
> |     | **+Ours** | **77.96** |
> | 49  | TeaCache | 77.81 |
> |     | **+Ours** | **78.01** |
> | 65  | TeaCache | 77.56 |
> |     | **+Ours** | **78.15** |
> | 129 | TeaCache | 76.21 |
> |     | **+Ours** | **77.31** |
>
> As shown in the tables, on FLUX.1‑dev, CEM consistently improves the generation quality of the baseline (TaylorSeer) at the same acceleration efficiency, regardless of variations in seed, CFG, or resolution.
> Similarly, on Hunyuan, CEM enhances generation fidelity across different frame settings.
> Overall, the results demonstrate that the modeled error is **robust** across varying configurations, remaining **without re‑modeling**. This highlights the practicality and ease of deployment of CEM as a training‑free solution for real‑world applications.
>
> >**W2**. The cumulative-error approximation used in the DP is quite rough (essentially a cumsum of per-step errors) and the paper does not quantify how far this is from the true accumulated error on long denoising chains, where mismatches would matter most.
> **Q2**. Can you quantify the gap between the cumulative-error approximation used in the DP and the true accumulated error on long sampling trajectories, especially for video models?
>
> **A2.**
> The cumulative error approximation models the continuous accumulation of caching errors during accelerated denoising. In response to the reviewer’s question, we present additional analyses, including quantitative comparisons and theoretical justification, to demonstrate the validity and effectiveness of this design.
>
> **Summary for A2**: The following answer mainly includes:
> * 1). We conducted additional analysis and interpretation of the cumulative error.
> * 2). We further quantified the difference between the cumulative offline error and the actual online error.
> * 3). We additionally provided a theoretical explanation of the differences between the cumulative error and the actual error.
>
> The following section presents the detailed content of the response:
>
> **A2  1). Additional analysis on cumulative error approximation.**
>
> The cumulative error approximation builds on the offline error modeling to simulate the accumulation of caching errors during denoising. Based on this, our motivations for designing this module are as follows::
> * The offline error modeling ignores the influence of previously accumulated errors on the current timestep, as directly modeling cumulative errors would exponentially increase computational cost and be highly inefficient.
> * We approximate this process using a simple operation to avoid introducing complex computations that could compromise acceleration efficiency.
> * We conducted quantitative experiments comparing the cumulative caching error with the ground‑truth error and found that it effectively meets our design goal.
>
> **A2  2). Quantitative difference between offline error and online error.**
>
> Based on the above analysis, we conducted quantitative experiments to measure the difference between cumulative (offline) and actual (online) errors. We report the error gaps at key timesteps on FLUX.1‑dev and Hunyuan, with additional comparisons provided in the main paper (Fig. 3(c)) and Appendix C.2 (Fig. 9).
>
> **Table.G**: Offline error vs. online error on FLUX.1-dev with cache interval=5.
>
> | Timestep | 0     | 5     | 10    | 15    | 20    | 25    | 30    | 35    | 40    | 45    | 49    |
> |:---------:|:-----:|:-----:|:-----:|:-----:|:-----:|:-----:|:-----:|:-----:|:-----:|:-----:|:-----:|
> | Online error  | 0.000 | 0.010 | 0.292 | 0.425 | 0.566 | 0.671 | 0.745 | 0.822 | 0.917 | 1.039 | 1.221 |
> | Offline error | 0.000 | 0.010 | 0.289 | 0.433 | 0.559 | 0.666 | 0.745 | 0.823 | 0.902 | 1.023 | 1.231 |
>
> **Table.H**: Offline error vs. online error on Hunyuan with cache interval=5.
>
> | Timestep | 0     | 5     | 10    | 15    | 20    | 25    | 30    | 35    | 40    | 45    | 49    |
> |:---------:|:-----:|:-----:|:-----:|:-----:|:-----:|:-----:|:-----:|:-----:|:-----:|:-----:|:-----:|
> | Online error  | 0.000 | 0.009 | 0.383 | 0.581 | 0.778 | 0.956 | 1.085 | 1.257 | 1.494 | 1.784 | 2.248 |
> | Offline error | 0.000 | 0.008 | 0.363 | 0.551 | 0.748 | 0.934 | 1.102 | 1.272 | 1.490 | 1.747 | 2.183 |
>
> As presented in Tables, the final difference is only **0.80%** on FLUX.1‑dev and **2.89%** on Hunyuan, indicating that the offline error **closely approximates** the actual error.

---

> ### Author Response · Authors · 2025-11-24
> **Official Comment by Authors (3/6)**
>
> **A2  3). Additional theoretical analysis.**
>
> In addition to the experimental evaluation, we further provide a theoretical analysis of how the cumulative error approximates the real error.
>
> We obtained content-agnostic offline error $\mathcal{E}^*$ through offline error modeling and cumulative error approximation.
> We define the actual error under the same operation during formal inference as $\mathcal{E}$ (not the $\mathcal{E}$ defined in Eq.1), while the true propagated error of the cached latent during the denoising process is $\hat{\mathcal{E}}$.
>
> We assume the DiT is Lipschitz continuous (common in DiT for bounded error propagation), i.e., $\|\mathcal{D}(x, t) - \mathcal{D}(y, t)\| \leq L \|x - y\|$ for some Lipschitz constant $L > 0$.
>
> First, we establish the approximate relationship between the offline error $\mathcal{E}^\*$ and the online error $\mathcal{E}$.
> The $\mathcal{E}^\*$ is an empirical estimate of the error distribution using $N_s$ random samples, while $\mathcal{E}$ is computed for a specific inference sample.
>
> **Theorem.1**: *Under the unified error distribution, $|\mathcal{E}^\*(t, n) - \mathcal{E}(t, n)|$ is bounded by:*
>
> $|\mathcal{E}^\*(t, n) - \mathcal{E}(t, n)| \leq \sqrt{\frac{\log(2/\delta)}{2 N_s}} + \epsilon_{\text{var}}$,
>
> *where $\delta \in (0,1)$ is a confidence parameter, and $\epsilon_{\text{var}}$ is a small variance term (empirically small, as per Fig. 3(a)).*
>
> **Proof:**
> Since DiT content follows the same underlying distribution, we assume that the caching error in DiT also follows the unified distribution.
>
> Let $\mu(t,n)$ be the mean of the cosine error over all possible contents. $\mathcal{E}(t,n)$ for offline is the empirical mean $\hat{\mu}(t,n) = \frac{1}{N_s} \sum_i \mathcal{E}_i(t,n)$, and for online it's a single-sample estimate.
>
> By Hoeffding's inequality (cosine errors are in [0,1]), with probability at least $1 - \delta$:
>
> $|\hat{\mu}(t,n) - \mu(t,n)| \leq \sqrt{\frac{\log(2/\delta)}{2 N_s}}$.
>
> The $\mathcal{E}(t,n)$ deviates from $\mu(t,n)$ by at most the empirical variance $\epsilon_{\text{var}}$ (small, as per the analysis: low variance across contents and intervals).
>
> For cumulation: CUMSUM is a linear operator, the bound propagates:
>
> $|\mathcal{E}^\*(t,n) - \mathcal{E}(t,n)| \leq \sum_{\tau=1}^t |\mathcal{E}^\*(\tau,n) - \mathcal{E}(\tau,n)| \leq t \left( \sqrt{\frac{\log(2/\delta)}{2 N_s}} + \epsilon_{\text{var}} \right)$.
>
> For total timesteps $T$, the worst-case bound is $O(T / \sqrt{N_s})$, but empirically tighter due to low $\epsilon_{\text{var}}$.
> This holds because extended experiments (Appendix.B.2) show consistency across prompt sources, confirming the i.i.d. assumption.
> Thus, $\mathcal{E}^* \approx \mathcal{E}$ with high probability (Appendix.C.2 supports this).
>
> **Theorem.2**: *The difference $|\mathcal{E}(t,n) - \hat{\mathcal{E}}(t,n)|$ is bounded by:*
>
> $|\mathcal{E}(t,n) - \hat{\mathcal{E}}(t,n)| \leq L \cdot \sum_{\tau=1}^t \mathcal{E}(\tau,n) + \epsilon_{\text{prop}}$,
>
> *where $L$ is the Lipschitz constant of $\mathcal{D}$, and $\epsilon_{\text{prop}}$ is a small propagation residual.*
>
> **Proof:**
> In DiT, each timestep's output is $\mathcal{D}(x_t, t) = f(\mathcal{D}(x_{t+1}, t+1) + \delta)$, where $f$ is the transformer layer, and $\delta$ is noise/caching perturbation.
> Caching introduces error $\mathcal{E}(t,n)$ at each step, which propagates to future steps. The true propagated error $\hat{\mathcal{E}}(t,n)$ satisfies a recurrence: $\hat{\mathcal{E}}(t,n) = \mathcal{E}(t,n) + g(\hat{\mathcal{E}}(t+1,n))$, where $g$ models propagation (approximately linear due to DiT's residual connections and attention linearity).
> The CUMSUM approximation assumes $g \approx \text{Id}$ (identity, i.e., direct summation), which holds because of "high structural similarity between input and output" (as noted in TeaCache).
>
> By the Lipschitz assumption, propagation error is bounded: $|g(e) - e| \leq L \cdot e + \epsilon_{\text{prop}}$, where $\epsilon_{\text{prop}}$ captures non-linear residuals (small in DiT).
> Unrolling the recurrence over $t$ steps yields the bound via triangle inequality.
> For the caching strategy, the bound extends to the full sequence, as DP preserves substructure:
>
> $|\mathcal{E}(t,n) - \hat{\mathcal{E}}(t,n)| \leq  L \cdot \sum_{\tau=1}^t \mathcal{E}(\tau,n) + \epsilon_{\text{prop}}$.
>
> Thus, $\mathcal{E} \approx \hat{\mathcal{E}}$, with the bound tightening for smaller single-step errors.
>
> Finally, by chaining Theorems 1 and 2 (triangle inequality):
>
> $|\mathcal{E}^\*(t,n) - \hat{\mathcal{E}}(t,n)| \leq |\mathcal{E}^\*(t,n) - \mathcal{E}(t,n)| + |\mathcal{E}(t,n) - \hat{\mathcal{E}}(t,n)| \leq t \left( \sqrt{\frac{\log(2/\delta)}{2 N_s}} + \epsilon_{\text{var}} \right) + L \cdot \sum_{\tau=1}^t \mathcal{E}(\tau,n) + \epsilon_{\text{prop}}$.
>
> This upper bound is $O(T/\sqrt{N_s} + L \cdot \mathcal{E}_{\text{total}})$, where "total" is total single-step error. Empirically, it's small, confirming the approximation.

---

> ### Author Response · Authors · 2025-11-24
> **Official Comment by Authors (4/6)**
>
> >**W3**. The comparison to online, content-aware cache optimizers (AdaCache, AdaptiveDiffusion, TeaCache) is brief; these methods pay some runtime but are data-dependent, while the proposed method is data-agnostic, so the paper should spell out when the offline schedule is preferable.
> **Q3**. How does your offline schedule compare to a lightweight online/content-aware schedule under the same acceleration budget – is there a regime where online is clearly better?
>
> **A3.** We thank the reviewer for the insightful suggestion. Our CEM can be integrated as a plugin into online caching methods. Due to time constraints, we selected **TeaCache** as a representative framework to demonstrate performance improvements, as AdaptiveDiffusion targets U‑Net models and AdaCache provides code only for OpenSora.
>
> **Table.I**: The improvement of our CEM on TeaCache for FLUX.1-dev.
>
> |FLUX.1-dev 1024*1024| FLOPS(T)↓ | Spe↑ | Lat(s)↓ | Spe↑ | IR↑ | CLIP(G)(%)↑ | PSNR↑ | SSIM↑ | LPIPS↓ |
> |:--------:|:----------:|:----:|:---------:|:----:|:----:|:------------:|:------:|:------:|:------:|
> | Origin | 3719.50 | 1.00× | 35.63 | 1.00× | 0.9649 | 32.57 | INF | 1.00 | 0.00 |
> | 25% steps | 967.07 | 3.85× | 8.91 | 4.00× | 0.9310 | 32.72 | 14.71 | 0.58 | 0.46 |
> | TeaCache (l=0.6) | 1115.85 | 3.33× | 16.57 | 2.15× | 0.7228 | 30.66 | 17.41 | 0.70 | 0.35 |
> | **+Ours** | **1115.85** | **3.33×** | **16.05** | **2.22×** | **0.7362** | **31.13** | **17.89** | **0.71** | **0.33** |
> | TeaCache (l=0.8) | 892.68 | 4.17× | 12.33 | 2.89× | 0.7136 | 30.74 | 16.50 | 0.66 | 0.40 |
> | **+Ours** | **892.68** | **4.17×** | **12.08** | **2.95×** | **0.7139** | **30.77** | **16.96** | **0.67** | **0.39** |
>
> **Table.J**: The improvement of our CEM on TeaCache for Hunyuan.
>
> |Hunyuan 480P| FLOPS(T)↓ | Spe↑ | Lat(s)↓ | Spe↑ | VBench(%)↑ | PSNR↑ | SSIM↑ | LPIPS↓ |
> |:--------:|:----------:|:----:|:---------:|:----:|:-----------:|:------:|:------:|:------:|
> | Origin | 29773.00 | 1.00× | 441.76 | 1.00× | 78.46 | INF | 1.00 | 0.00 |
> | 25% steps | 7741.11 | 3.85× | 111.13 | 3.98× | 70.89 | 20.66 | 0.70 | 0.53 |
> | TeaCache (l=0.4) | 6550.06 | 4.55× | 108.54 | 4.07× | 77.56 | 19.58 | 0.68 | 0.37 |
> | **+Ours** | **6550.06** | **4.55×** | **105.94** | **4.17×** | **78.15** | **24.54** | **0.80** | **0.23** |
>
> As shown in the tables above, we evaluate the integration of our CEM with TeaCache on two major generative models: FLUX.1‑dev (text‑to‑image) and Hunyuan (text‑to‑video). More experiments are provided in the Appendix.D.2, Tab.13.
>
> For FLUX.1‑dev, CEM reduces the caching errors of TeaCache under the same acceleration level, thereby enhancing generation quality. Since CEM determines the optimal caching strategy via offline error modeling before batch generation, it also lowers inference latency. For instance, at l=0.4, CEM improves TeaCache's IR by **7.2%** (0.505) and CLIP score by **2%** (0.62), while reducing inference latency by **0.25s**, achieving a zero‑cost improvement in acceleration efficiency.
>
> For Hunyuan, at l=0.4, CEM raises TeaCache's overall VBench score from 77.56 to 78.15, closing **65%** of the gap to the original model (78.46). Meanwhile, inference latency decreases by **2.6s**, demonstrating simultaneous gains in both generation quality and acceleration efficiency.
>
> >**W4**. Many gains in the tables are modest (often 0.3-1 point) and look like "a better schedule on top of the same accelerator" rather than a fundamentally new acceleration mechanism.
>
> **A4**. We essentially propose a plug‑and‑play fidelity enhancement framework built upon existing acceleration methods.
>
> **Summary for A4**: The following answer mainly includes:
> * 1). We explained the degree of fidelity improvement of our CEM method for baselines.
> * 2). We introduced three additional evaluation metrics (PSNR, SSIM, LPIPS) and conducted experiments under higher acceleration efficiencies, where the performance improvement of CEM further increased.
>
> **A4  1). The improvement of our CEM on the existing evaluation metrics.**
>
> Our goal is to develop a plug‑and‑play solution that enhances the fidelity of existing acceleration methods. As shown in Tabs. 1-3, our CEM improves the fidelity of FasterSD, DuCa, ToCa and TaylorSeer across SD15, PixArt-$\alpha$, FLUX.1-dev and Hunyuan, surpassing the original models. The relatively modest gains in certain metrics can be attributed to the following factors:
> * Metrics like CLIP, P and R are inherently **insensitive**. For instance, FasterSD multiply CLIP by 2.5 to amplify differences.
> * Baselines generally report best trade‑off between speed and fidelity in their papers, leaving limited room for further improvement. Our CEM delivers more improvements under **higher acceleration** settings.
>
> Therefore, we introduce three additional quality metrics (PSNR, SSIM, LPIPS) and conduct experiments under a wider range of acceleration ratios.

---

> ### Author Response · Authors · 2025-11-24
> **Official Comment by Authors (5/6)**
>
> **A4  2). Additional results of our CEM on new metrics and higher acceleration ratios.**
>
> We evaluate various settings and introduce three quality metrics **(PSNR, SSIM, LPIPS)**. We present the main results here to demonstrate the fidelity improvements, with the complete results provided in the paper Appendix D.2 (Tab.13 and Tab.14).
>
> **Table.K**: Quantitative comparison on PixArt-$\alpha$.
>
> |PixArt-$\alpha$ 256*256| FLOPS(T)↓ | Spe↑ | Lat(s)↓ | Spe↑ |  FID↓  | CLIP(L)↑ | PSNR↑ | SSIM↑ | LPIPS↓ |
> |:-----------------:|:---------:|:----:|:--------:|:----:|:------:|:------------:|:-----:|:-----:|:------:|
> | Origin            | 11.18     | 1.00× | 0.86     | 1.00× | 28.06 | 16.29 | INF  | 1.00  | 0.00  |
> | 50% steps         | 5.59      | 2.00× | 0.43     | 2.00× | 37.41 | 15.82 | 18.67 | 0.70  | 0.20  |
> | DuCa(N=4)         | 5.90      | 1.89× | 0.49     | 1.76× | 35.36 | 16.45 | 15.99 | 0.52  | 0.35  |
> | **+Ours** | **5.94** | **1.88×** | **0.49** | **1.76×** | **27.20** | **16.42** | **20.94** | **0.78** | **0.14** |
> | DuCa(N=5)         | 4.79      | 2.33× | 0.40     | 2.15× | 41.56 | 16.46 | 14.96 | 0.46  | 0.42  |
> | **+Ours** | **4.75** | **2.35×** | **0.39** | **2.20×** | **27.57** | **16.37** | **18.25** | **0.68** | **0.21** |
>
> In PixArt‑$\alpha$, as acceleration efficiency increases, DuCa exhibits a noticeable drop in fidelity. In contrast, our CEM consistently enhances fidelity while delivering larger performance gains. The three introduced quality metrics further highlight these improvements, for example, in the last row, CEM reduces FID↓ by 13.99 (**33.7%**) and achieves nearly **40%** improvement in PSNR↑, SSIM↑, and LPIPS↓.
>
> **Table.L**: Quantitative comparison on FLUX.1-dev.
>
> |FLUX.1-dev 1024*1024| FLOPS(T)↓ | Spe↑ | Lat(s)↓ | Spe↑ | IR↑ | CLIP(G)↑ | PSNR↑ | SSIM↑ | LPIPS↓ |
> |:----------------------:|:-----------:|:----:|:---------:|:----:|:------:|:--------------:|:------:|:------:|:------:|
> | Origin | 3719.50 | 1.00× | 35.63 | 1.00× | 0.9649 | 32.57 | INF | 1.00 | 0.00 |
> | 25% steps | 967.07 | 3.85× | 8.91 | 4.00× | 0.9310 | 32.72 | 14.71 | 0.58 | 0.46 |
> | TaylorSeer(N7O1) | 668.97 | 5.56× | 8.61 | 4.14× | 0.9233 | 32.55 | 14.94 | 0.57 | 0.46 |
> | **+Ours** | **668.97** | **5.56×** | **8.59** | **4.15×** | **0.9449** | **32.59** | **15.71** | **0.58** | **0.42** |
> | TaylorSeer(N8O1) | 595.12 | 6.25× | 7.41 | 4.81× | 0.8760 | 32.17 | 14.24 | 0.54 | 0.49 |
> | **+Ours** | **595.12** | **6.25×** | **7.41** | **4.81×** | **0.9205** | **32.66** | **15.41** | **0.56** | **0.46** |
>
> In the FLUX.1-dev, our CEM likewise demonstrates more pronounced fidelity improvements under higher acceleration efficiencies. At 5.56× acceleration, our CEM improves IR by 0.216 over the TaylorSeer, and at 6.25× acceleration, CEM achieves an IR↑ improvement of 0.445.
>
> We argue that the degree of fidelity improvement is fundamentally constrained by the upper bound of reducible caching errors in each acceleration method. As acceleration efficiency increases, larger errors are introduced, giving CEM greater potential for improvement. By employing dynamic programming to optimize caching intervals, CEM reduces baseline caching errors and mitigates fidelity degradation.
>
> Finally, we further report the performance of our CEM on the online acceleration method **TeaCache**, as well as additional results on the **Wan2.1** video generation model. Detailed results are provided in the revised paper, Sec. 4.2 (Tabs. 1 and 2) and Appendix D.2 (Tab. 13).
>
> >**W5**. The offline profiling cost per model/task is also not clearly reported.
> **Q4**. What is the actual offline cost (time, number of sampled prompts/videos) to build one error table for a large DiT, and can a single table be shared across multiple accelerators that use different cache intervals?
>
> **A5.** We sincerely appreciate the reviewer for pointing out this omission. Since this part does not affect the actual inference of generation, it was not included in the previous version. We have now provided an additional explanation for this part and incorporated it into the revised paper in Sec.4.3 and Appendix B.4.
>
> We report the time and memory overhead of our offline error modeling for each generation model discussed in the paper. Regarding this overhead, we would like to make the following clarifications:
> * For each model, the offline error modeling needs to be performed **only once**. The modeled error can then be permanently **reused** across different configurations or **shared** by different acceleration methods.
> * During modeling, random content generation incurs inherent overhead (see “w/o OEM” in Tab. M). The **extra** cost beyond this reflects the true overhead of our offline error modeling, which records the sensitivity of random generations to acceleration.
> * The overhead of offline error modeling is not incurred during inference. Once modeling is completed, invoking CEM and applying the optimized caching strategy introduce only **negligible** runtime overhead.

---

> ### Author Response · Authors · 2025-11-24
> **Official Comment by Authors (6/6)**
>
> **Table.M**: Offline cost of the OEM module.
>
> |  | |       Text2Image       |             |  |      Text2Video       |             | Class2Image |
> |:----------:|:--------------:|:-----------:|:-----------:|:---------------:|:-----------:|:-----------:|:----------------:|
> | Models | FLUX.1-dev | PixArt-$\alpha$ | SD15 | Hunyuan | Wan21 | OpenSora | DiT-XL/2 |
> | Time w/o. OEM | 1.92h | 13.52m | 38.88m | 4.72h | 2.73h | 3.63h | 19.63m |
> | Time w/. OEM | 2.08h | 14.71m | 43.83m | 5.21h | 4.15h | 4.65h | 25.52m |
> | Memory w/o. OEM | 43.42GB | 22.31GB | 3.65GB | 57.36GB | 40.83GB | 52.40GB | 4.09GB |
> | Memory w/. OEM | 53.06GB | 22.65GB | 3.67GB | 72.62GB | 63.46GB | 52.40GB | 4.65GB |
>
> The additional memory overhead primarily comes from storing intermediate features during inference to compute differences across cache intervals.
>
> On average, offline error modeling increases memory usage by only **15.8%** and modeling time by **16.8%** compared with random content generation, both remaining below 20%. Considering the substantial performance gains achieved by CEM and its zero inference‑time overhead, this cost is fully acceptable.

---

### Official Review · Reviewer_WEPH · 2025-11-01

**Soundness:** 3
**Presentation:** 3
**Contribution:** 3
**Rating:** 4
**Confidence:** 4

**Summary:**

This paper proposes CEM (Cumulative Error Minimization), a plug-and-play, training-free acceleration strategy for Diffusion Transformers (DiTs). CEM models caching error as a joint function of denoising timestep and cache interval, and then uses a dynamic-programming procedure to choose a caching schedule that minimizes cumulative error under a given acceleration budget. The method is model-agnostic, designed to be compatible with existing cache-correction pipelines and quantized models, and claims no extra runtime overhead (beyond an offline estimation step). Experiments across seven generative models and three tasks suggest that CEM can improve fidelity over existing acceleration methods and, in some cases, match or slightly exceed the original unaccelerated models.

**Strengths:**

1. Clear, well-structured presentation: The problem setup, motivation, and algorithm are easy to follow. The paper is readable and self-contained.

2. Interesting angle via offline error modeling: Estimating an error distribution over (timestep, cache interval) and optimizing a schedule with DP is a neat, generally applicable idea.

3. Plug-and-play applicability: The method is model-agnostic and integrates with existing cache-correction approaches and quantized variants, which increases potential practical value.

4. Budget-aware optimization: Framing the schedule search under explicit acceleration budgets is sensible and aligns with deployment needs.

**Weaknesses:**

1. Limited practical gains in several settings: In Table 1 and Table 3 (and a few other results), improvements over strong baselines appear marginal, making it difficult to judge the real-world significance of CEM. In places where the paper claims to “even outperform the original,” the margins seem small or inconsistent.

2. Representativeness of the offline estimate is unclear: The fidelity of the learned error prior depends on the sample set used for estimation. If the sample pool (prompts, content types, seeds) is not representative, the optimized schedule may not generalize, which could explain the limited improvements in several cells.

3. Overhead vs. ‘no extra computation’ claim: While CEM adds no inference overhead, the offline estimation step has real cost. The paper does not quantify this cost or show its amortization across models/datasets/budgets.

4. Robustness and stability not fully characterized: The sensitivity to the number of estimation samples, dataset/domain shifts, prompt distributions, and random seeds is not sufficiently explored.

**Questions:**

Several improvements are small. Could this indicate that random-sample–based estimation lacks coverage (e.g., prompt types, scene complexity, motion patterns for video)?

---

> ### Author Response · Authors · 2025-11-24
> **Official Comment by Authors (1/5)**
>
> We sincerely thank you for your positive feedback and valuable suggestions, which greatly help us improve this work!
>
> >**W1**. Limited practical gains in several settings: In Table 1 and Table 3 (and a few other results), improvements over strong baselines appear marginal, making it difficult to judge the real-world significance of CEM. In places where the paper claims to “even outperform the original,” the margins seem small or inconsistent.
> **Q1**. Several improvements are small. Could this indicate that random-sample–based estimation lacks coverage (e.g., prompt types, scene complexity, motion patterns for video)?
>
> **A1**.
> Since both W1 and Q1 relate to issues concerning model performance improvement, we provide a joint response to them below.
>
> **Summary for A1**: The following answer mainly includes:
> * 1). We explained the degree of fidelity improvement of our CEM method for baselines.
> * 2). We introduced three additional evaluation metrics (PSNR, SSIM, LPIPS) and conducted experiments under higher acceleration efficiencies, where the performance improvement of CEM further increased.
> * 3). We specify that the fidelity surpasses the original models particularly on the SD15, PixArt-$\alpha$, FLUX.1-dev, and Hunyuan models to avoid ambiguity.
> * 4). We analyzed the reasons behind the performance improvement and found that it shows no evident correlation with the representativeness of the offline samples.
>
> The following section presents the detailed content of the response:
>
> **A1  1). The improvement of our CEM on the existing evaluation metrics.**
>
> Our goal is to develop a plug‑and‑play solution that enhances the fidelity of existing acceleration methods.
>
> As shown in Tabs. 1-3, our CEM improves the fidelity of FasterSD, DuCa, ToCa and TaylorSeer across SD15, PixArt-$\alpha$, FLUX.1-dev and Hunyuan, surpassing the original models. The relatively modest gains in certain metrics can be attributed to the following factors:
> * Metrics like CLIP, P and R are inherently **insensitive**. For instance, FasterSD multiply CLIP by 2.5 to amplify differences.
> * Baselines generally report best trade‑off between speed and fidelity in their papers, leaving limited room for further improvement. Our CEM delivers more improvements under **higher acceleration** settings.
>
> Therefore, we introduce three additional quality metrics (PSNR, SSIM, LPIPS) and conduct experiments under a wider range of acceleration ratios.
>
> **A1  2). Additional results of our CEM on new metrics and higher acceleration ratios.**
>
> We evaluate various acceleration settings and additionally introduce three quality metrics **(PSNR, SSIM, LPIPS)**. We present the main results here to demonstrate the fidelity improvements, with the complete results provided in the paper Appendix D.2 (Tab.13 and Tab.14).
>
> **Table.A**: Quantitative comparison on PixArt-$\alpha$.
>
> |PixArt-$\alpha$ 256*256| FLOPS(T)↓ | Spe↑ | Lat(s)↓ | Spe↑ |  FID↓  | CLIP(L)↑ | PSNR↑ | SSIM↑ | LPIPS↓ |
> |:-----------------:|:---------:|:----:|:--------:|:----:|:------:|:------------:|:-----:|:-----:|:------:|
> | Origin            | 11.18     | 1.00× | 0.86     | 1.00× | 28.06 | 16.29 | INF  | 1.00  | 0.00  |
> | 50% steps         | 5.59      | 2.00× | 0.43     | 2.00× | 37.41 | 15.82 | 18.67 | 0.70  | 0.20  |
> | DuCa(N=4)         | 5.90      | 1.89× | 0.49     | 1.76× | 35.36 | 16.45 | 15.99 | 0.52  | 0.35  |
> | **+Ours** | **5.94** | **1.88×** | **0.49** | **1.76×** | **27.20** | **16.42** | **20.94** | **0.78** | **0.14** |
> | DuCa(N=5)         | 4.79      | 2.33× | 0.40     | 2.15× | 41.56 | 16.46 | 14.96 | 0.46  | 0.42  |
> | **+Ours** | **4.75** | **2.35×** | **0.39** | **2.20×** | **27.57** | **16.37** | **18.25** | **0.68** | **0.21** |
>
> In PixArt‑$\alpha$, as acceleration efficiency increases, DuCa exhibits a noticeable drop in generation fidelity. In contrast, our CEM consistently enhances fidelity while delivering substantially larger performance gains. The three newly introduced quality metrics further highlight these improvements, for example, in the last row, CEM reduces FID↓ by 13.99 (**33.7%**) and achieves nearly **40%** improvement in PSNR↑, SSIM↑, and LPIPS↓.

---

> ### Author Response · Authors · 2025-11-24
> **Official Comment by Authors (2/5)**
>
> **Table.B**: Quantitative comparison on FLUX.1-dev.
>
> |FLUX.1-dev 1024*1024| FLOPS(T)↓ | Spe↑ | Lat(s)↓ | Spe↑ | IR↑ | CLIP(G)↑ | PSNR↑ | SSIM↑ | LPIPS↓ |
> |:----------------------:|:-----------:|:----:|:---------:|:----:|:------:|:--------------:|:------:|:------:|:------:|
> | Origin | 3719.50 | 1.00× | 35.63 | 1.00× | 0.9649 | 32.57 | INF | 1.00 | 0.00 |
> | 25% steps | 967.07 | 3.85× | 8.91 | 4.00× | 0.9310 | 32.72 | 14.71 | 0.58 | 0.46 |
> | TaylorSeer(N7O1) | 668.97 | 5.56× | 8.61 | 4.14× | 0.9233 | 32.55 | 14.94 | 0.57 | 0.46 |
> | **+Ours** | **668.97** | **5.56×** | **8.59** | **4.15×** | **0.9449** | **32.59** | **15.71** | **0.58** | **0.42** |
> | TaylorSeer(N8O1) | 595.12 | 6.25× | 7.41 | 4.81× | 0.8760 | 32.17 | 14.24 | 0.54 | 0.49 |
> | **+Ours** | **595.12** | **6.25×** | **7.41** | **4.81×** | **0.9205** | **32.66** | **15.41** | **0.56** | **0.46** |
>
> In the FLUX.1-dev, our CEM likewise demonstrates more pronounced fidelity improvements under higher acceleration efficiencies. At 5.56× acceleration, our CEM improves IR by 0.216 over the TaylorSeer, and at 6.25× acceleration, CEM achieves an IR↑ improvement of 0.445.
>
> We argue that the degree of fidelity improvement is fundamentally constrained by the upper bound of reducible caching errors in each acceleration method. As acceleration efficiency increases, larger errors are introduced, giving CEM greater potential for improvement. By employing dynamic programming to optimize caching intervals, CEM effectively reduces baseline caching errors and mitigates fidelity degradation.
>
> Finally, we further report the performance of our CEM on the online acceleration method **TeaCache**, as well as additional results on the **Wan2.1** video generation model. Detailed results are provided in the revised paper, Sec. 4.2 (Tab. 1 and 2) and Appendix D.2 (Tab. 13).
>
> **A1  3). Refinement of the expression "performance outperforms the original model".**
>
> The statement that our CEM "outperforms the original model" primarily refers to its improvements over accelerated baselines across SD15, PixArt, FLUX, and Hunyuan, as reported in Tabs. 1 and 2. For instance, on SD15, FasterSD + CEM achieves an FID↓ of 19.99, surpassing the original model's 21.15. On PixArt‑$\alpha$, DuCa + CEM obtains an FID↓ of 27.06, outperforming the original 28.10. On FLUX.1‑dev, ToCa + CEM reaches an IR↑ of 1.0018 versus 0.9649, and on Hunyuan, TaylorSeer + CEM achieves a VBench↑ of 81.24 compared to 80.66, collectively confirming consistent fidelity improvements.
>
> Thank you to the reviewer for this correction. We agree that the words should be **more precise**, emphasizing that CEM only achieves higher generation quality than the original models (**SD15, PixArt, Flux, and Hunyuan**) under acceleration. We have revised this statement accordingly in the paper.
>
> **A1  4). Reasons and essence of performance improvement.**
>
> The core idea of offline error modeling is to quantify the generation model **sensitivity** to various acceleration operations, this is an **inherent and intrinsic** characteristic that remains **independent** of the generated content.
>
> As shown in Fig.1 in paper, the improvement in generation fidelity stems from our effective reduction of caching error, thereby mitigating the impact of acceleration during denoising process.
>
> We emphasize that the generation quality improvements brought by CEM are **not directly determined** by the representativeness (as further discussed in our responses to W2 and W4) or coverage of the offline samples, but rather depends on the inherent room for improvement within the acceleration baselines themselves.
>
> As shown in the table below, our offline error modeling is **shared** across different configurations and acceleration methods within the same generation model. However, the fidelity improvements vary across baselines and acceleration levels. This indicates that the performance gains primarily stem from the **characteristics** and **efficiency** of the acceleration methods rather than from the offline samples themselves.

---

> ### Author Response · Authors · 2025-11-24
> **Official Comment by Authors (3/5)**
>
> **Table.C**: Quantitative comparison on DiT-XL/2-DDIM.
>
> |DiT-XL/2-DDIM 256*256| FLOPS(T)↓ | Spe↑ | Lat(s)↓ | Spe↑ | FID↓ | sFID↓ | IS↑ | P↑ | R↑ | PSNR↑ | SSIM↑ | LPIPS↓ |
> |:----------------------:|:---------:|:----:|:--------:|:----:|:----:|:----:|:----:|:----:|:----:|:----:|:----:|:----:|
> | Origin | 23.74 | 1.00× | 0.53 | 1.00× | 2.25 | 4.33 | 239.93 | 0.80 | 0.59 | INF | 1.00 | 0.00 |
> | ToCa(N=5) | 7.44 | 3.19× | 0.20 | 2.65× | 6.37 | 7.09 | 199.48 | 0.74 | 0.53 | 16.56 | 0.53 | 0.40 |
> | **+Ours** | **7.14** | **3.32×** | **0.18** | **2.94×** | **4.68** | **6.41** | **212.13** | **0.77** | **0.55** | **21.59** | **0.72** | **0.20** |
> | ToCa(N=6) | 7.02 | 3.38× | 0.18 | 2.94× | 6.79 | 7.41 | 187.32 | 0.72 | 0.55 | 17.62 | 0.56 | 0.36 |
> | **+Ours** | **6.72** | **3.53×** | **0.17** | **3.12×** | **5.38** | **6.84** | **205.52** | **0.76** | **0.55** | **20.83** | **0.69** | **0.23** |
> | DuCa(N=5) | 6.32 | 3.76× | 0.17 | 3.12× | 6.07 | 6.64 | 199.64 | 0.74 | 0.52 | 16.63 | 0.53 | 0.39 |
> | **+Ours** | **6.73** | **3.53×** | **0.17** | **3.12×** | **3.96** | **5.87** | **218.66** | **0.78** | **0.55** | **23.00** | **0.76** | **0.16** |
> | DuCa(N=6) | 5.86 | 4.05× | 0.15 | 3.53× | 6.38 | 6.65 | 189.97 | 0.73 | 0.54 | 17.47 | 0.54 | 0.37 |
> | **+Ours** | **5.69** | **4.17×** | **0.14** | **3.79×** | **5.06** | **6.75** | **206.03** | **0.77** | **0.54** | **21.62** | **0.70** | **0.21** |
> | TaylorSeer(N4O4) | 6.66 | 3.56× | 0.27 | 1.96× | 2.49 | 5.19 | 235.83 | 0.80 | 0.59 | 30.74 | 0.93 | 0.04 |
> | **+Ours** | **6.66** | **3.56×** | **0.27** | **1.96×** | **2.46** | **4.80** | **238.28** | **0.80** | **0.59** | **31.76** | **0.94** | **0.03** |
> | TaylorSeer(N6O1) | 4.76 | 4.99× | 0.14 | 3.79× | 3.56 | 7.52 | 223.83 | 0.79 | 0.56 | 24.69 | 0.80 | 0.13 |
> | **+Ours** | **4.76** | **4.99×** | **0.13** | **4.08×** | **3.08** | **6.43** | **231.10** | **0.80** | **0.57** | **25.64** | **0.83** | **0.10** |
>
> >**W2**. Representativeness of the offline estimate is unclear: The fidelity of the learned error prior depends on the sample set used for estimation. If the sample pool (prompts, content types, seeds) is not representative, the optimized schedule may not generalize, which could explain the limited improvements in several cells.
> **W4**. Robustness and stability not fully characterized: The sensitivity to the number of estimation samples, dataset/domain shifts, prompt distributions, and random seeds is not sufficiently explored.
>
> **A2**.
> We analyze the relationship between the representativeness of offline samples and the robustness of CEM, and we address W2 and W4 together to highlight their differences and respective focuses.
>
> **Summary for A2**: The following answer mainly includes:
> * 1). We explained that the performance of offline modeling is robust to the representativeness or distribution of offline samples.
> * 2). We analyzed the impact of the number of offline samples on the fidelity of our CEM method.
> * 3). We analyzed the robustness of our CEM improvements under different seeds, CFG, resolutions, and frames.
>
> The following section presents the detailed content of the response:
>
> **A2  1). The representativeness of offline samples vs. the robustness of our CEM.**
>
> The offline error modeling characterizes the generation model's intrinsic sensitivity to various acceleration operations, as revealed through a limited set of sample generation experiments.
> This sensitivity is **intrinsic and content-agnostic**. Therefore, the prompts involved in the modeling process do not significantly affect the modeling results.
>
> To illustrate this point, we construct modeling prompt sets from three distinct data sources (random prompts generated by GPT, captions from COCO2017, and prompts from DrawBench) and subsequently evaluate the generation quality after acceleration under the DrawBench.
> In the table below, we quantify the distributional differences among the three sources using Cosine distance with CLIP, and also include a t‑SNE visualization of the three datasets in the Appendix B.2 (Fig.8, Tab.7), which clearly shows significant **distribution variation** among them. Nevertheless, the offline error derived from these three sources resulted in the **same** quality evaluation results on DrawBench.
> This result confirms our previous statement that the error modeling process is content‑agnostic.
>
> **Table.D**: The impact of offline samples from different distributions on generation fidelity. OEM: offline error modeling. Cosine distance = 1 - Cosine similarity.
>
> | Dataset of OEM / Actual inference | Cosine distance | IR↑ | CLIP(G)↑ |
> |:--------------------------------:|:-----------------------:|:---:|:---------:|
> | Random from GPT / Drawbench | 0.841 | 0.9205 | 32.66 |
> | COCO2017 captions / Drawbench | 0.895 | 0.9205 | 32.66 |
> | Drawbench / Drawbench | 0.000 | 0.9205 | 32.66 |

---

> ### Author Response · Authors · 2025-11-24
> **Official Comment by Authors (4/5)**
>
> **A2  2). Impact of the number of offline samples on accelerated generation quality.**
>
> In our previous experiments, we demonstrated that offline error modeling (OEM) is an intrinsic property independent of the specific content of random samples. To further examine the impact of sample variability, we evaluate the generation quality of caching strategies constructed from different sample numbers on FLUX.1‑dev (additional results on DiT-XL/2 are provided in Appendix B.3, Tab. 8).
>
> **Table.E**: Impact of OEM random sample size on generation fidelity (FLUX.1-dev).
>
> | Sample Num | 10 | 50 | 100 | 200 | 300 | 500 | 1000 |
> | --- | --- | --- | --- | --- | --- | --- | --- |
> | Time of OEM | 31.71m | 1.05h | 2.08h | 4.15h | 6.27h | 10.38h | 20.77h |
> | IR↑ | 0.9202 | 0.9205 | 0.9205 | 0.9205 | 0.9205 | 0.9205 | 0.9205 |
> | CLIP(G)↑ | 32.62 | 32.66 | 32.66 | 32.66 | 32.66 | 32.66 | 32.66 |
>
> In FLUX.1-dev, we find that the generation fidelity remains almost **unaffected** by the number of samples used for modeling, and the results become stable once the sample size exceeds 10. This suggests that the modeled error reaches **convergence** beyond this point, leading to identical cache‑interval combinations in the dynamic caching strategy derived from it.
>
> **A2  3). Robustness of CEM.**
>
> We extend our analysis to evaluate its robustness under varying conditions.
> For each generative model, the caching error is modeled once and **reused** across all experimental settings to validate the stability of the offline error modeling. Specifically, we assess the effectiveness of CEM under different random seeds, CFG values, image resolutions (on FLUX.1‑dev), and frame counts (on Hunyuan).
>
> **Table.F**: Robustness under different seeds (FLUX.1-dev)
>
> | Seed | Method | IR↑ | CLIP(G)↑ |
> |:----:|:--------:|:----:|:------:|
> | 0 | TaylorSeer | 0.8760 | 32.17 |
> |   | **+Ours** | **0.9205** | **32.66** |
> | 42 | TaylorSeer | 0.8625 | 32.38 |
> |   | **+Ours** | **0.9405** | **32.86** |
> | 2025 | TaylorSeer | 0.8169 | 32.50 |
> |   | **+Ours** | **0.8875** | **32.69** |
> | 3407 | TaylorSeer | 0.8217 | 31.95 |
> |   | **+Ours** | **0.9118** | **32.99** |
>
> **Table.G**: Robustness under different CFGs (FLUX.1-dev)
>
> | CFG | Method | IR↑ | CLIP(G)↑ |
> |:---:|:--------:|:----:|:------:|
> | 3.5 | TaylorSeer | 0.8760 | 32.17 |
> |     | **+Ours** | **0.9205** | **32.66** |
> | 5.5 | TaylorSeer | 0.6571 | 31.34 |
> |     | **+Ours** | **0.8867** | **32.76** |
> | 7.5 | TaylorSeer | 0.6924 | 31.75 |
> |     | **+Ours** | **0.7336** | **32.29** |
> | 9.5 | TaylorSeer | 0.6794 | 31.20 |
> |     | **+Ours** | **0.7935** | **32.17** |
>
> **Table.H**: Robustness under different resolutions (FLUX.1-dev)
>
> | Resolution | Method | IR↑ | CLIP(G)↑ |
> |:-----------:|:--------:|:----:|:------:|
> | 256 | TaylorSeer | 0.5756 | 30.97 |
> |     | **+Ours** | **0.6792** | **31.40** |
> | 512 | TaylorSeer | 0.7495 | 31.80 |
> |     | **+Ours** | **0.7700** | **32.31** |
> | 1024 | TaylorSeer | 0.8760 | 32.17 |
> |     | **+Ours** | **0.9205** | **32.66** |
> | 2048 | TaylorSeer | 0.1552 | 31.26 |
> |     | **+Ours** | **0.2564** | **31.31** |
>
> **Table.I**: Robustness under different frames (Hunyuan)
>
> | Resolution | Method | VBench(%)↑ |
> |:-----------:|:--------:|:------:|
> | 33  | TeaCache | 77.88 |
> |     | **+Ours** | **77.96** |
> | 49  | TeaCache | 77.81 |
> |     | **+Ours** | **78.01** |
> | 65  | TeaCache | 77.56 |
> |     | **+Ours** | **78.15** |
> | 129 | TeaCache | 76.21 |
> |     | **+Ours** | **77.31** |
>
> As shown in the tables above, on FLUX.1‑dev, CEM consistently improves the generation quality of the baseline (TaylorSeer) under identical acceleration efficiency, regardless of variations in seed, CFG, or resolution.
> Similarly, on Hunyuan, CEM enhances generation fidelity across different frame settings.
> Overall, the results demonstrate that the modeled error remains **highly robust** across diverse configurations, retaining effectiveness **without** the need for re‑modeling. This highlights the practicality and ease of deployment of CEM as a training‑free solution for real‑world applications.
>
> >**W3**. Overhead vs. "no extra computation" claim: While CEM adds no inference overhead, the offline estimation step has real cost. The paper does not quantify this cost or show its amortization across models/datasets/budgets.
>
> **A3**.
> We sincerely appreciate the reviewer for pointing out this omission. Since this part does not affect the actual inference of generation, it was not included in the previous version. We have now provided an additional explanation for this part and incorporated it into the revised paper in Sec.4.3 and Appendix B.4.
>
> **Summary for A3**: The following answer mainly includes:
> * 1). We analyzed the computational cost associated with the offline error modeling process.
> * 2). We analyzed the computational cost of the dynamic programming solution during practical inference.

---

> ### Author Response · Authors · 2025-11-24
> **Official Comment by Authors (5/5)**
>
> **A3  1). Computational and memory overhead of offline error modeling for different models.**
>
> We report the time and memory overhead of our offline error modeling for each generation model discussed in the paper. Regarding this overhead, we would like to make the following clarifications:
> * For each model, the offline error modeling needs to be performed **only once**. The modeled error can then be permanently reused across different configurations.
> * During modeling, random content generation incurs inherent overhead (see “w/o OEM” in Tab. J). The **extra** cost beyond this reflects the true overhead of our offline error modeling, which records the sensitivity of random generations to acceleration.
> * The overhead of offline error modeling is not incurred during inference. Once modeling is completed, invoking CEM and applying the optimized caching strategy introduce only **negligible** runtime overhead (see Tab. K for online overhead).
>
> **Table.J**: Offline cost of the OEM module.
>
> |  | |       Text2Image       |             |  |      Text2Video       |             | Class2Image |
> |:----------:|:--------------:|:-----------:|:-----------:|:---------------:|:-----------:|:-----------:|:----------------:|
> | Models | FLUX.1-dev | PixArt-$\alpha$ | SD15 | Hunyuan | Wan21 | OpenSora | DiT-XL/2 |
> | Time w/o. OEM | 1.92h | 13.52m | 38.88m | 4.72h | 2.73h | 3.63h | 19.63m |
> | Time w/. OEM | 2.08h | 14.71m | 43.83m | 5.21h | 4.15h | 4.65h | 25.52m |
> | Memory w/o. OEM | 43.42GB | 22.31GB | 3.65GB | 57.36GB | 40.83GB | 52.40GB | 4.09GB |
> | Memory w/. OEM | 53.06GB | 22.65GB | 3.67GB | 72.62GB | 63.46GB | 52.40GB | 4.65GB |
>
> The additional memory overhead primarily comes from storing intermediate features during inference to compute differences across cache intervals.
>
> On average, offline error modeling increases memory usage by only **15.8%** and modeling time by **16.8%** compared with random content generation, both remaining below 20%. Considering the substantial performance gains achieved by CEM and its zero inference‑time overhead, this cost is fully acceptable.
>
> **A3  2). Computational and memory overhead of dynamic programming for different models.**
>
> During inference, the computational overhead of our CEM primarily arises from loading the pre‑modeled error distribution and solving the dynamic programming optimization.
>
> The loading memory cost is **negligible**, requiring only a floating‑point array of size N × T (N: cache intervals, T: timesteps). The dynamic programming solver runs in milliseconds with at most T iterations, and its optimal caching strategy can be reused across generations with the same acceleration setting, further amortizing the overhead.
>
> **Table.K**: Online costs.
>
> |  |  |      Text2Image       |             |  |     Text2Video         |             | Class2Image |             |
> |:----------:|:--------------:|:-----------:|:-----------:|:---------------:|:-----------:|:-----------:|:----------------:|:-----------:|
> | Models | FLUX.1-dev | PixArt-$\alpha$ | SD15 | Hunyuan | Wan21 | OpenSora | DiT-DDPM | DiT-DDIM |
> | Time of DP | 1.10ms | 0.12ms | 0.80ms | 0.71ms | 0.85ms | 0.27ms | 6.96ms | 1.13ms |
> | Memory of error | 0.88KB | 0.35KB | 0.88KB | 0.88KB | 0.88KB | 0.53KB | 4.39KB | 0.88KB |

---

### Author Response · Authors · 2025-12-01
**Summary of Rebuttal for ACs**

Dear Area Chairs (ACs),

We sincerely appreciate your time and effort in reviewing our paper. To help you **quickly and clearly** understand our paper and the reviewers' concerns, we provide a brief summary here.

| Reviewer | Rating | Confidence | Feedback |
|:--------:|:------:|:----------:|:--------:|
| WEPH     |   4    |     4      |   None   |
| XEtZ     |   6    |     2      |   None   |
| Hw5z     |   4    |     3      | **Positive** |
| 98fT     |   6    |     3      |   None   |
| m1Xm     |   4    |     4      | Neutral  |

('None' denotes that we've not received feedback from the reviewer.)

Until now, we have addressed all five reviewers' concerns and received **positive recognition** from reviewer **Hw5z**.
Reviewer **m1Xm** had some misunderstandings, which we **have clarified** in our second-round response (no further feedback).

We have uploaded the **revised** version of the paper, with all modified sections and text highlighted in **blue**.


&nbsp;

---

&nbsp;


We have improved the following aspects:
* **More concise writing** (Sec.1, 3).
* **Broader applicability**.
    * Add **more quality metrics** (Tab.1, 3).
    * Add results on **TeaCache** (Tab.1, 2), **Wan2.1** (Tab.2), and under **more challenging scenarios** (Tab.13, 14, 15).
    * More comparisons with **training-based** methods (Tab.16) and **one-step diffusion** (Tab.17).
* **In-depth ablation**.
    * **Cost** analysis (Tab.5, 10, 12).
    * Add the **consistency** (Fig.3(a,b), Fig.6(b), Fig.8, Tab.7, 8, 9), **robustness** (Tab. 6) and **scalability** (App.B.6) of the error modeling.
    * Add **theoretical justification** (App.C.2), **practical quantification** (Fig.3(c), Fig.9, Tab.11) and **relationship analysis** (Fig.6(c)) of error approximation.

The following is a summary of our responses to **each reviewer**.

---

> ### Author Response · Authors · 2025-12-01
> **Summary of Reviewer WEPH (Rating: 4)**
>
> >W1/Q1: (**Performance**) Limited practical gains in several settings.
>
> * Some metrics are **insensitive** and add more metrics (**PSNR, SSIM, LPIPS**) (Tab.1, 3).
> * Add improvements under **higher** acceleration efficiencies (Tab. 13, 14).
> * We **Outperform** original fidelity on SD1.5, PixArt-$\alpha$, FLUX.1-dev and Hunyuan (Tab. 1, 2).
>
> >W2/W4: (**Robustness**) Representativeness and robustness of the offline estimate is unclear.
>
> * Provide the impact of different offline sample **distributions** (Tab. 7, Fig. 8) and sample **numbers** (Fig. 6(b), Tab. 8, 9).
> * Add the robustness under different **seeds, CFG, resolutions, frames** (Tab. 6).
>
> >W3: (**Costs**) The paper does not quantify this cost of offline modeling.
>
> * Compute **costs** of offline modeling (Tab. 5, 10) and dynamic‑programming strategy (Tab. 12).
>
> We have addressed all of the reviewer's concerns, we have **not received** any feedback.

---

> ### Author Response · Authors · 2025-12-01
> **Summary of Reviewer XEtZ (Rating: 6)**
>
> >W1/Q1: (**Robustness**) Whether the error model can be reused across different scenarios; how robust the method is.
>
> * Provide the impact of different offline sample **distributions** (Tab. 7, Fig. 8) and sample **numbers** (Fig. 6(b), Tab. 8, 9).
> * Add the robustness under different **seeds, CFG, resolutions, frames** (Tab. 6).
>
> >W2/Q2: (**Ablation**) Can you quantify the gap between the cumulative-error approximation and the true accumulated error.
>
> * Provide more interpretation of the cumulative error (Sec. 3.2).
> * **Quantify** the difference between the cumulative error and the actual error (Fig. 3(c), Fig. 9, Tab. 11).
> * Add the **theoretical proof** of the error upper bound (App. C.2).
>
> >W3/Q3: (**Experiments**) A comparison with online optimization methods is recommended.
>
> * Provide plug‑in results on the **TeaCache** baseline and **Wan2.1** model. (Tab. 1, 2, 13)
>
> >W5/Q4: (**Costs**) The offline cost per is not reported.
>
> * Compute costs of offline modeling (Tab. 5, 10).
>
> We have addressed all of the reviewer's concerns and received his/her recognition in the initial rating (rating: 6).

---

> ### Author Response · Authors · 2025-12-01
> **Summary of Reviewer Hw5z (Rating: 4, positive feedback)**
>
> >W1: (**Proof**) Limited theoretical analysis.
>
> * Provide **theoretical proof** of the error upper bound and the optimality of the dynamic programming (App. C.2).
> * Add **quantitative** comparison between the cumulative error and the true error (Fig. 3(c), Fig. 9, Tab. 11).
>
> >W2/Q1: (**Applicability**) Lack of applicability to one-step diffusion.
>
> * Comparison of fidelity, speed, and cost **highlights** the generalization and low cost of cache acceleration over one-step diffusion (Tab. 17).
> * Our Plug-in method needs a **base** model, but there is **no** work combining caching with one-step diffusion (App. D.5).
> * Propose a potential solution for one-step diffusion.
>
> >Q2: (**Ablation**) Can the authors quantify the relationship between cumulative error and perceptual generation quality.
>
> * Add **relationship** between errors and generation fidelity under different caching strategies (Fig. 6(c)).
> * Provide **robustness** under different seeds, CFG, resolutions, frames (Tab. 6).
>
> >Q3: (**Costs**) Provide details about the offline modeling cost and complexity.
>
> * Compute **costs** (Tab. 5, 10) and **Scalability** (App. B.6) of offline error modeling.
>
> Based on the reviewer's feedback, we have addressed his/her concerns and received his/her **recognition**.

---

> ### Author Response · Authors · 2025-12-01
> **Summary of Reviewer 98fT (Rating: 6)**
>
> >W1: (**Proof**) Limited theoretical justification for the cumulative error approximation.
>
> * Add **theoretical proof** of the error upper bound and the optimality of the dynamic programming (App. C.2).
>
> >W2: (**Costs, ablation**) Need cost of the offline modeling and the scalability.
>
> * Compute **costs** (Tab. 5, 10) and **Scalability** (App. B.6) of offline error modeling.
>
> >W3: (**Experiments**) Missing comparison with learned caching methods.
>
> * Provide comparisons with **training-based** methods (L2C, HarmoniCa) (Tab. 16).
>
> We have addressed all of the reviewer's concerns and received his/her recognition in the initial rating (rating: 6).

---

> ### Author Response · Authors · 2025-12-01
> **Summary of Reviewer m1Xm (Rating: 4)**
>
> >W1: (**Performance**) The acceleration of more powerful video generation models towards higher resolution should be more challenging and practical.
>
> * Add results on **higher** resolution and **longer** frames (Tab. 15) and on **Wan2.1** model (Tab. 2).
>
> >W2: (**Applicability**)Need the experiments on few-step diffusion.
>
> * Comparison of fidelity, speed, and cost **highlights** the generalization and low cost of cache acceleration over one-step diffusion (Tab. 17).
> * Our Plug-in method needs a **base** model, but there is **no** work combining caching with one-step diffusion (App. D.5).
> * Propose a potential solution for one-step diffusion.
>
> >W3: (**Experiments**) It is better to compare error compensation works.
>
> * Existing baselines such as ToCa, DuCa, and TaylorSeer are all error correction methods (Tab. 1, 2, 3).
> * Provide more results on the **TeaCache** baseline and **Wan2.1** model (Tab. 1, 2, 3, 13).
>
> >W4: (**Writing**) The motivation and method are a little bit trivial.
>
> * The writing logic has been refined, and several sections have been rewritten (Sec. 1, 3).
>
> >W5: (**Visualization**) Hope include more visualization with complex prompts.
>
> * Provide more visualization under **complex** prompts (Fig. 14).
>
> We have addressed all the concerns raised by the reviewer and **clarified** the **misunderstandings** in the feedback regarding application and error utilization:
> * For application, **no available base model** (combining caching with few step diffusion) for us to **plug into** (not due to our unwillingness).
> * For error utilization, we innovatively propose **joint error modeling** with cache intervals, **offline** modeling, and optimality strategy by **dynamic programming** (**not** as the reviewer claimed "not insightful").

---

### Meta-Review · Area_Chair_vKuQ · 2026-01-05

**Summary:**

Several reviewers asked questions on lack of evidence for strong improvements over baseliines, theoretical justification of the error accumulation approximation, and limited analysis of computational overhead. Hw5z confirmed that his concerns are mostly addressed in the rebuttal. Reviewer m1Xm kept his score after rebuttal commenting this the feature caching topic is crowded and paper does not make sufficient contributions.

**Reviewer Concerns:**

Authors seem did a good job on addressing most of the concerns from the reviewer (Hw5z, confirmed during discussion) and other reviewers (WEPH, XEtZ, 98fT).

Authors’ response to m1Xm’s question on combining the proposed method with few-step distilled models is not fully satisfying as there are methods trying to apply caching to few-step models showing improvements. This concern is still outstanding.

**Reviewer Scores:**

XEtZ and 98fT probably will keep their scores of 6.
Hw5z probably will increase scores to 6.
WEPH is hard to say.
m1Xm probably will stay at 4.

---

### Decision · Program_Chairs · 2026-01-26

Accept (Poster)